# LESS IS MORE: CLUSTERED CROSS-COVARIANCE CONTROL FOR OFFLINE RL

**Nan Qiao**[1,3]* **Sheng Yue**[2]† **Shuning Wang**[1,3], **Yongheng Deng**[3] **& Ju Ren**[3]
[1]Central South University, [2]Sun Yat-sen University, [3]Tsinghua University

## ABSTRACT

A fundamental challenge in offline reinforcement learning is distributional shift. Scarce data or datasets dominated by out-of-distribution (OOD) areas exacerbate this issue. Our theoretical analysis and experiments show that the standard squared error objective induces a harmful TD cross covariance. This effect amplifies in OOD areas, biasing optimization and degrading policy learning. To counteract this mechanism, we develop two complementary strategies: partitioned buffer sampling that restricts updates to localized replay partitions, attenuates irregular covariance effects, and aligns update directions, yielding a scheme that is easy to integrate with existing implementations, namely Clustered Cross-Covariance Control for TD ($C^4$). We also introduce an explicit gradient-based corrective penalty that cancels the covariance induced bias within each update. We prove that buffer partitioning preserves the lower bound property of the maximization objective, and that these constraints mitigate excessive conservatism in extreme OOD areas without altering the core behavior of policy constrained offline reinforcement learning. Empirically, our method showcases higher stability and up to $30\%$ improvement in returns over prior methods, especially with small datasets and splits that emphasize OOD areas. The implementation code is available at `https://github.com/NanMuZ/C4`.

## 1 INTRODUCTION

Offline reinforcement learning (RL) learns policies from fixed datasets without further interaction, which is essential when exploration is risky or expensive (Sutton & Barto, 2018). Although large benchmarks with millions of transitions report strong performance (Agarwal et al., 2019; Fujimoto & Gu, 2021; Kumar et al., 2020a; Chen et al., 2023), real deployments usually offer much smaller datasets with narrow state and action coverage (Nguyen-Tang & Arora, 2024; Cheng et al., 2023). Limited coverage enlarges out-of-distribution (OOD) areas and stresses standard training pipelines (Tkachuk et al., 2024; Foster et al., 2022; Jia et al., 2024).

To mitigate distribution shift and provide safety margins, conservative objectives have been widely adopted (Fujimoto & Gu, 2021; An et al., 2021; Lyu et al., 2022; Kostrikov et al., 2021a; Peng et al., 2019). These methods implicitly assume that the dataset covers the relevant parts of the space (Kumar et al., 2020a; Chen et al., 2024). Under weak coverage they can become overly cautious exactly where improvement is needed, and in extreme cases, this overconservatism can destabilize policy learning (Cheng et al., 2023; Li et al., 2023b).

A second and less analyzed failure mode arises from the value fitting objective itself. Under distribution shift and limited data, the procedure of temporal difference (TD) learning induces detrimental bias and feature co-adaptation that can culminate in training collapse, as noted by prior work (Kumar et al., 2022; Yue et al., 2023). We identify the core cause: TD learning that minimizes the second moment of the residual, $\mathbb{E}[\delta^2]$, generates a *harmful* cross-time covariance of gradient features, which becomes dominant under severe OOD area. Specifically, our theory and experiments show that in OOD areas, TD updates induce three implicit regularizers. Two are beneficial for generalization, akin to the beneficial implicit regularization produced by noise in supervised learning (Mulayoff & Michaeli, 2020; Damian et al., 2021). The third is a cross-time covariance of gradient features, and

---

*Part of this work was done when Nan Qiao and Shuning Wang worked at Tsinghua University.
†Corresponding to: `yuesh5@mail.sysu.edu.cn`

acts against the intended optimization objective and, under severe OOD, causes pronounced gradient interference and instability.

We address this challenge with two complementary strategies that operate locally on the geometry of the replay data. First, we partition the buffer by gradient features and train with single-cluster mini-batches, which removes the between-partition mean covariance. Second, we add an explicit gradient-based corrective penalty with a tunable coefficient that mitigates the covariance-driven bias within each update. To prevent conservative objectives from becoming over-restrictive in extreme out-of-distribution areas, we include a lightweight divergence-based term that is neutral on distribution and activates only in OOD areas, which reduces unnecessary suppression while preserving the core behavior of existing conservative methods.

Putting these pieces together, our contributions are threefold. First, we identify a data-limited failure mode in which the squared TD objective induces a harmful implicit regularizer that degrades generalization and can trigger training collapse. Second, we propose $C^4$, which constrains cross-region covariance to significantly curb this effect, and we introduce a gradient-based corrective penalty that further cancels within-cluster covariance. Third, while $C^4$ calls for small adjustments to sampling and loss in practice, it remains effectively "plug-and-play" for numerous offline RL algorithms with the optimization goals preserved. In experiments on small datasets and OOD-emphasized splits, $C^4$ delivers substantial and stable gains, with improvements exceeding 30% on several benchmarks.

## 2 RELATED WORK

**Reinforcement learning with small static datasets.** Traditional RL suffers from poor sample efficiency, and offline RL aims to address this issue by learning policies from fixed, pre-collected datasets without any interaction with the environment (Lillicrap et al., 2019; Lu et al., 2022). Under this offline learning paradigm, conventional off-policy RL approaches are prone to substantial value overestimation when there is a large deviation between the policy and data distributions (Kumar et al., 2020b; Qiao et al., 2026). Existing offline RL methods address this issue by following several directions, such as constraining the learned policy to be "close" to the behavior policy (Fujimoto et al., 2019; Kumar et al., 2019; Fujimoto & Gu, 2021), regularizing value function on OOD samples (Kumar et al., 2020a; Kostrikov et al., 2021b), enforcing strict in-sample learning (Brandfonbrener et al., 2021; Kostrikov et al., 2022), and performing pessimistic policy learning with uncertainty-based reward or value penalties (Yu et al., 2020; An et al., 2021). Most existing offline RL methods adopt the pessimism principle and avoid policy evaluation on OOD samples (Fujimoto et al., 2019; Yu et al., 2021a). This approach curbs error accumulation, but on small or weakly covered datasets, it can become overly conservative and cause large performance drops (Li et al., 2023b). This suggests a renewed bottleneck in sample efficiency. Recent work, such as DOGE and TSRL, mitigates the issue by admitting carefully chosen out-of-distribution samples, for example, those within a convex hull or those that are dynamics-explainable (Li et al., 2023b; Cheng et al., 2023). However, these methods operate at the level of data selection rather than RL itself.

**Implicit regularization in deep reinforcement learning.** Deep RL, driven by the deadly triad, often exhibits overestimation, out of distribution representation coupling, and value divergence, effects that intensify when data are scarce and noisy (Sutton & Barto, 2018; Baird, 1995; Tsitsiklis & Van Roy, 1997; van Hasselt et al., 2018; Yue et al., 2022; Li et al., 2023b). Classical stabilizers such as target networks, Double Q, TD3, and normalization, together with linear and dynamical analyses, mostly mitigate symptoms without addressing the mechanism that fuels self excitation and OOD coupling (Mnih et al., 2015; Hasselt, 2010; Fujimoto et al., 2018; Bhatt et al., 2024; Ioffe & Szegedy, 2015; Achiam et al., 2019). In the offline regime, policy constraint and pessimistic approaches regulate what is learned to reduce OOD evaluation errors and error accumulation (Fujimoto et al., 2019; Kumar et al., 2019; Fujimoto & Gu, 2021; Peng et al., 2019; Kumar et al., 2020b; An et al., 2021). DR3 instead acts on feature geometry by penalizing the inner product over two features, which reveals and counters an implicit regularizer implicated in these instabilities (Nikulin et al., 2023; Kang et al., 2023; Kumar et al., 2023; 2022). LayerNorm provides consistent stabilization with NTK and spectral contraction explanations and with scale decoupling that further weakens this effect, aligning with observations on representation stability and implicit bias (Ghosh & Bellemare, 2020; Kumar et al., 2021; Durugkar & Stone, 2018; Yue et al., 2023). Current work has not yet recognized that this effect can also be suppressed at the sampling level.

**Clustering-based reinforcement learning** Some recent work partitions heterogeneous offline datasets into interpretable behavior clusters to enable stable learning in local in-distribution regions (Mao et al., 2024; Wang et al., 2024; Hu et al., 2025; Wang et al., 2023). SORL alternates trajectory clustering and policy updates in an expectation maximization manner to reveal diverse high-quality behaviors (Mao et al., 2024). Behavior-aware deep clustering extracts near single-peaked subsets and improves stability and returns (Wang et al., 2024). Probabilistic approaches model latent behavior policies with Gaussian mixtures and derive closed-form improvement operators for the implicit clustering (Li et al., 2023a). Diffusion QL fits multi-peaked behavior policy distributions with diffusion models and mitigates mode mixing bias (Wang et al., 2023). Online skill discovery shows the value of learning separable skills in a latent space and informs the design of clustering offline (Achiam et al., 2018). However, most recent efforts focus on diverse policy training and have not yet connected to offline reinforcement learning under small datasets.

## 3 PRELIMINARY

**Offline reinforcement learning.** We consider the standard Markov decision process (MDP) $\mathcal{M} = (\mathcal{S}, \mathcal{A}, T, r, d_0, \gamma)$, with state space $\mathcal{S}$, action space $\mathcal{A}$, transition dynamics $T : \mathcal{S} \times \mathcal{A} \to \mathcal{P}(\mathcal{S})$, reward function $R : \mathcal{S} \times \mathcal{A} \to [0, 1]$, and initial state distribution $\mu : \mathcal{S} \to \mathcal{P}(\mathcal{S})$, where $\mathcal{P}(\mathcal{S})$ represents the set of distributions over $\mathcal{S}$. Subsequently, a policy $\pi(\mathbf{a} \mid \mathbf{s})$ induces the discounted occupancy $d^\pi(\mathbf{s}, \mathbf{a}) = (1 - \gamma) \sum_{t=0}^\infty \gamma^t \Pr(\mathbf{s}_t = \mathbf{s}, \mathbf{a}_t = \mathbf{a} \mid \pi)$ and maximizes the return $\mathbb{E}\left[ \sum_{t=0}^\infty \gamma^t r(\mathbf{s}_t, \mathbf{a}_t) \right]$ with $\mathbf{s}_{t+1} \sim T(\cdot \mid \mathbf{s}_t, \mathbf{a}_t)$. In the offline setting a fixed dataset $\mathcal{D} = \{(\mathbf{s}, \mathbf{a}, r, \mathbf{s}', \mathbf{a}')\}_{i=1}^{|\mathcal{D}|}$ is collected by a behavior policy $\pi_\beta$. Its empirical distribution $\hat{d}_\beta$ approximates $d^{\pi_\beta}$ and serves as the notion of data support. The key challenge is extrapolation error which tends to assign spuriously high values to actions outside the support of $\hat{d}_\beta$. This motivates us to design both policy evaluation and policy improvement to remain near the behavior distribution.

**Policy evaluation in offline reinforcement learning.** In offline RL, value functions are estimated from a fixed dataset. The $Q_\phi(\mathbf{s}, \mathbf{a})$ is obtained by solving the temporal-difference regression problem

$$\min_\phi \mathcal{L}_{\mathrm{TD}}(\phi) = \mathbb{E}_{(\mathbf{s}, \mathbf{a}, r, \mathbf{s}') \sim \mathcal{D}} \Big( Q_\phi(\mathbf{s}, \mathbf{a}) - \big[ r(\mathbf{s}, \mathbf{a}) + \gamma \, \mathbb{E}_{\mathbf{a}' \sim \pi(\cdot \mid \mathbf{s}')} Q_{\phi'}(\mathbf{s}', \mathbf{a}') \big] \Big)^2. \tag{1}$$

where $Q_{\phi'}$ denotes the target critic corresponding to $Q_\phi$. We have the temporal difference residual $\delta \equiv r(\mathbf{s}, \mathbf{a}) + \gamma \, \mathbb{E}_{\mathbf{a}' \sim \pi(\cdot \mid \mathbf{s}')} Q_{\phi'}(\mathbf{s}', \mathbf{a}') - Q_\phi(\mathbf{s}, \mathbf{a})$. Thus, the Problem (1) is equivalent to minimizing the second moment of the temporal difference residual $\delta$ under the dataset distribution, $i.e. \min \mathbb{E}\left[ \delta^2 \right]$. To make the evaluation robust near the dataset support, we reason about small perturbations of the critic through its layer features. For a unit direction $\mathbf{w}$ and a small magnitude $k \geq 0$, the first order expansion of the head gives

$$Q_\psi(x + k\mathbf{w}) \approx Q_\psi(x) + k \langle \mathbf{w}, \nabla_x Q_\psi(x) \rangle \quad \text{for } \psi \in \{\phi, \phi'\}. \tag{2}$$

**Policy improvement in offline reinforcement learning.** Offline RL improves the actor $\pi$ on the dataset states by maximizing expected value under the learned critic, *i.e.* $\max_\pi \mathbb{E}_{\mathbf{s} \sim \mathcal{D}} \mathbb{E}_{\mathbf{a} \sim \pi(\cdot \mid \mathbf{s})} \big[ Q_\phi(\mathbf{s}, \mathbf{a}) \big]$. To mitigate extrapolation error, we add an explicit proximity regularizer that keeps action selection close to the behavior policy, which yields the generic objective

$$\max_\pi \mathbb{E}_{\mathbf{s} \sim \mathcal{D}} \Big[ \mathbb{E}_{\mathbf{a} \sim \pi(\cdot \mid \mathbf{s})} Q_\phi(\mathbf{s}, \mathbf{a}) - \alpha \, D\big( \pi(\cdot \mid \mathbf{s}), \, \pi_\beta(\cdot \mid \mathbf{s}) \big) \Big], \tag{3}$$

where $\alpha > 0$ controls the regularization strength and $D(\cdot, \cdot)$ measures divergence or distance between action distributions. Different algorithms instantiate $D$ with KL or Rényi divergences and use MSE or MMD as practical proximity surrogates (Wu et al., 2019; Jaques et al., 2019; Metelli et al., 2020; Fujimoto & Gu, 2021; Kumar et al., 2019). In addition, some methods adopt implicit behavior regularization (Kumar et al., 2020b; Yu et al., 2021b; Lyu et al., 2022; An et al., 2021). We provide details of these behavior regularizers in the Appendix A.

## 4 CROSS COVARIANCE EFFECTS IN THE TD SECOND MOMENT

In this section, we turn our attention to a conventional technique from reinforcement learning, temporal difference learning, and analyze how its second moment changes when the evaluation point moves slightly in the feature space toward the OOD area, following two observations.

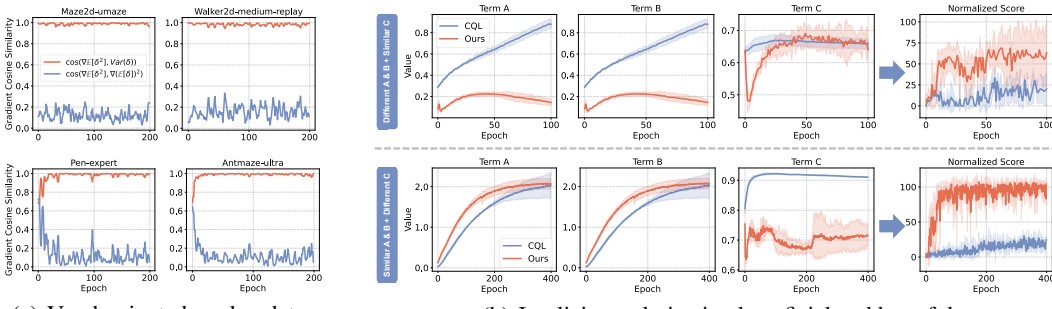

(a) Var dominated grad updates
(b) Implicit regularization beneficial and harmful

Figure 1: Left four panels report cosine similarities between $\nabla\mathbb{E}[\delta^2]$ and $\nabla\text{Var}[\delta]$ versus $\nabla(\mathbb{E}[\delta])^2$ under $\mathbb{E}[\delta^2] = (\mathbb{E}[\delta])^2 + \text{Var}[\delta]$, showing that the variance term dominates across benchmarks. Right four panels track $\text{Var}[\delta] \approx \gamma^2(k')^2 A + k^2 B - 2kk'C$ and the score, where larger $A$, $B$ and smaller $C$ correlate with better performance, indicating $A$ and $B$ act as beneficial implicit regularizers while $C$ is harmful.

**Observation 1: Variance of TD residual plays an important role in Problem** (1).
To connect the TD loss in Eq. (1) with a feature space view, we start from the identity

$$\mathbb{E}[\delta^2] = \big(\mathbb{E}[\delta]\big)^2 + \text{Var}[\delta], \tag{4}$$

which reduces the task to understanding how the variance changes under small displacements. As indicated by the left four panels of Fig. 1(a), our analysis focuses on how $\text{Var}[\delta]$ responds to small perturbations in feature space. To probe out of distribution directions while keeping the input domain of $Q_\phi$ fixed, we consider directional displacements $x \mapsto x + k\,\mathbf{w}$, which sets up the subsequent Taylor analysis. To probe out of distribution directions without changing the input domain of $Q_\phi$, we evaluate the heads at displaced features $x + k\mathbf{w}$. Then, using the first-order Taylor approximation, the sample variance of the Q-values at an OOD area, along $\mathbf{w}$ can be represented as

$$\begin{aligned}
\text{Var}\big(Q_\psi(x + k\mathbf{w})\big) &\approx \text{Var}\big(Q_\psi(x) + k\langle\mathbf{w}, \nabla_x Q_\psi(x)\rangle\big) \\
&= \text{Var}\big(Q(x) + k\langle\mathbf{w}, \nabla_x Q_\psi(x)\rangle\big) \\
&= k^2 \text{Var}\big(\langle\mathbf{w}, \nabla_x Q_\psi(x)\rangle\big).
\end{aligned} \tag{5}$$

Similarly, we have $\text{Var}\big(Q_{\phi'}(\mathbf{s}', \pi(\mathbf{s}'))\big) = (k')^2\text{Var}\big(\langle\mathbf{w}', \nabla_{x'} Q_{\phi'}(x')\rangle\big)$ where $x' = (\mathbf{s}', \mathbf{a}')$ denotes the next state action pairs of $x = (\mathbf{s}, \mathbf{a})$ come from the dataset (detailed proof in Appendix B). This keeps the analysis attached to the empirical support while we probe into support behavior virtually.

**Observation 2: Implicit regularization of covariance should be well controlled.**
We now state the main result that decomposes the variance change into a supervised style part and a term that is unique to temporal difference learning.

**Theorem 1.** *All expectations, variances, and covariances below are taken over $k, k', \mathbf{w}, \mathbf{w}'$. With the first order approximation for $Q_\phi$ in feature space, the variance satisfies*

$$\text{Var}[\delta] \approx \underbrace{\gamma^2(k')^2 \text{Var}\big(\langle\mathbf{w}', \nabla_{x'} Q_{\phi'}(x')\rangle\big) + k^2 \text{Var}\big(\langle\mathbf{w}, \nabla_x Q_\phi(x)\rangle\big)}_{\textit{implicit regularizer in noisy supervised learning, denote as Term(A) and Term (B)}}$$
$$\underbrace{- 2\gamma kk' \text{Cov}\big(\langle\mathbf{w}', \nabla_{x'} Q_{\phi'}(x')\rangle, \langle\mathbf{w}, \nabla_x Q_\phi(x)\rangle\big)}_{\textit{additional cross term unique to TD learning, denote as Term(C)}}. \tag{6}$$

*where $x$ and $x'$ are drawn from $\mathcal{D}$, with $x'$ being the next state action pair that follows $x$. By Eq. (4), this variance decomposition directly controls the TD second moment minimized by Eq. (1).*

*Sketch of proof.* Expand $Q_{\phi'}$ at $x'$ and $Q_\phi$ at $x$ using the first order rule in feature space, separate the zero displacement part and the linear part, apply variance rules and bilinearity of covariance, then drop higher order terms. Full details are given in the Appendix B.

Equation (6) has a natural interpretation. The first bracket penalizes large feature gradients and recovers the implicit regularizer from noisy supervised learning, which reduces $\text{Var}[Q]$ in out-of-distribution regions per Eq. (5) and improves generalization. Same as supervised learning, noise induces beneficial implicit regularization, and our effect is analogous(Mulayoff & Michaeli, 2020;

Damian et al., 2021). The second bracket is TD-specific because it couples feature gradients across a transition. This cross-term is misaligned with optimization and, since it enters Eq. (6) with a negative sign while the TD loss is minimized, updates tend to increase it, turning it into a harmful implicit regularizer that can drive collapse in pronounced OOD regimes. Fig. 1(b) corroborates this by showing that the first two terms act beneficially while the cross term grows under TD minimization, with $A$, $B$, and $C$ approximated by traces of denoised gradient covariance.

Although Eq. (6) may look close to the conclusion of (Kumar et al., 2022; Yue et al., 2023), our analysis and takeaways are different. We derive the result directly from the TD loss and the second moment identity rather than from Lyapunov style or gradient stability arguments, and we do not assume optimizer specific behavior or noise alignment. Our decomposition retains two next state contributions that are missing in prior work, namely $\mathrm{Var}\big(\langle \mathbf{w}', \nabla_{x'} Q_{\phi'}(x') \rangle\big)$ and its scaling by $\gamma^2 k'^2$, which clarify when sensitivity to the future head dominates even if current state terms are controlled. We also pair $(x, x')$ from the dataset instead of policy rollouts so the analysis stays on empirical support while out of distribution effects are introduced by virtual feature displacements inside the head. In experiments, this pairing choice and the explicit treatment of the next state variance and the TD cross covariance improve stability and returns.

## 5 $C^4$: CLUSTERED CROSS-COVARIANCE CONTROL FOR TD

This section turns the TD-variance model into two control objectives and develops an EM-style procedure that clusters gradient pairs and samples within a single cluster per update. Subsection 5.1 derives a matrix target from TD variance and sets the size and sign control objectives. Subsection 5.2 introduces clustering of stacked gradient pairs and shows why single-cluster sampling removes the between-cluster driver while within-cluster alignment controls the sign. Subsection 5.3 specifies a mixture-regularized objective, minibatch estimators, and the overall training procedure. All formal proofs are deferred to Appendix C.

### 5.1 PROBLEM FORMULATION: FROM TD VARIANCE TO A MATRIX TARGET

For a transition with feature-space gradients $g' = \nabla_{x'} Q_{\phi'}(x')$ and $g = \nabla_{x_i} Q_\phi(x)$, the one–step variance admits

$$\mathrm{Var}[\delta_i] \approx \underbrace{\gamma^2 k'^2 \, \mathrm{Var}\big(\langle \mathbf{w}', g' \rangle\big) + k^2 \, \mathrm{Var}\big(\langle \mathbf{w}, g \rangle\big)}_{\text{implicit regularizer as in noisy supervised learning}} - \underbrace{2\gamma k k' \, \mathrm{Cov}\big(\langle \mathbf{w}', g' \rangle, \langle \mathbf{w}, g \rangle\big)}_{\text{TD cross term}}, \quad (7)$$

and, under a minibatch sampling law,

$$\mathrm{Cov}\big(\langle \mathbf{w}', g' \rangle, \langle \mathbf{w}, g \rangle\big) \le \|C\|_2 \ \le \ \|C\|_F, \qquad C = \mathrm{Cov}(g', g) \in \mathbb{R}^{m \times m}, \quad (8)$$

Thus, to make the TD cross term harmless, we (i) shrink a size proxy of $C$ (trace/spectral norm). , and (ii) adding a penalty to offset the covariance $-2\gamma k k' \mathrm{Cov}\big(\langle \mathbf{w}', g' \rangle, \langle \mathbf{w}, g \rangle\big)$.

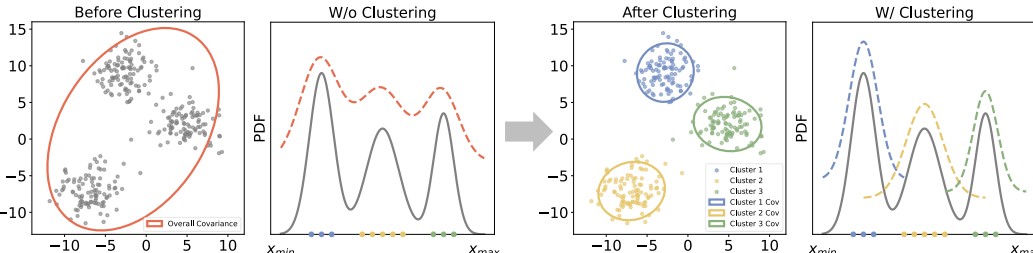

Figure 2: Intuition behind clustering and TD covariance. The left panel shows that without clustering, the overall covariance ellipse mixes within-cluster spread and between-cluster offsets, so the TD cross term couples unrelated modes and its sign can drift. Right panel clusters the stacked gradients $y = [g', g]$ and samples each minibatch from a single cluster, which removes the between-cluster driver and leaves updates governed by local within-cluster covariance $C_z$. The result is more local TD updates, weaker spurious coupling across modes, and improved stability in OOD directions.

## 5.2 CLUSTERING THE STACKED GRADIENT PAIRS

Building on the matrix–target formulation in Section 5.1, where the TD variance decomposes as in Eq. (7) and the cross term is controlled by the matrix $C$ via Eq. (8), Fig. 2 shows that treating the whole dataset as a single cloud mixes within–cluster spread and between–cluster offsets, which makes the TD cross term unstable. This motivates clustering the stacked gradient pairs and sampling single–cluster minibatches so that updates are governed by local within–cluster statistics.

Let $y_i = [g', g] \in \mathbb{R}^{2m}$ and partition the dataset into $K$ clusters in $y$-space. For cluster $z$, define means $\mu'_z, \mu_z$, variances $\Sigma'_z, \Sigma_z$, and cross covariance $C_z = \mathrm{Cov}(g', g \mid Z = z)$.

**Theorem 2** (Single-cluster sampling removes the between-cluster driver). *With cluster label $Z$, the cross covariance decomposes as*

$$C = \mathbb{E}\big[ C_Z \big] \ + \ \mathrm{Cov}\big(\mu'_Z, \mu_Z\big). \tag{9}$$

*If each minibatch is drawn from a single cluster $z$, the between-cluster term in Eq. (9) vanishes in that batch and, for any unit $\mathbf{w}', \mathbf{w}$,*

$$\big| -2\gamma kk' \mathrm{Cov}\big(\langle \mathbf{w}', g' \rangle, \langle \mathbf{w}, g \rangle\big)\big| \ \leq \ 2\gamma kk' \|C_z\|_2 \ \leq \ 2\gamma kk' \sqrt{\mathrm{tr}\,\Sigma'_z} \sqrt{\mathrm{tr}\,\Sigma_z}. \tag{10}$$

*Sketch of proof. Apply the law of total covariance to obtain Eq. (9). Then use $|a^\top M b| \leq \|M\|_2$ and the operator Cauchy–Schwarz inequality.*

The theorem splits the source of the cross covariance into a within–cluster component and a between–cluster component. If each minibatch is drawn from a single cluster, the between–cluster term vanishes in that update, so the TD cross term is fully determined by the within–cluster statistics $C_z$. This yields the batch–level bound $2\gamma kk' \|C_z\|_2 \leq 2\gamma kk' \sqrt{\mathrm{tr}\,\Sigma'_z} \sqrt{\mathrm{tr}\,\Sigma_z}$. Thus single–cluster sampling together with a penalty on $\|C_z\|$ stabilizes TD updates by suppressing the harmful cross term to a scale controlled by within–cluster variances.

## 5.3 MIXTURE-REGULARIZED OBJECTIVE AND TRAINING PROCEDURE

We fit a $K$–component Gaussian mixture on $\{y_i\}$ with parameters

$$p(y) = \sum_{z=1}^{K} p_z\, \mathcal{N}\big(y \mid \mu_z, \Omega_z\big), \qquad \mu_z = \begin{bmatrix} \mu'_z \\ \mu_z \end{bmatrix}, \ \ \Omega_z = \begin{bmatrix} \Sigma'_z & C_z \\ C_z^\top & \Sigma_z \end{bmatrix}. \tag{11}$$

Coupling TD fitting with a spectral proxy gives

$$\min_{\phi, \{p_z, \mu_z, \Omega_z\}} \quad \mathcal{L}_{\mathrm{TD}}(\phi) + \lambda \sum_{z=1}^{K} p_z \|C_z\|_F^2 \quad \text{s.t.} \quad p_z \geq 0, \ \sum_z p_z = 1, \ \Omega_z \succ 0, \tag{12}$$

where $\|C_z\|_F^2 = \mathrm{tr}(C_z C_z^\top)$ upper bounds $\|C_z\|_2^2$ and is easy to estimate per minibatch.

Given a single-cluster minibatch $B \subset z$, we estimate

$$\widehat{C}_z(B) = \mathrm{Cov}_B(g', g), \qquad \widehat{\mathcal{R}}_{\mathrm{cross}}(B) = \|\widehat{C}_z(B)\|_F^2 + \beta \big(\mathrm{tr}\,\widehat{C}_z(B)\big)^2, \tag{13}$$

$$\mathcal{J}(\phi, B) = \mathcal{L}_{\mathrm{TD}}(\phi, B) \ + \ \lambda \widehat{\mathcal{R}}_{\mathrm{cross}}(B), \tag{14}$$

which implements Eq. (12) stochastically and, by Eq. (10), controls the harmful term in each update.

$C^4$ repeatedly clusters stacked gradient pairs to estimate within-cluster cross covariance $C_z$ and then performs critic updates using single-cluster minibatches. Each update minimizes the TD loss plus a Frobenius-style penalty on $\widehat{C}_z$. By Theorem 2, this bounds the harmful cross-term batch-wise. The result is a minibatch distribution tailored to reduce $\|C\|$ and stabilize TD in OOD directions.

## 6 TRAINING ON CLUSTERED BUFFERS

The last section examines periodic dataset clustering to control cross-covariance. The clustering design remains an open challenge. In offline RL, TD updates occur during evaluation, and policy

---

**Algorithm 1** $C^4$: Single-Cluster Offline Update for TD (EM-style)

---

**Input** offline dataset $\mathcal{D}$, number of clusters $K$, regularizers $\lambda, \beta$, iterations $T$.
**Initialize** mixture $\{p_z, \mu_z, \Omega_z\}_{z=1}^K$, critic $\phi$.
**for** $t = 1, \ldots, T$ **do**
*Compute gradients:* for each sample $i$, form $g'$, $g$ and stack $y_i = [g', g]$.
*E–step:* $r_{iz} \propto p_z \mathcal{N}(y_i \mid \mu_z, \Omega_z)$, normalize $\sum_z r_{iz} = 1$.
*M–step:* $p_z \leftarrow \frac{1}{n} \sum_i r_{iz}, \quad \mu_z \leftarrow \frac{1}{N_z} \sum_i r_{iz} y_i, \quad \Omega_z \leftarrow \frac{1}{N_z} \sum_i r_{iz} (y_i - \mu_z)(y_i - \mu_z)^\top + \epsilon I$, extract $C_z$.
*Critic minibatch:* sample cluster $z \sim \text{Cat}(\{p_z\})$, draw minibatch $B$ with weights $r_{iz}$.
*Penalty and update:* compute $\widehat{C}_z(B) = \text{Cov}_B(g', g)$, minimize batch loss $\mathcal{J}(\phi, B)$ in Eq. (14).
**end for**
**Output** trained critic $\phi$ and mixture $\{p_z, \mu_z, \Omega_z\}$.

---

improvement imposes policy constraints. Periodic clustering can reshape data geometry and shift support across clusters, which may compromise these constraints during improvement. Designing clustering that preserves them is an important direction.

To this end, this section explains why training with *clustered buffers* (single-cluster minibatches as in Algorithm 1) has limited impact on the policy improvement objective and, in fact, provably *optimizes a computable lower bound* of the canonical mixture objective. We specialize the discussion to the CQL-style improvement surrogate and connect each step to Appendix D.

**CQL improvement target and a global lower bound.** For a policy $\pi$ and state $s$, consider the transformed CQL target

$$\mathcal{U}_{\text{CQL}}(\pi\,;s) := V^\pi(s) - \alpha \left( (I - \gamma P^\pi)^{-1} \chi^2(\pi \| \pi_\beta) \right)(s), \tag{15}$$

where $\chi^2$ is the Pearson divergence (Definition 1). Lemma 2 in the Appendix implies the *statewise* lower bound

$$\mathcal{U}_{\text{CQL}}(\pi\,;s) \geq \mathbb{E}_\pi[Q(s,a)] - \bar{\rho} \alpha \sup_{s'} \chi^2(\pi \| \pi_\beta)(s'), \qquad \bar{\rho} = \frac{1}{1-\gamma}, \tag{16}$$

and the same form holds when a KL penalty or its nonnegative combination is used. Thus, within a short local window where $V^\pi$ is linearized by $\mathbb{E}_\pi[Q]$, policy updates that increase the r.h.s. of Eq. (16) improve a *global* lower bound to Eq. (15). This shows that replacing the long-horizon transformed penalty by a tractable per-state divergence primarily tightens the bound and does not qualitatively alter the improvement direction.

**Mixtures, clustering, and why per-cluster training is safe.** Let the behavior be a mixture $\nu = \sum_{m=1}^M w_m \nu_m$ (e.g., a Gaussian mixture induced by clustered replay). Lemma 3 in the Appendix shows $f$-divergences are convex in the *second* argument:

$$D_f(\pi \| \nu) \leq \sum_{m=1}^M w_m D_f(\pi \| \nu_m), \tag{17}$$

with the two special cases (Pearson, KL) holding verbatim. Plugging Eq. (17) into Eq. (16) yields the *cluster-decomposed* lower bound

$$\mathbb{E}_\pi[Q(s,a)] - \bar{\rho}\left( \alpha \sum_m w_m \chi^2(\pi\|\nu_m)(s) + \beta \sum_m w_m \text{KL}(\pi\|\nu_m)(s) \right) \tag{18}$$

for any nonnegative combination of Pearson and KL. Consequently, optimizing the per-cluster surrogate

$$\mathcal{J}_z(\pi) := \mathbb{E}_\pi[Q(s,a)] - \bar{\rho}\left( \alpha \chi^2(\pi\|\nu_z)(s) + \beta \text{KL}(\pi\|\nu_z)(s) \right) \tag{19}$$

and sampling $z \sim w$ gives an *unbiased* stochastic gradient of the weighted sum $\sum_m w_m \mathcal{J}_m(\pi)$ (Lemma 3 in Appendix), which is a computable lower bound to the mixture objective with $\nu$ inside each divergence. Thus, *partitioning the buffer and training per cluster does not bias the direction*: it maximizes a principled lower bound to the original improvement target, with the gap controlled by divergence convexity. Additionally, the $f$-divergence case is illustrative rather than exclusive. Any policy constraint satisfying $D(\pi\| \sum_z w_z \nu_z) \leq \sum_z w_z D(\pi\|\nu_z)$ can be used in our algorithm. See Appendix A for admissible $D$ choices.

**Stability from adding KL and quantitative caps.** Proposition 3 in the Appendix proves that adding a KL term yields strong concavity of the local surrogate and quantitative step caps:

$$\text{near } \theta_\beta: \quad \text{strong concavity} \geq m\beta\bar{\rho}, \qquad \|\theta^\star - \theta_\beta\| \leq \frac{\|\nabla_\theta \mathbb{E}_\pi[Q]\big|_{\theta_\beta}\|}{m\beta\bar{\rho}}. \tag{20}$$

For the Gaussian-mean case,

$$\mu^\star = \mu_\beta + \kappa^\star \Sigma_\beta g, \qquad (2\alpha\bar{\rho}\, e^{\kappa^{\star 2} R} + \beta\bar{\rho})\, \kappa^\star = 1, \qquad \kappa^\star \leq \frac{1}{\beta\bar{\rho}} = \frac{1-\gamma}{\beta}, \tag{21}$$

and Pearson inflation is bounded by $\chi^2(\pi_{\mu^\star}\|\pi_\beta) \leq \beta/(2\alpha) - 1$ when $\alpha > 0$. By convexity in the second argument (Lemma 3), the same caps hold *per cluster* and therefore under cluster sampling in expectation. This shows that partitioning the buffer *does not* compromise the known CQL stabilizing effects. If anything, it makes the constants local and often tighter.

**Takeaway for practice.** Equations (16)–(18) show that training with clustered buffers maximizes a certified lower bound to the original mixture objective (mixture placed inside the divergence), with unbiased gradients under random cluster selection. KL caps Eq. (20)–(21) carry over per cluster, so the induced change to the policy improvement direction is small (same direction, cluster-adaptive step). In the CQL special case, the $\bar{\rho}$-weighted divergence still upper-bounds the propagated penalty, and convexity guarantees let us replace the mixture by a sum over cluster penalties without loosening control. Therefore, splitting the buffer into clusters has a limited effect on the improvement rule while *improving* its computability and stability, exactly the properties exploited by $C^4$ in Algorithm 1.

**Pointers to Appendix.** Formal statements and proofs are in Appendix D: Lemma 2 (global operator freeze), Lemma 3 (convexity in the second argument and unbiased cluster gradients), Proposition 2 (isotropic penalties and local update direction), and Proposition 3 (KL stabilization and caps). Remark 2 quantifies poor-coverage regimes and explains why adding KL prevents runaway steps. The same reasoning applies per cluster.

## 7 EXPERIMENTAL RESULTS

In this section, we empirically evaluate our proposed method $C^4$. The experiments are organized to address the following questions:

- Q1. How does it perform on offline RL relative to existing approaches across standard benchmarks, particularly under reduced-data regimes?
- Q2. How is performance influenced by factors such as the number of initial clusters and the quality of the data?
- Q3. Can the plug-and-play method $C^4$ adapt to different types of algorithms?

### 7.1 IMPLEMENT

For a fair and comprehensive evaluation, we compare our method against Behavior Cloning (BC) and a broad set of state-of-the-art offline reinforcement learning algorithms. For standard offline RL backbones, we include CQL (Kumar et al., 2020a), TD3+BC (or TD3BC) (Fujimoto & Gu, 2021), and IQL (Kostrikov et al., 2022). To directly probe performance in reduced-data regimes (Q1), we further consider data-efficient algorithms such as DOGE (Li et al., 2023b) and TSRL (Cheng et al., 2023). DOGE is selected for its strong out-of-distribution generalization through state-conditioned distance functions, while TSRL exploits temporal symmetry in system dynamics to improve sample efficiency. In addition, we include recent high-performing methods, BPPO (Zhuang et al., 2023), which leverages PPO-style clipping for monotonic improvement, and A2PR (Liu et al., 2024), which employs adaptive regularization with a VAE-augmented policy.

We further study the plug-and-play nature of $C^4$ and its relation to other regularization techniques by comparing against methods that act as generic regularizers or share similar design principles. In particular, to mitigate gradient collapse in sparse data regimes, we evaluate DR3 (Kumar et al., 2022) and LN (Yue et al., 2023). We additionally consider SORL (Mao et al., 2024), which is closely

Table 1: Normalized scores on MuJoCo locomotion tasks using reduced-size datasets (10k samples). Abbreviations fr, mr, and me denote full-replay, medium-replay, and medium-expert, respectively.

| Task | TD3+BC | CQL | IQL | DOGE | BPPO | TSRL | A2PR | Ours |
|---|---|---|---|---|---|---|---|---|
| Ant-me | 52.0±18.2 | 74.0±25.0 | 66.0±10.5 | 82.0±16.4 | 85.5±13.7 | 83.6±12.4 | 66.7±10.2 | **100.9±5.0** |
| Ant-m | 46.0±17.4 | 62.0±22.1 | 56.0±9.3 | 69.0±15.2 | 78.0±11.9 | 72.2±10.6 | 64.0±9.3 | **84.5±6.1** |
| Ant-mr | 31.0±15.5 | 36.0±16.3 | 41.0±10.8 | 46.0±12.9 | 52.0±10.8 | 49.4±13.1 | 44.0±9.1 | **65.8±6.9** |
| Ant-e | 72.0±25.0 | 94.0±29.4 | 82.0±15.0 | 98.0±20.3 | 103.0±11.2 | 100.7±12.6 | 88.0±10.6 | **109.6±2.7** |
| Ant-fr | 70.0±24.0 | 92.0±26.5 | 80.0±15.0 | 96.0±20.0 | 102.0±11.7 | 99.8±12.0 | 86.0±10.4 | **107.6±3.2** |
| Hopper-m | 30.7±13.2 | 50.1±22.3 | 61.0±6.2 | 55.6±8.3 | 55.0±7.8 | 60.9±4.1 | 55.9±8.4 | **69.2±12.7** |
| Hopper-mr | 11.3±4.7 | 13.2±2.0 | 16.2±3.0 | 19.1±3.3 | 45.1±8.7 | 23.5±8.8 | 12.5±5.9 | **45.9±8.4** |
| Hopper-me | 22.6±13.9 | 43.2±6.9 | 51.7±7.0 | 36.8±34.5 | 27.9±15.2 | 56.6±13.9 | 49.7±10.7 | **81.3±6.0** |
| Hopper-e | 53.6±17.1 | 56.1±26.4 | 60.9±9.6 | 62.2±21.7 | 85.0±17.9 | 76.7±20.4 | 80.0±16.8 | **107.0±2.8** |
| Hopper-fr | 32.0±13.5 | 45.0±22.0 | 56.0±6.3 | 54.0±8.4 | 60.0±9.7 | 53.4±11.3 | 55.0±8.9 | **65.3±9.4** |
| Walker2d-m | 11.2±19.2 | 54.1±15.5 | 34.2±5.2 | 53.7±12.6 | 54.7±11.4 | 47.3±10.1 | 5.9±5.2 | **65.9±7.8** |
| Walker2d-mr | 9.3±6.6 | 13.8±5.3 | 17.7±8.9 | 15.5±9.2 | 29.5±8.7 | 27.6±12.4 | 34.4±8.9 | **55.4±5.9** |
| Walker2d-me | 12.4±15.7 | 26.0±14.0 | 38.0±12.2 | 42.5±11.4 | 61.3±12.2 | 50.9±26.4 | 56.5±11.5 | **96.3±10.4** |
| Walker2d-e | 29.5±23.5 | 56.0±29.4 | 16.2±3.2 | 81.2±18.6 | 102.0±9.7 | 104.9±10.6 | 98.0±7.9 | **109.5±0.3** |
| Walker2d-fr | 14.2±19.5 | 55.0±16.0 | 36.0±5.6 | 55.5±12.3 | 52.0±10.9 | 44.3±10.4 | 48.0±9.6 | **77.3±7.1** |
| Halfcheetah-m | 25.9±8.4 | 41.7±2.2 | 35.6±2.9 | 42.8±2.9 | 28.5±3.2 | 43.3±2.8 | 37.1±2.7 | **46.3±3.1** |
| Halfcheetah-mr | 29.1±8.3 | 16.3±4.9 | 34.1±6.3 | 26.3±3.1 | 34.4±4.2 | 27.7±3.8 | 23.6±4.7 | **43.1±5.3** |
| Halfcheetah-me | 23.5±13.6 | 39.7±6.4 | 14.3±7.3 | 33.1±8.8 | 22.3±9.6 | 37.2±14.9 | 32.4±8.3 | **46.0±3.5** |
| Halfcheetah-e | 26.4±4.2 | 5.8±1.3 | -1.1±3.8 | 1.4±3.1 | 6.5±3.4 | 42.0±26.4 | 36.0±7.8 | **75.8±5.2** |
| Halfcheetah-fr | 28.0±8.6 | 45.0±2.4 | 33.0±3.0 | 43.0±3.1 | 37.0±3.6 | 41.0±3.0 | 39.0±4.1 | **58.1±3.4** |
| Locomotion-Avg. | 31.5 | 46.0 | 41.4 | 50.7 | 56.1 | 57.2 | 50.6 | **75.7** |
| AntMaze-Avg. | 6.3 | 10.7 | 21.4 | 20.5 | 16.5 | 22.0 | 16.2 | **27.0** |
| Maze2D-Avg. | 49.7 | 57.4 | 103.6 | 106.1 | 120.6 | 115.7 | 111.4 | **126.9** |
| Adroit-Avg. | 1.4 | 7.3 | 15.2 | 8.4 | **23.1** | 15.7 | -0.1 | 21.6 |

related in motivation as it performs data clustering, but does so at the trajectory level rather than in the gradient space as $C^4$ does. Across all comparisons, $C^4$ is instantiated as a plug-in module on top of existing backbones, allowing us to isolate its effect on performance.

## 7.2 MAIN RESULTS

To answer Q1, we focus our primary evaluation on a *data-scarce regime*. Specifically, we restrict each D4RL MuJoCo locomotion task to only **10k** state-action pairs (approximately 1% of the full dataset). This setting places all methods in a challenging low-data regime, thereby highlighting their generalization capabilities. As summarized in Table 1, we benchmark a wide spectrum of algorithms, including standard backbones (TD3+BC, CQL, IQL), data-efficient methods (DOGE, TSRL), and stronger OOD-aware baselines (BPPO, A2PR). Table 1 primarily reports the normalized scores on the locomotion benchmarks and summarizes the main results on AntMaze, Maze2D, and Adroit, while detailed per-task results for these three domains are deferred to Tables 3, 4, and 5 in Appendix E. To complement the tabular comparison, Figure 3 provides a holistic visualization by plotting normalized scores across tasks. On the MuJoCo locomotion benchmarks, despite the extreme data sparsity, our approach recovers nearly 75% of expert performance on average and consistently outperforms all competing baselines, achieving an average improvement of more than **30%** over the best alternative. This demonstrates that $C^4$ substantially enhances data efficiency in offline RL.

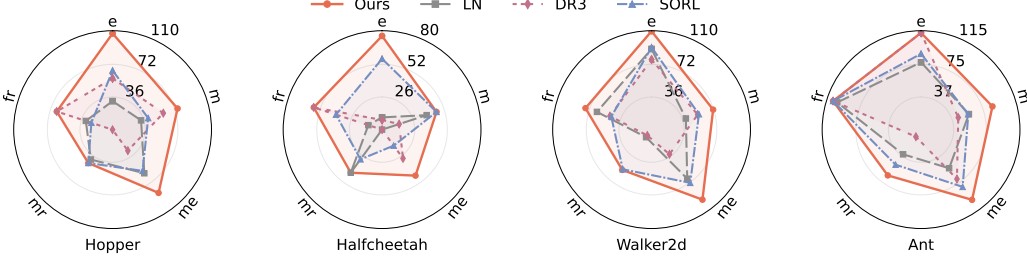

Figure 3: Radar charts comparing normalized scores on D4RL MuJoCo locomotion tasks (10k samples).

To examine Q2, we move beyond raw performance and analyze computational efficiency and sensitivity to key hyperparameters. Figure 4 reports wall-clock training time over 300K optimization steps, comparing both full algorithms and plug-in regularizers. Incorporating $C^4$ into standard

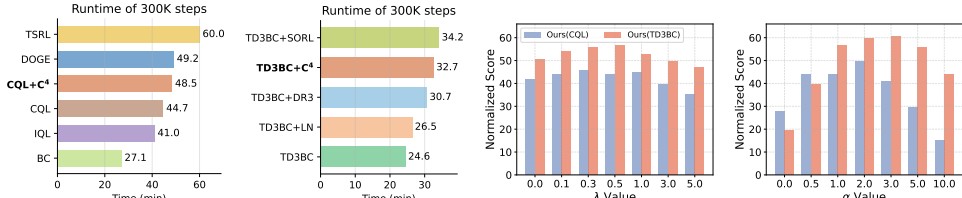

Figure 4: Wall-clock runtime comparisons, and performance sensitivity to hyperparameters $\lambda$ and $\alpha$.

backbones such as CQL and TD3+BC introduces only moderate overhead, yielding a runtime profile comparable to other lightweight regularizers (LN, DR3, SORL) and substantially more efficient than complex data-efficient baselines such as TSRL and DOGE. We also study the sensitivity of $C^4$ to its regularization strength $\lambda$ and the base algorithm coefficient $\alpha$. Performance remains stable over a broad range of these values, indicating that $C^4$ does not require delicate tuning. Additional ablations in Appendix E further show that reasonable variations in the number of initial clusters and the quality/coverage of the dataset lead to smooth changes in performance, supporting the robustness of the gradient-space clustering mechanism.

Finally, to directly address Q3 regarding the plug-and-play property of $C^4$, we evaluate it as an add-on regularizer for both CQL and TD3+BC. Using identical training protocols, we compare each backbone to its variants augmented with LN, DR3, and $C^4$ (Figures 5 and 6). Across both algorithm families, incorporating $C^4$ yields the most consistent and substantial improvements throughout training, whereas LN and DR3 provide only moderate or task-dependent gains. This indicates that $C^4$ effectively targets the variance components in gradient space that limit offline RL performance, while preserving the inductive biases of the underlying algorithm. In practice, $C^4$ can therefore be treated as a drop-in module that robustly enhances a variety of existing offline RL methods.

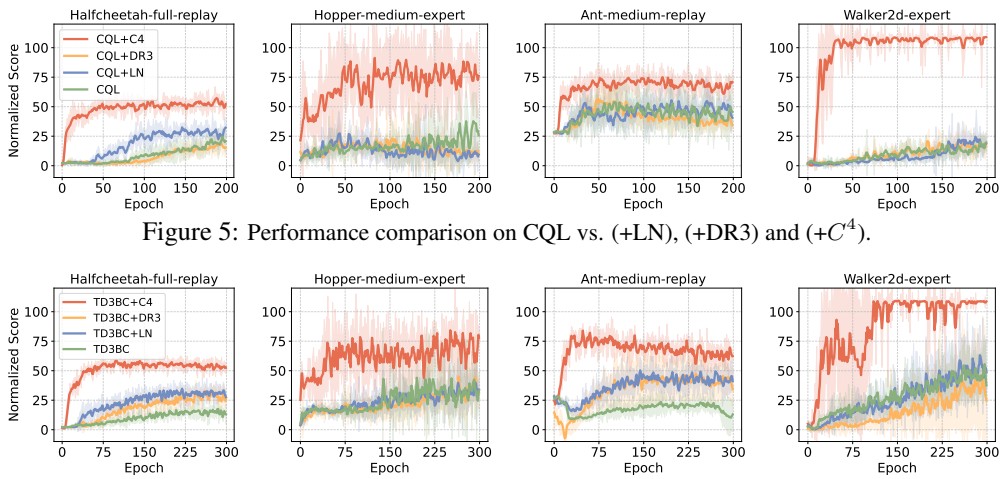

Figure 5: Performance comparison on CQL vs. (+LN), (+DR3) and (+$C^4$).

Figure 6: Performance comparison on TD3BC vs. (+LN), (+DR3) and (+$C^4$).

Additionally, extended implementation, benchmark descriptions, experimental details, complete results, and ablations are provided in Appendix E due to space constraints.

## 8 CONCLUSION

This work identifies harmful TD cross covariance as a key driver of instability under weak coverage in offline RL. $C^4$ counters this effect with partitioned buffer sampling that localizes updates and with an explicit gradient-based penalty that offsets bias while preserving the objective's lower bound. The method reduces excessive conservatism, improves stability, and integrates with standard pipelines without heavy tuning. Experiments on small data and splits that emphasize out-of-distribution states show consistent gains in return, with improvements up to about 30% and smoother learning dynamics. These results indicate that clustered cross-covariance control for TD is a practical and effective approach for achieving robust offline RL under weak coverage.

## 9 ACKNOWLEDGEMENTS

This research was supported in part by the National Natural Science Foundation of China under Grants 62432004 and 62572496, the Shenzhen Science and Technology Program under Grant JCYJ20250604175500001, the Young Elite Scientist Sponsorship Program by CAST under Contract ZB2025-218, and a grant from the Guoqiang Institute, Tsinghua University.

## 10 REPRODUCIBILITY STATEMENT

We release the full codebase in supplementary files. The repository contains scripts for end-to-end runs, environment setup instructions, and exact configurations. The paper provides the algorithm pseudo-code and a complete list of hyperparameters. We will include data preparation steps, evaluation scripts, random seed control, and instructions to regenerate all main tables and figures.

## 11 ETHICS STATEMENT

This work targets decision-making under distribution shift in offline reinforcement learning. The method can improve reliability, yet misuse may create harm. Potential risks include job displacement, unsafe behaviors in autonomous systems, privacy exposure when training on sensitive data, and misleading outputs from generative components. We recommend careful auditing, limited deployment in controlled settings, and human oversight. Use should focus on supporting decisions rather than replacing human judgment. Privacy protection, transparency, continuous monitoring, and rollback procedures are necessary. We will provide a usage checklist to encourage responsible application.

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

# OUTLINE

## A  COMMON POLICY PROXIMITY CONSTRAINTS IN OFFLINE RL

In this section, we measure per–state proximity between the learned policy $\pi$ and the behavior policy $\pi_\beta$ and average over the dataset state distribution $\rho_\mathcal{D}$, and the global constraint is

$$\mathcal{C}(\pi\|\pi_\beta) = \mathbb{E}_{s\sim\rho_\mathcal{D}}\left[D\big(\pi(\cdot\mid s),\|,\pi_\beta(\cdot\mid s)\big)\right]. \tag{22}$$

Furthermore, the mixture upper bound we will check is $D\left(\pi\middle\|\sum_i w_i\pi_{\beta_i}\right) \leq \sum_i w_i D(\pi\|\pi_{\beta_i}), w_i \geq 0, \sum_i w_i = 1$ in Corollary 1.

**KL.** The Kullback–Leibler divergence is

$$D_{\mathrm{KL}}(\pi\|\pi_\beta) = \int_\mathcal{A} \pi(a\mid s)\log\frac{\pi(a\mid s)}{\pi_\beta(a\mid s)}\mathrm{d}a, \tag{23}$$

finite only when $\mathrm{supp}(\pi) \subseteq \mathrm{supp}(\pi_\beta)$, and the reverse form $D_{\mathrm{KL}}(\pi_\beta\|\pi)$ is behavior–sample estimable, and as an $f$–divergence KL is convex in the second argument, hence it satisfies $D_{\mathrm{KL}}\left(\pi\middle\|\sum_i w_i\pi_{\beta_i}\right) \leq \sum_i w_i D_{\mathrm{KL}}(\pi\|\pi_{\beta_i})$.

**Rényi.** The Rényi divergence of order $\alpha > 0, \alpha \neq 1$ is

$$D_\alpha^{\mathrm{R}}(\pi\|\pi_\beta) = \frac{1}{\alpha-1}\log\int_\mathcal{A} \pi(a\mid s)^\alpha\pi_\beta(a\mid s)^{1-\alpha}, \mathrm{d}a, \tag{24}$$

and $D_\alpha^{\mathrm{R}} \to D_{\mathrm{KL}}$ as $\alpha \to 1$, and for $\alpha \in (0,1]$ it is convex in the second argument and obeys $D_\alpha^{\mathrm{R}}\left(\pi\middle\|\sum_i w_i\pi_{\beta_i}\right) \leq \sum_i w_i D_\alpha^{\mathrm{R}}(\pi\|\pi_{\beta_i}), \quad \alpha \in (0,1]$, while for $\alpha > 1$ this inequality is not generally guaranteed.

**$\chi$-div.** The $\chi$–divergence family generated by a convex $\chi$ with $\chi(1) = 0$ is

$$D_\chi(\pi\|\pi_\beta) = \int \chi\left(\frac{\pi(a|s)}{\pi_\beta(a|s)}\right)\pi_\beta(a|s), \mathrm{d}x, \tag{25}$$

with the Pearson chi–square special case

$$D_{\chi^2}(\pi\|\pi_\beta) = \mathbb{E}_{x\sim Q}\left[\left(\frac{\pi(a|s)}{\pi_\beta(a|s)} - 1\right)^2\right], \tag{26}$$

as $f$–divergences they are convex in the second argument and satisfy $D_\chi\left(\pi\middle\|\sum_i w_i\pi_{\beta_i}\right) \leq \sum_i w_i D_\chi(\pi\|\pi_{\beta_i})$.

**JSD.** The Jensen–Shannon divergence is

$$\mathrm{JSD}(\pi\|\pi_\beta) = \tfrac{1}{2}D_{\mathrm{KL}}(\pi) + \tfrac{1}{2}D_{\mathrm{KL}}(\pi_\beta\|\pi'), \ \ \pi' = \tfrac{1}{2}(\pi + \pi_\beta), \tag{27}$$

symmetric and bounded in $[0, \log 2]$, and being an $f$–divergence it is convex in each argument and thus $\mathrm{JSD}\left(\pi\middle\|\sum_i w_i\pi_{\beta_i}\right) \leq \sum_i w_i\mathrm{JSD}(\pi\|\pi_{\beta_i})$.

**W1.** The 1–Wasserstein distance admits the Kantorovich–Rubinstein dual

$$W_1(\pi, \pi_\beta) = \sup_{|f|_{\mathrm{Lip}}\leq 1}\left(\mathbb{E}_{x\sim P}[f(x)] - \mathbb{E}_{y\sim Q}[f(y)]\right), \tag{28}$$

which is a supremum of affine functionals and hence convex in each argument, and therefore it obeys $W_1\left(\pi, \sum_i w_i\pi_{\beta_i}\right) \leq \sum_i w_i W_1(\pi, \pi_{\beta_i})$.

**MSE.** A practical surrogate for discrete actions is the mean squared error between probability vectors

$$\mathrm{MSE}(\pi, \pi_\beta) = \frac{1}{|\mathcal{A}|}\sum_{a\in\mathcal{A}}\big(\pi(a\mid s) - \pi_\beta(a\mid s)\big)^2, \tag{29}$$

or parameter–space MSE for Gaussian policies in continuous actions, and since $\pi_\beta \mapsto \|\pi - \pi_\beta\|_2^2$ is convex we have $\mathrm{MSE}\left(\pi, \sum_i w_i\pi_{\beta_i}\right) \leq \sum_i w_i\mathrm{MSE}(\pi, \pi_{\beta_i})$.

**MMD.** With a positive–definite kernel $k$ and feature map $\phi$ in the associated RKHS, the maximum mean discrepancy is

$$\mathrm{MMD}^2(\pi, \pi_\beta) = \big|\mu_\pi - \mu_{\pi_\beta}\big|_{\mathcal{H}}^2, \qquad \mu_\pi = \mathbb{E}_{x \sim \pi}[\phi(x)], \tag{30}$$

and because the kernel mean embedding is linear while the squared norm is convex, it satisfies $\mathrm{MMD}^2\left(\pi, \sum_i w_i \pi_{\beta_i}\right) \leq \sum_i w_i \mathrm{MMD}^2(\pi, \pi_{\beta_i})$.

**Corollary 1.** *The mixture upper bound is valid for KL, all X–divergences including JSD, $W_1$, MSE and MMD, and for Rényi only when $\alpha \in (0, 1]$, and it is not generally valid for Rényi with $\alpha > 1$.*

## B   APPENDIX: PROOF DETAILS FOR THEOREM 1

**Assumption 1** (Local Smoothness). *Let $\mathcal{D}$ denote the dataset support. There exist constants $L, \rho > 0$ such that for any $x \in \mathcal{D}$ and perturbation $k$ with $|k| \leq \rho$, the critic $Q_\psi$ is differentiable (almost everywhere) and has an L-Lipschitz gradient. Under this assumption, Taylor's theorem gives*

$$\big|Q_\psi(x + k\mathbf{w}) - Q_\psi(x) - k\langle \mathbf{w}, \nabla_x Q_\psi(x)\rangle\big| \leq \tfrac{L}{2}k^2.$$

Thus, the expansion in Eq. (2) is a standard first-order approximation with a controlled $\mathcal{O}(k^2)$ remainder. [1] Similar local differentiability assumptions are common in recent theoretical work on standard offline RL (Yin et al., 2022; Li et al., 2023a; Qiao & Yue, 2026).

*Proof of Theorem 1.* Fix a dataset pair $(x, x')$. All expectations, variances, and covariances in this proof are taken with respect to the perturbation proposal $u = (k, k', \mathbf{w}, \mathbf{w}')$, while $(x, x')$ is held fixed. The perturbed error is

$$\delta = r + \gamma Q_{\phi'}\big(x' + k'\mathbf{w}'\big) - Q_\phi\big(x + k\mathbf{w}\big). \tag{31}$$

Apply the first–order expansion in feature space and discard $\mathcal{O}(k^2 + k'^2)$ terms:

$$Q_{\phi'}(x' + k_2'\mathbf{w}_2') \approx Q_{\phi'}(x') + k_2'\langle \mathbf{w}_2', \nabla_{x'}Q_{\phi'}(x')\rangle, \tag{32}$$

$$Q_\phi(x + k\mathbf{w}) \approx Q_\phi(x) + k\langle \mathbf{w}, \nabla_x Q_\phi(x)\rangle. \tag{33}$$

Suppose $k_1'\mathbf{w_1'}$ is OOD action (same as (An et al., 2021)) and $(\mathbf{s}', \mathbf{a}') \sim \mathcal{D}$, we plugging $(\mathbf{s}', \pi(\mathbf{s}')) = (\mathbf{s}', \mathbf{a}') + k_1'\mathbf{w_1'}$ into above:

$$Q_{\phi'}((\mathbf{s}', \pi(\mathbf{s}')) + k_1'\mathbf{w_1'}) = Q_{\phi'}((\mathbf{s}', \mathbf{a}') + k_1'\mathbf{w_1'} + k_2'\mathbf{w}_2') \approx Q_{\phi'}(x') + k'\langle \mathbf{w}', \nabla_{x'}Q_{\phi'}(x')\rangle$$

where $k_1'\mathbf{w}_1' + k_2'\mathbf{w}_2' = k'\mathbf{w}'$ and $x' = (\mathbf{s}', \mathbf{a}')$ denotes the next state action pairs of $x = (\mathbf{s}, \mathbf{a})$ come from the dataset. Thus, we have

$$\delta \approx \underbrace{r + \gamma Q_{\phi'}(x') - Q_\phi(x)}_{\delta_{\mathrm{base}}} + \underbrace{\gamma k'\langle \mathbf{w}', \nabla_x Q_{\phi'}(x')\rangle - k\langle \mathbf{w}, \nabla_x Q_\phi(x)\rangle}_{\delta_{\mathrm{lin}}(u)}. \tag{34}$$

To show this, we suppose the Q-value predictions for the in-distribution state-action pairs coincide, *i.e.* $Q_\psi(x)$, $\psi \sim \{\phi, \phi'\}$, which is common used in RL (An et al., 2021). That is, the base term $\delta_{\mathrm{base}}$ is a constant. Therefore $\mathrm{Var}(\delta_{\mathrm{base}}) = 0$ and $\mathrm{Cov}(\delta_{\mathrm{base}}, \delta_{\mathrm{lin}}) = 0$, so

$$\mathrm{Var}[\delta] \approx \mathrm{Var}(\delta_{\mathrm{lin}}(u)). \tag{35}$$

Expanding the variance and using bilinearity of covariance yields

$$\mathrm{Var}[\delta] = \gamma^2 \, \mathrm{Var}\Big(k'\langle \mathbf{w}', \nabla_x Q_{\phi'}(x')\rangle\Big) + \mathrm{Var}\Big(k\langle \mathbf{w}, \nabla_x Q_\phi(x)\rangle\Big)$$
$$- 2\gamma \, \mathrm{Cov}\Big(k'\langle \mathbf{w}', \nabla_x Q_{\phi'}(x')\rangle, \; k\langle \mathbf{w}, \nabla_x Q_\phi(x)\rangle\Big). \tag{36}$$

When $k, k'$ are treated as fixed scalars inside the proposal, Eq. (36) reduces exactly to the three terms in the main text's Eq. (6), with moments taken over $(\mathbf{w}, \mathbf{w}')$. By the identity $\mathbb{E}[\delta^2] = (\mathbb{E}[\delta])^2 + \mathrm{Var}[\delta]$, the corresponding second–moment statement follows. □

---

[1] While deep ReLU networks are not globally smooth, they are locally Lipschitz almost everywhere (Hein & Andriushchenko, 2017). Our theoretical model captures the dominant first-order effects of cross-covariance within this local trust region, which aligns with our empirical observations of instability.

**Remark 1** (on the base term and the vanishing covariance). *In the per–sample proof above the randomness is only over the perturbation $u = (k, k', \mathbf{w}, \mathbf{w}')$ while $(x, x')$ is fixed, so the base term*

$$\delta_{\text{base}} = r + \gamma\, Q_{\phi'}(x') - Q_{\phi}(x)$$

*is a constant and therefore $\mathrm{Var}(\delta_{\text{base}}) = 0$ and $\mathrm{Cov}(\delta_{\text{base}}, \delta_{\text{lin}}) = 0$ exactly. When viewing Eq. (4) at the dataset level (averaging over $(x, x') \sim \mathcal{D}$), within the short locality window used in the main text on–support predictions are already accurate and conditional reward noise is negligible; hence $\delta_{\text{base}}$ is approximately constant over $\mathcal{D}$. In that case the $k$–dependent covariance term vanishes to first order, and all displacement dependence arises from $\mathrm{Var}(\delta_{\text{lin}})$.*

**Proposition 1** (block form under perturbation randomness). *Let*

$$S_2 = \mathrm{Cov}(k'\mathbf{w}'), \qquad S_1 = \mathrm{Cov}(k\mathbf{w}), \qquad N = \mathrm{Cov}(k'\mathbf{w}',\ k\mathbf{w}), \tag{37}$$

*where all covariances are with respect to the perturbation proposal. Then the variance contribution in Eq. (36) admits the quadratic form*

$$\mathrm{Var}[\delta] = \gamma^2 \langle \nabla_{x'} Q_{\phi'}(x'),\ S_2\, \nabla_{x'} Q_{\phi'}(x') \rangle + \langle \nabla_x Q_{\phi}(x),\ S_1\, \nabla_x Q_{\phi}(x) \rangle$$
$$- 2\gamma \langle \nabla_{x'} Q_{\phi'}(x'),\ N\, \nabla_x Q_{\phi}(x) \rangle. \tag{38}$$

**Corollary 2** (equal–direction simplification). *If $k = k' = k$ and $\mathbf{w} = \mathbf{w}' = \mathbf{w}$, and $\Omega = \mathrm{Cov}(\mathbf{w})$, then*

$$\mathrm{Var}[\delta] = k^2 \langle \gamma\nabla_{x'} Q_{\phi'}(x') - \nabla_x Q_{\phi}(x),\ \Omega\left(\gamma\nabla_{x'} Q_{\phi'}(x') - \nabla_x Q_{\phi}(x)\right) \rangle. \tag{39}$$

*In particular, if $\mathbf{w}$ is isotropic on $\mathbb{S}^{m-1}$ so that $\Omega = \frac{1}{m}I$, then*

$$\mathrm{Var}[\delta] = \frac{k^2}{m} \left\| \gamma\nabla_{x'} Q_{\phi'}(x') - \nabla_x Q_{\phi}(x) \right\|_2^2. \tag{40}$$

## C    APPENDIX: PROOFS FOR SECTION 5

*Proof of Theorem 2.* Let $C = \mathrm{Cov}(g', g)$. With cluster label $Z \in \{1, \dots, K\}$,

$$\mathrm{Cov}(g', g) = \mathbb{E}\left[(g' - \mathbb{E}g')(g - \mathbb{E}g)^\top\right] = \mathbb{E}\left[\ \mathbb{E}\left[(g' - \mathbb{E}g')(g - \mathbb{E}g)^\top \mid Z\right]\ \right]$$

$$= \mathbb{E}\left[\ \mathbb{E}\left[(g' - \mu'_Z + \mu'_Z - \mathbb{E}g')(g - \mu_Z + \mu_Z - \mathbb{E}g)^\top \mid Z\right]\ \right]$$

$$= \mathbb{E}[\ \mathrm{Cov}(g', g \mid Z)\ ] + \mathrm{Cov}(\mu'_Z, \mu_Z), \tag{41}$$

which is Eq. (9). If a minibatch is drawn from a fixed $z$, then the effective covariance is $C_z$. For any unit $\mathbf{w}', \mathbf{w}$, $|\mathbf{w}'^\top C_z \mathbf{w}| \le \|C_z\|_2$ by the definition of the operator norm. Moreover, $\|C_z\|_2 \le \|\Sigma'_z\|_2^{1/2}\|\Sigma_z\|_2^{1/2}$ by the operator Cauchy-Schwarz inequality, and $\|\Sigma\|_2 \le \mathrm{tr}(\Sigma)$ for PSD $\Sigma$, yielding Eq. (10). $\qquad\square$

**Lemma 1** (Alignment inside a cluster controls sign and magnitude). *Let the SVD of $C_z$ be $C_z = U\Sigma V^\top$ with singular values $\sigma_1 \ge \sigma_2 \ge \cdots \ge 0$. For unit $\mathbf{w}', \mathbf{w}$,*

$$\mathbf{w}'^\top C_z \mathbf{w} \ge \sigma_1 \cos\theta' \cos\theta - \sigma_2 \sin\theta' \sin\theta, \tag{42}$$
$$\theta' = \angle(\mathbf{w}', u_1), \qquad \theta = \angle(\mathbf{w}, v_1).$$

*Choosing $\mathbf{w}' = u_1$ and $\mathbf{w} = v_1$ yields $\mathbf{w}'^\top C_z \mathbf{w} = \sigma_1 \ge 0$. The cross term in Eq. (8) is then nonpositive. Sketch of proof. Expand in the singular bases and bound the residual with $\sigma_2$ via Cauchy-Schwarz.*

*Proof.* Let $C_z = U\Sigma V^\top$ with $\Sigma = \mathrm{diag}(\sigma_1, \sigma_2, \dots)$. For unit $\mathbf{w}', \mathbf{w}$,

$$\mathbf{w}'^\top C_z \mathbf{w} = (U^\top \mathbf{w}')^\top \Sigma (V^\top \mathbf{w}) = \sum_j \sigma_j (u_j^\top \mathbf{w}')(v_j^\top \mathbf{w}). \tag{43}$$

Let $\cos\theta' = u_1^\top \mathbf{w}'$ and $\cos\theta = v_1^\top \mathbf{w}$. By Cauchy-Schwarz and $\sum_{j\ge2}(u_j^\top \mathbf{w}')^2 = \sin^2\theta'$, $\sum_{j\ge2}(v_j^\top \mathbf{w})^2 = \sin^2\theta$, we obtain

$$\sum_{j\ge2} \sigma_j (u_j^\top \mathbf{w}')(v_j^\top \mathbf{w}) \ge -\sigma_2 \sqrt{\sum_{j\ge2}(u_j^\top \mathbf{w}')^2} \sqrt{\sum_{j\ge2}(v_j^\top \mathbf{w})^2} = -\sigma_2 \sin\theta' \sin\theta. \tag{44}$$

Combining Eq. (43) and Eq. (44) gives Eq. (42). Choosing $\mathbf{w}' = u_1$, $\mathbf{w} = v_1$ yields $\mathbf{w}'^\top C_z \mathbf{w} = \sigma_1 \ge 0$. Thus, the proof is completed. $\qquad\square$

## D    MIXTURE OF POLICIES

**Definition 1** (f divergence and special cases). *Let $f : (0, \infty) \to \mathbb{R}$ be a proper lower semicontinuous convex function with $f(1) = 0$. Let $\pi$ and $\nu$ be probability measures on the action space that admit densities with respect to a common reference measure. Assume $\pi \ll \nu$ so that $\frac{\pi}{\nu}$ is well defined $\nu$ almost everywhere. The $f$ divergence is*

$$D_f(\pi\|\nu) = \int \nu(a) \, f\left(\frac{\pi(a)}{\nu(a)}\right) \, da.$$

*It satisfies $D_f(\pi\|\nu) \geq 0$ and $D_f(\pi\|\nu) = 0$ if and only if $\pi = \nu$ almost everywhere whenever $f$ is strictly convex at $1$. Special cases: are*

$$\chi^2(\pi\|\nu) = D_{(t-1)^2}(\pi\|\nu) = \int \frac{\pi(a)^2}{\nu(a)} \, da - 1$$

*and*

$$\mathrm{KL}(\pi\|\nu) = D_{t \log t}(\pi\|\nu) = \int \pi(a) \log \frac{\pi(a)}{\nu(a)} \, da.$$

**Lemma 2** (Global operator freeze and lower bound). *For any nonnegative measurable function $h$ and any policy $\pi$,*

$$(I - \gamma P^\pi)^{-1}h = \sum_{t=0}^{\infty} \gamma^t (P^\pi)^t h \leq \frac{1}{1-\gamma} \|h\|_\infty.$$

*Let $\bar{\rho} := \frac{1}{1-\gamma}$. For the transformed CQL improvement target at a state $s$*

$$V^\pi(s) - \alpha\big((I - \gamma P^\pi)^{-1}\chi^2(\pi\|\pi_\beta)\big)(s)$$

*one has the global statewise lower bound*

$$V^\pi(s) - \alpha\big((I - \gamma P^\pi)^{-1}\chi^2(\pi\|\pi_\beta)\big)(s) \; \geq \; \mathbb{E}_\pi[Q(s,a)] - \alpha\,\bar{\rho}\,\sup_{s'}\chi^2(\pi\|\pi_\beta)(s').$$

*The same bound holds with $\mathrm{KL}$ or any nonnegative combination of $\chi^2$ and $\mathrm{KL}$.*

*Proof.* Monotonicity and the geometric series bound yield

$$(I - \gamma P^\pi)^{-1}h \leq \sum_{t=0}^{\infty} \gamma^t \|h\|_\infty = \frac{1}{1-\gamma}\|h\|_\infty.$$

Set $h(\cdot) = \chi^2(\pi\|\pi_\beta)(\cdot)$, then linearize $V^\pi(s)$ to $\mathbb{E}_\pi[Q(s,a)]$ at the working point. Both penalties are nonnegative, hence the inequality is preserved. $\qquad\square$

**Lemma 3** (Convexity in the second argument and clusterwise training). *Let $f$ satisfy the definition above. Let $\nu = \sum_{m=1}^{M} w_m \nu_m$ with $w_m \geq 0$ and $\sum_m w_m = 1$. Assume $\pi \ll \nu_m$ for every $m$ with $w_m > 0$. Then*

$$D_f(\pi\|\nu) \leq \sum_{m=1}^{M} w_m D_f(\pi\|\nu_m).$$

*Consequently the surrogate*

$$\mathcal{J}(\pi) = \mathbb{E}_\pi[Q] - \bar{\rho}\left(\alpha \sum_m w_m \chi^2(\pi\|\nu_m) + \beta \sum_m w_m \mathrm{KL}(\pi\|\nu_m)\right)$$

*is a computable lower bound to the same expression with the mixture $\nu$ placed inside each divergence. Sampling a cluster index $M \sim w$ and updating with the per cluster objective*

$$\mathbb{E}_\pi[Q] - \bar{\rho}\big(\alpha\chi^2(\pi\|\nu_M) + \beta\mathrm{KL}(\pi\|\nu_M)\big)$$

*gives an unbiased estimator of $\nabla\mathcal{J}(\pi)$.*

*Special cases: For Pearson and for Kullback Leibler the same inequality holds, therefore the same lower bound and the same stochastic cluster training rule apply when $\nu_m$ are Gaussian components of a mixture.*

*Proof.* Define the perspective $g(u, v) = v f(u/v)$ for $u > 0$ and $v > 0$. Since $f$ is convex, $g$ is jointly convex. For each $a$,

$$g\Big(\pi(a), \sum_m w_m \nu_m(a)\Big) \leq \sum_m w_m g\big(\pi(a), \nu_m(a)\big).$$

Integrate over $a$ to obtain

$$D_f(\pi \| \nu) \leq \sum_m w_m D_f(\pi \| \nu_m).$$

Taking expectation with respect to $M \sim w$ proves that the per cluster gradient is an unbiased estimator of the gradient of the weighted sum objective. □

**Proposition 2** (General isotropic penalty and update direction). *Fix a state $s$ and let the policy class be $\pi_\mu$ with a mean parameter $\mu$. Let $\Sigma$ be positive definite and assume the penalty has the isotropic form*

$$\Psi(\mu - \mu_\beta) = \psi\big((\mu - \mu_\beta)^\top \Sigma^{-1} (\mu - \mu_\beta)\big)$$

*with $\psi$ differentiable and strictly increasing on $[0, \infty)$. Let $Q(s, a)$ be linearized at $a = \mu_\beta$ with gradient $g(s)$. Consider*

$$\max_\mu \quad \lambda := \bar{\rho} c, \ c \geq 0.$$

*Any maximizer satisfies*

$$\mu^\star(s) = \mu_\beta + \kappa^\star(s) \, \Sigma \, g(s)$$

*where $\kappa^\star(s) > 0$ is the unique solution of*

$$2\lambda \, \psi'\big(\kappa^2 R\big) \kappa = 1, \qquad R := g(s)^\top \Sigma g(s).$$

*Special cases: If $\psi(r) = e^r - 1$ which corresponds to Pearson under equal covariances then*

$$\kappa^\star(s) = \sqrt{\frac{1}{2R} \, W\Big(\frac{R}{2\lambda^2}\Big)}$$

*where $W$ is the Lambert function given by $W(z)e^{W(z)} = z$. The argument $R/(2\lambda^2)$ is positive when $R > 0$ and $\lambda > 0$, therefore the principal branch gives a positive $\kappa^\star$. If $\psi(r) = \frac{r}{2}$ which corresponds to Kullback Leibler under equal covariances then*

$$\kappa^\star(s) = \frac{1}{\lambda}.$$

*For a Gaussian mixture behavior one applies the convexity lemma to replace the mixture penalty by the weighted sum of per cluster penalties and solves the same scalar equation per cluster with the corresponding $\Sigma$ and $\mu_\beta$.*

*Proof.* Let $\Delta := \mu - \mu_\beta$ and define

$$F(\Delta) = \Delta^\top g - \lambda \, \psi\big(\Delta^\top \Sigma^{-1} \Delta\big).$$

The first order stationarity condition is

$$\nabla_\Delta F(\Delta) = g - 2\lambda \, \psi'\big(\Delta^\top \Sigma^{-1} \Delta\big) \Sigma^{-1} \Delta = 0.$$

Therefore $\Delta$ is colinear with $\Sigma g$. Let $\Delta = \kappa \Sigma g$ and $R = g^\top \Sigma g$. Then

$$2\lambda \, \psi'(\kappa^2 R) \, \kappa = 1.$$

Strict increase of $\psi'$ implies a unique positive root. Substituting $\psi(r) = e^r - 1$ gives the Lambert equation

$$\kappa \, e^{\kappa^2 R} = \frac{1}{2\lambda}$$

which reduces to the stated form by $y = \kappa^2 R$ and $ye^{2y} = R/(2\lambda^2)$. Substituting $\psi(r) = r/2$ gives $\kappa = 1/\lambda$. □

**Proposition 3** (Adding KL gives stabilization and quantitative bounds in Gaussian cases). *Consider for a fixed state $s$ the surrogate*

$$\mathbb{E}_\pi[Q(s,a)] - \bar{\rho}\Big(\alpha\,\chi^2(\pi\|\pi_\beta) + \beta\,\mathrm{KL}(\pi\|\pi_\beta)\Big), \qquad \alpha \geq 0,\ \beta > 0.$$

*Assume the policy class admits a local quadratic lower bound on KL around a reference parameter $\theta_\beta$*

$$\mathrm{KL}(\pi_\theta\|\pi_{\theta_\beta}) \geq \frac{m}{2}\,\|\theta - \theta_\beta\|^2$$

*for some constant $m > 0$. Then near $\theta_\beta$ the surrogate is $m\beta\bar{\rho}$ strongly concave in $\theta$ and the unique maximizer satisfies*

$$\|\theta^\star - \theta_\beta\| \leq \frac{\big\|\nabla_\theta\mathbb{E}_\pi[Q]\big|_{\theta_\beta}\big\|}{m\beta\bar{\rho}}.$$

*Special cases: If $\pi_\mu$ and $\pi_\beta$ are Gaussians with equal covariance $\Sigma_\beta$ and parameter $\theta = \mu$, then the maximizer has the form*

$$\mu^\star = \mu_\beta + \kappa^\star\Sigma_\beta g, \qquad (2\alpha\bar{\rho}\,e^{\kappa^{\star 2}R} + \beta\bar{\rho})\,\kappa^\star = 1, \qquad R = g^\top\Sigma_\beta g.$$

*This implies the step size cap*

$$\kappa^\star \leq \frac{1}{\beta\bar{\rho}} = \frac{1-\gamma}{\beta}$$

*and if $\alpha > 0$ one also has*

$$\chi^2\big(\pi_{\mu^\star}\|\pi_\beta\big) = e^{\kappa^{\star 2}R} - 1 \leq \frac{\beta}{2\alpha} - 1.$$

*For a Gaussian mixture behavior policy the convexity lemma yields*

$$\mathrm{KL}\bigg(\pi\,\Big\|\,\sum_m w_m\nu_m\bigg) \leq \sum_m w_m\mathrm{KL}(\pi\|\nu_m), \qquad \chi^2\bigg(\pi\,\Big\|\,\sum_m w_m\nu_m\bigg) \leq \sum_m w_m\chi^2(\pi\|\nu_m),$$

*So the same lower bound structure and per-cluster step control apply.*

*Proof.* The local quadratic lower bound on KL implies that subtracting $\beta\bar{\rho}\mathrm{KL}$ adds curvature at least $m\beta\bar{\rho}$, which gives strong concavity and uniqueness near $\theta_\beta$. The parameter step bound follows from the basic inequality for the maximization of a strongly concave function. In the Gaussian mean case

$$\mathrm{KL}(\mathcal{N}(\mu,\Sigma_\beta)\|\mathcal{N}(\mu_\beta,\Sigma_\beta)) = \frac{1}{2}(\mu - \mu_\beta)^\top\Sigma_\beta^{-1}(\mu - \mu_\beta)$$

and the stationarity condition becomes

$$g - 2\alpha\bar{\rho}e^{\Delta^\top\Sigma_\beta^{-1}\Delta}\Sigma_\beta^{-1}\Delta - \beta\bar{\rho}\Sigma_\beta^{-1}\Delta = 0.$$

With $\Delta = \kappa\Sigma_\beta g$ this reduces to the scalar equation in $\kappa$ and yields the stated bounds. For mixtures, apply convexity in the second argument to both divergences and optimize per cluster. $\square$

**Remark 2** (Effect under poor coverage and relation to mixture training). *If a measurable set $A$ satisfies $\int_A \pi(a \mid s)\,da = \delta > 0$ while $\varepsilon_A = \int_A \pi_\beta(a \mid s)\,da$ is small, then*

$$\chi^2(\pi\|\pi_\beta) \geq \frac{\delta^2}{\varepsilon_A} - 1.$$

*This shows the sensitivity of a pure Pearson penalty when coverage is poor. Adding the KL term enforces curvature and gives the cap*

$$\kappa^\star \leq \frac{1-\gamma}{\beta}, \qquad \chi^2(\pi_{\mu^\star}\|\pi_\beta) \leq \frac{\beta}{2\alpha} - 1,$$

*which stabilizes the update in low density regions. When the behavior policy is a Gaussian mixture, convexity in the second argument of both divergences yields additive upper bounds that justify training by data clusters with random cluster selection while increasing a computable lower bound to the true mixture objective.*

# E  EXPERIMENT DETAILS

## E.1  EXTENDED IMPLEMENTATION

Our method is designed to be plug-and-play and can be readily incorporated into numerous existing offline RL frameworks. In the experiments, we instantiate it on top of both CQL and TD3+BC. Unless explicitly stated, CQL serves as the base algorithm, and our method is indicated with the notation *"Ours"*. We implement all experiments in PyTorch 2.1.2 on Ubuntu 20.04.4 LTS with four NVIDIA GeForce RTX 3090 GPUs. The actor and critic are ReLU multilayer perceptrons with four hidden layers of width 256. We train with Adam using batch size 256, learning rate $1 \times 10^{-4}$ for the actor, learning rate $3 \times 10^{-4}$ for the critic, and discount factor $\gamma = 0.99$. Unless stated otherwise, we use default hyperparameters $\alpha$ following CQL (Kumar et al., 2020a). In addition, we use the network's penultimate layer (the learned features) as a computationally tractable surrogate for $\nabla_x Q_\psi(x)$. [2] [3] We follow the D4RL protocol for normalized scores. Given a task score and the corresponding random and expert scores (as reported in Table 15), the normalized score is defined as

$$\text{normalized score} = 100 \times \frac{\text{score} - \text{random score}}{\text{expert score} - \text{random score}} .$$

Unless noted, we report the mean normalized score over 3 to 5 repetitions per setting.

## E.2  BENCHMARK

We evaluate our method on the widely recognized offline RL benchmark D4RL (Fu et al., 2021; Todorov et al., 2012), which encompasses several domains, including Locomotion, Maze2D, AntMaze, Adroit, and Kitchen as illustrated in Figure 7. The locomotion domain includes four continuous robotic control tasks ("HalfCheetah", "Hopper", "Walker2d", and "Ant"), each providing four dataset quality levels (expert, medium-expert, medium, and medium-replay). The Maze2D domain requires a 2D agent to navigate to fixed goal positions across three maze sizes ("umaze", "medium", and "large"), featuring both dense and sparse reward variants. In addition, we consider the Adroit suite of high-dimensional robotic manipulation tasks ("pen", "door", "relocate", and "hammer"), using both "human" and "cloned" datasets. We further include the AntMaze benchmark for long-horizon navigation with sparse rewards, covering eight maze configurations ("umaze", "umaze-diverse", "medium-play", "medium-diverse", "large-play", "large-diverse", "ultra-play", and "ultra-diverse"). Finally, we evaluate on the Franka Kitchen environment, a 9-DoF multi-task manipulation benchmark, using the "mixed", "partial", and "complete" settings, which require the agent to accomplish multiple goal-conditioned sub-tasks within a single episode.

Subsequently, we evaluate $C^4$ by addressing the following key questions:

1. **Evaluation on more benchmarks with reduced data:** How does $C^4$ compare against recent state-of-the-art offline RL, particularly within the challenging 10k-sample setting?

2. **Generalization:** How does our method perform across a broader suite of datasets, especially in extremely small data regimes?

3. **Scalability and consistency:** Is there a performance trade-off on large datasets, and does our method consistently outperform baselines across the entire spectrum of data regimes?

4. **Covariance control under $C^4$:** Under the $C^4$ intervention, can the covariance be effectively maintained at a low level?

5. **Robustness and efficiency:** How sensitive is the method to initialization? What do ablation studies reveal, and does $C^4$ introduce additional computational overhead?

## E.3  EVALUATION ON MORE BENCHMARKS WITH REDUCED DATA

To rigorously evaluate performance in the small-data regime, the primary motivation for our approach, Table 2 reports normalized D4RL scores across all tasks containing at most **10k** transitions. This

---

[2]To avoid expensive double backpropagation, we use the feature layer as a proxy. This controls input sensitivity via the chain rule, akin to layer-wise Lipschitz constraints (Miyato et al., 2018; Gouk et al., 2021).

[3]We treat this as a practical engineering approximation. Our code provides both versions, and we recommend the exact $\nabla_x Q$-based implementation for theoretical rigor.

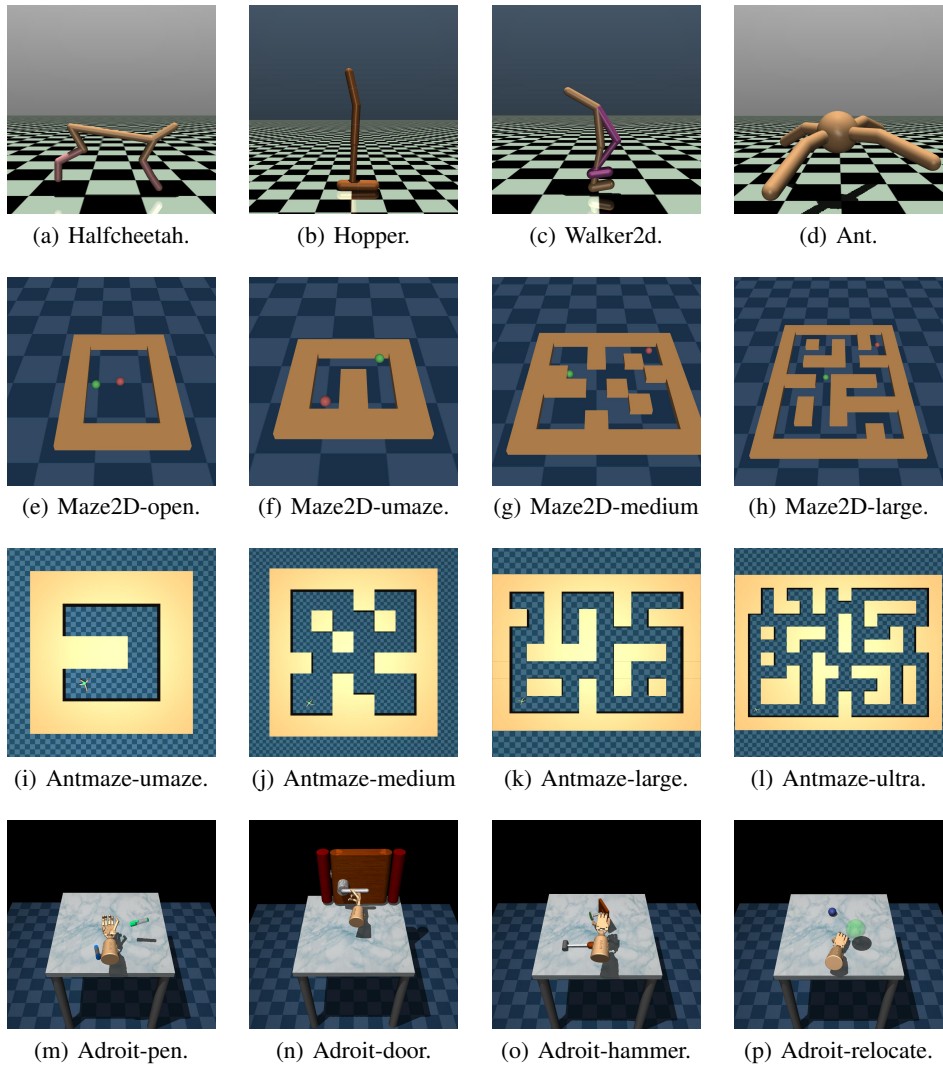

(a) Halfcheetah.  (b) Hopper.  (c) Walker2d.  (d) Ant.

(e) Maze2D-open.  (f) Maze2D-umaze.  (g) Maze2D-medium  (h) Maze2D-large.

(i) Antmaze-umaze.  (j) Antmaze-medium  (k) Antmaze-large.  (l) Antmaze-ultra.

(m) Adroit-pen.  (n) Adroit-door.  (o) Adroit-hammer.  (p) Adroit-relocate.

Figure 7: Locomotion, Maze2D, AntMaze and Adroit tasks.

selection encompasses the sample-reduced locomotion tasks (Hopper, HalfCheetah, Walker2d) as well as naturally sparse datasets: Kitchen-complete, Pen-human, and Door-human. Our method demonstrates superior generalization, achieving the highest average score of **58.1** and consistently outperforming strong baselines like BPPO and TSRL in these strictly data-constrained environments.

Furthermore, we evaluated additional D4RL datasets beyond locomotion, including Adroit, AntMaze, and Maze2D, with results reported in Fig. 8 and Tables 3, 4, and 5. The results show that our method delivers consistently strong performance across these diverse benchmarks and achieves remarkable gains in the vast majority of small data regimes.

### E.4 GENERALIZATION AND COMPATIBILITY WITH SOTA OFFLINE RL METHODS

**Improvements on standard baselines.** We first examine the generalization capability of $C^4$ as a plug-in module for established baselines. Table 6 compares CQL with and without the $C^4$ module on the full D4RL locomotion datasets. Rather than degrading performance, $C^4$ increases the total normalized score from 695.5 to 742.1. In tasks where the base algorithm already performs strongly (e.g., Hopper-medium-expert), $C^4$ maintains competitive performance, while in more challenging settings (e.g., Hopper-medium, Walker2d-medium-replay) it yields notable gains. These observations suggest that as data density increases, the gradient distribution becomes more stable and the $C^4$

Table 2: Normalized D4RL scores on tasks with ≤10k samples per dataset. We compare performance on reduced-sample locomotion tasks and inherently small datasets (Kitchen, Pen, Door).

| Task (≤10k) | BC | TD3BC | CQL | IQL | DOGE | BPPO | TSRL | A2PR | Ours |
|---|---|---|---|---|---|---|---|---|---|
| Hopper-m | 28.8 | 30.7 | 50.1 | 61.0 | 55.6 | 55.0 | 60.9 | 55.9 | **69.2** |
| Hopper-mr | 19.7 | 11.3 | 13.2 | 16.2 | 19.1 | 45.1 | 23.5 | 12.5 | **45.9** |
| Hopper-me | 38.2 | 22.6 | 43.2 | 51.7 | 36.8 | 27.9 | 56.6 | 49.7 | **81.3** |
| Halfcheetah-m | 40.2 | 25.9 | 41.7 | 35.6 | 42.8 | 28.5 | 43.3 | 37.1 | **46.3** |
| Halfcheetah-mr | 25.2 | 29.1 | 16.3 | 34.1 | 26.3 | 34.4 | 27.7 | 23.6 | **43.1** |
| Halfcheetah-me | 33.7 | 23.5 | 39.7 | 14.3 | 33.1 | 22.3 | 37.2 | 32.4 | **46.9** |
| Walker2d-m | 25.4 | 11.2 | 54.1 | 34.2 | 53.7 | 54.7 | 47.3 | 5.9 | **65.9** |
| Walker2d-mr | 2.5 | 9.3 | 13.8 | 17.7 | 15.5 | 29.5 | 27.6 | 34.4 | **55.4** |
| Walker2d-me | 35.1 | 12.4 | 26.0 | 38.0 | 42.5 | 61.3 | 50.9 | 56.5 | **96.3** |
| Kitchen-c | 33.8 | 0.0 | 31.3 | 51.0 | 10.2 | **91.5** | 5.7 | 8.3 | 55.5 |
| Pen-h | 9.5 | 9.5 | 41.2 | 71.5 | 35.8 | **117.8** | 85.7 | -0.1 | 85.3 |
| Door-h | 0.6 | 0.6 | 10.7 | 4.3 | -1.1 | **25.9** | 0.3 | -0.3 | 5.8 |
| Average | 24.4 | 15.5 | 31.8 | 35.8 | 30.9 | 49.5 | 38.9 | 26.3 | **58.1** |

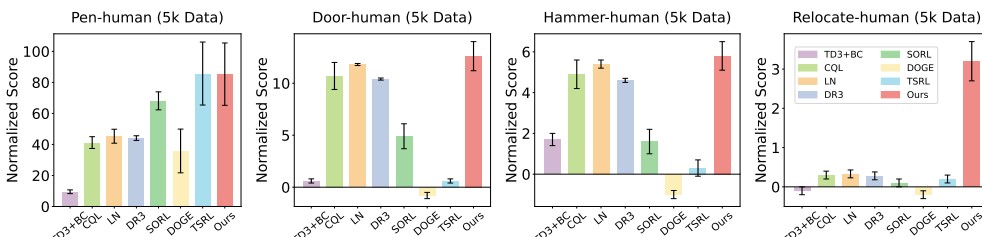

Figure 8: Comparison on Adroit tasks over different data sizes.

penalty naturally adjusts, avoiding over-regularization and retaining the strengths of the underlying algorithm.

**Compatibility with stronger offline RL methods.** Beyond standard baselines, we investigate whether $C^4$ can further enhance recent state-of-the-art (SOTA) algorithms, specifically BPPO (Zhuang et al., 2023) and A2PR (Liu et al., 2024). While these methods achieve strong performance on standard benchmarks, they remain susceptible to overfitting in strictly limited data regimes (e.g., the 10k-sample setting). Theoretically, both algorithms rely on temporal-difference (TD) learning for value estimation: A2PR utilizes Q-learning updates, while BPPO employs a SARSA-style value warm-up. Consequently, they inherit the vulnerability to high cross-covariance terms in the TD error when data is scarce. We postulate that $C^4$ serves as a complementary regularizer to these approaches by explicitly controlling the covariance induced by TD updates. To test this, we integrate $C^4$ into BPPO and A2PR without altering their core mechanisms.

The results, summarized in Tables 7 and 8, indicate substantial performance gains. For **BPPO** (Table 7), adding $C^4$ consistently improves performance across diverse tasks, raising the average normalized score from 38.3 to 53.5—a relative improvement of 39.6%. For **A2PR** (Table 8), the effect is even more pronounced: the method achieves an average score of 57.6 compared to the baseline of 31.0, corresponding to an 85.9% relative gain. These findings suggest that uncontrolled cross-covariance is a fundamental bottleneck even for advanced offline RL methods, and $C^4$ provides a robust, algorithm-agnostic solution to mitigate this issue in small-data regimes.

## E.5 SCALABILITY AND CONSISTENCY

A central question for any regularization technique designed for data scarcity is whether it hinders learning when data is abundant. In this subsection, we examine whether our method imposes a performance trade-off on large datasets and how consistently it scales across varying data budgets.

**Adaptation to large-scale benchmarks.** To address the first question, we assess how our plug-in approach compares against state-of-the-art methods specifically tuned for general offline RL bench-

Table 3: Average normalized score on D4RL Adroit tasks with reduced-size datasets ($\leq$ 10k samples).

| Task | Size | TD3BC | CQL | IQL | LN | DR3 | SORL | DOGE | BPPO | TSRL | A2PR | **Ours** |
|---|---|---|---|---|---|---|---|---|---|---|---|---|
| Pen-human | 5k | 9.5 | 41.2 | 71.5 | 45.3 | 44.1 | 68.1 | 35.8 | **117.8** | 85.7 | -0.1 | 85.3 |
| Hammer-human | 5k | 1.7 | 4.9 | 4.3 | 5.4 | 4.6 | 1.6 | -1.0 | 2.4 | 0.3 | -0.3 | **5.8** |
| Door-human | 5k | 0.6 | 10.7 | 2.9 | 11.8 | 10.4 | 4.9 | -0.8 | **25.9** | 0.6 | -0.2 | 12.6 |
| Relocate-human | 5k | -0.1 | 0.3 | 1.5 | 0.33 | 0.28 | 0.1 | -0.2 | -0.1 | 0.2 | 0.1 | **3.2** |
| Pen-cloned | 10k | -0.2 | 1.8 | 35.9 | 32.5 | 2.0 | 30.1 | 33.5 | 30.4 | 38.4 | -0.1 | **58.2** |
| Hammer-cloned | 10k | 0.0 | 0.5 | 1.5 | 0.6 | 0.4 | 0.7 | -0.2 | **8.4** | 0.4 | -0.3 | 3.5 |
| Door-cloned | 10k | -0.3 | -0.6 | 2.6 | -0.2 | -0.3 | -0.6 | 0.0 | -0.1 | 0.1 | -0.3 | **2.8** |
| Relocate-cloned | 10k | -0.2 | -0.3 | **1.4** | -0.2 | -0.1 | -0.3 | 0.0 | 0.2 | -0.2 | 0.2 | **1.4** |
| Average | - | 1.4 | 7.3 | 15.2 | 12.0 | 7.7 | 13.1 | 8.4 | **23.1** | 15.7 | -0.1 | 21.6 |

Table 4: Average score on D4RL Maze2D tasks with reduced-size datasets (10k samples).

| Task | TD3BC | CQL | IQL | LN | DR3 | SORL | DOGE | BPPO | TSRL | A2PR | **Ours** |
|---|---|---|---|---|---|---|---|---|---|---|---|
| Maze2D-open | 12.3 | 52.3 | 63.6 | 22.1 | 37.0 | 58.0 | 70.1 | 75.4 | 57.0 | 35.7 | **83.0** |
| Maze2D-umaze-dense | 104.5 | 74.1 | 103.6 | 86.5 | 117.6 | 63.4 | 115.1 | 51.4 | **138.0** | 125.5 | 130.3 |
| Maze2D-umaze | 13.5 | 35.9 | 68.6 | 17.9 | 76.9 | 72.5 | 52.8 | 58.9 | 76.9 | **100.6** | 87.8 |
| Maze2D-medium-dense | 20.8 | 38.0 | 100.1 | 68.8 | 85.7 | 119.4 | 100.9 | 81.2 | 119.9 | 108.7 | **145.9** |
| Maze2D-medium | 76.6 | 80.1 | 104.4 | 107.3 | 93.2 | 96.4 | 127.7 | 132.0 | 118.0 | **156.0** | 122.3 |
| Maze2D-large-dense | 91.9 | 78.5 | 181.7 | 105.8 | 133.5 | 115.4 | 149.3 | **217.2** | 160.0 | 155.9 | 166.1 |
| Maze2D-large | 28.5 | 42.7 | 103.2 | 115.4 | 82.2 | 58.3 | 126.5 | **228.1** | 140.0 | 97.3 | 152.9 |
| Average | 49.7 | 57.4 | 103.6 | 74.8 | 89.4 | 83.3 | 106.1 | 120.6 | 115.7 | 111.4 | **126.9** |

marks using full replay buffers (containing millions of transitions). Table 9 reports the normalized scores on locomotion tasks. CQL+$C^4$ demonstrates robust performance, surpassing standard baselines like IQL and TD3BC, and remaining highly competitive with recent strong methods such as BPPO and A2PR. While our primary contribution lies in solving the small-sample dilemma, these results confirm that $C^4$ is a safe and versatile plug-in: it significantly boosts data efficiency in sparse regimes while automatically adapting to large-scale datasets without requiring manual tuning or deactivation.

**Consistency across varying data budgets.** To scrutinize the scaling behavior more granularly, we visualize the performance trends as the training set size grows. First, we benchmark eight offline RL algorithms on the challenging Adroit tasks (Pen, Hammer, Door, and Relocate) under cloned regimes, as shown in Fig. 10. Each curve traces the performance across budgets of $\{1, 5, 10, 100, 500\}$ thousand state-action pairs. Our method attains the highest returns on most manipulation tasks and scales favorably with data availability, pointing to strong robustness and transfer capability under contact-rich dynamics.

Separately, we assess seven offline RL algorithms on the standard Gym domains (HalfCheetah, Hopper, Walker2d, and Ant) across diverse dataset qualities (medium-expert, expert, medium, medium-replay, and full-replay), as shown in Fig. 9. These curves utilize budgets of $\{3, 5, 10, 50, 100\}$ thousand pairs. While all baselines naturally benefit from larger datasets, our approach consistently achieves the top trajectories and exhibits the clearest gains on replay datasets. This suggests that $C^4$ not only adapts to scale but also generalizes better to varied dynamics and distribution shifts throughout the learning spectrum.

### E.6   COVARIANCE CONTROL UNDER $C^4$

We monitor the cross covariance between current and next-state input gradients throughout training. For a mini-batch B, define $g_i = \nabla_x Q_\phi(x_i), g_i' = \nabla_{x'} Q_{\phi'}(x_i')$. The empirical cross covariance matrix is $\Sigma_\mathcal{B} = \frac{1}{|\mathcal{B}|-1} \sum_{i \in \mathcal{B}} \left( g_i' - \bar{g}' \right) \left( g_i - \bar{g} \right)^\top$. We report the dimension-normalized trace $\mathrm{tr}_n(\Sigma_\mathcal{B}) = \frac{1}{m} \mathrm{tr}(\Sigma_\mathcal{B})$, which targets the harmful TD component in the variance decomposition $\mathrm{Var}[\delta] \approx A +$

Table 5: Normalized score on D4RL AntMaze tasks with reduced-size datasets (10k samples).

| Task | TD3BC | CQL | IQL | LN | DR3 | SORL | DOGE | BPPO | TSRL | A2PR | **Ours** |
|------|-------|-----|-----|-----|-----|------|------|------|------|------|------|
| AntMaze-umaze | 20.9 | 21.1 | 69.8 | 48.1 | 41.1 | 65.3 | **78.9** | 53.3 | 74.3 | 69.2 | 68.7 |
| AntMaze-umaze-diverse | 16.8 | 43.3 | 58.1 | 29.7 | 0.5 | 39.1 | 43.8 | 45.1 | 57.6 | 27.5 | **65.9** |
| AntMaze-medium-diverse | 0.0 | 0.0 | 0.0 | 0.0 | 0.0 | 0.0 | 0.0 | 0.3 | 0.0 | 0.0 | **7.2** |
| AntMaze-medium-play | 0.0 | 0.0 | 0.4 | 0.0 | 0.0 | 0.0 | 0.0 | 0.0 | 0.0 | 0.3 | **2.9** |
| AntMaze-large-diverse | 0.0 | 0.0 | 0.0 | 0.0 | 0.0 | 0.0 | 0.0 | 0.0 | 0.0 | 0.0 | **9.8** |
| AntMaze-large-play | 0.0 | 0.0 | 0.0 | 0.0 | 0.0 | 0.0 | 0.0 | 0.0 | 0.0 | 0.2 | **7.3** |
| Average | 6.3 | 10.7 | 21.4 | 13.0 | 6.9 | 17.4 | 20.5 | 16.5 | 22.0 | 16.2 | **27.0** |

Table 6: Comparison of CQL with and without $C^4$ on full-size (see Table 15) D4RL locomotion datasets. The module improves the total score without inducing performance degradation.

| Method | Hop.-m | Hop.-mr | Hop.-me | Half.-m | Half.-mr | Half.-me | Wal.-m | Wal.-mr | Wal.-me |
|--------|--------|---------|---------|---------|----------|----------|--------|---------|---------|
| CQL | 58.5 | 95.0 | **105.4** | 41.0 | **45.5** | 91.6 | 72.5 | 77.2 | **108.8** |
| CQL + $C^4$ | **85.9** | **100.7** | 89.4 | **48.5** | 44.7 | **91.6** | **81.8** | **90.9** | 108.6 |

$B - C, C = 2\gamma k k' \operatorname{Cov}\Big(\langle \mathbf{w}', \nabla_{x'} Q_{\phi'}(x') \rangle, \; \langle \mathbf{w}, \nabla_x Q_\phi(x) \rangle\Big)$. In Fig. 11 we visualize $\operatorname{tr}_n(\Sigma_\mathcal{B})$ over training. Across Locomotion tasks, $C^4$ keeps $\operatorname{tr}_n(\Sigma_\mathcal{B})$ substantially lower than LN, CQL, and DR3 during most of training, with the gap most pronounced on replay datasets. This indicates that $C^4$ suppresses the cross covariance while preserving stable learning.

### E.7 ROBUSTNESS AND EFFICIENCY

In this subsection, we empirically study the robustness of our method to key hyperparameters through ablation studies and analyze its computational efficiency. We also provide visual insights into the clustering mechanism.

**Cluster Visualization.** To intuitively understand the learned representations, we visualize the clustering structure for CQL+$C^4$ and TD3BC+$C^4$ in Fig. 12 and Fig. 13, respectively. For each method, we sample snapshots across the full training process and plot the stacked gradient pairs after t-SNE reduction to two dimensions, with points colored by their mixture assignments. The figures demonstrate that our method effectively separates distinct gradient modes throughout representative stages of training.

**Sensitivity to the number of clusters $K$.** We first investigate the impact of the number of clusters $K$ on performance. Table 10 reports a sensitivity analysis on 10k-sample D4RL tasks. We observe that performance improves rapidly as $K$ increases from 1 and quickly plateaus around $K = 3$–5. Performance remains stable even when $K$ is as large as 20, indicating that the method is not overly sensitive to this choice. Based on these results, we recommend $K = 5$ as a robust default for small, low-coverage datasets.

For larger sample regimes, we further examine the *effective* number of clusters used by the algorithm when initialized with $K = 20$. As shown in Table 11, only a handful of clusters are effectively occupied, and this number decreases as the dataset becomes denser (from $10^4$ to $\sim 10^6$ samples). This suggests that for large or high-coverage datasets, $K$ can be safely reduced without sacrificing performance.

**Sensitivity to the penalty weight $\lambda$.** We next study the effect of the penalty weight $\lambda$ in the cross-covariance regularizer. Table 12 summarizes the results across medium ("-m"), medium-replay ("-mr"), and medium-expert ("-me") D4RL tasks. We find that the optimal $\lambda$ depends on dataset quality and the dispersion of the feature space. For high-quality datasets (e.g., medium-expert), trajectories tend to form locally dense clusters with small within-cluster covariance, and a relatively small penalty ($\lambda \in [0.05, 0.1]$) suffices. In contrast, replay-style datasets or those with pronounced distribution shift exhibit more scattered features and larger cross-covariances, benefiting from larger

Table 7: Performance of BPPO with and without $C^4$ on 10k-sample datasets (normalized scores).

| Method | Ant | Halfcheetah | Hopper | Walker2d | AntMaze | Pen | Door | Kitchen | Avg. |
|---|---|---|---|---|---|---|---|---|---|
| BPPO | 85.5 | 22.3 | 27.9 | 61.3 | 63.4 | 30.4 | -0.1 | 15.9 | 38.3 |
| BPPO + $C^4$ | 87.1 | 36.1 | 56.0 | 85.2 | 66.2 | 74.5 | -0.1 | 23.1 | **53.5 (+39.6%)** |

Table 8: Performance of A2PR with and without $C^4$ on 10k-sample datasets.

| Method | Ant | Halfcheetah | Hopper | Walker2d | AntMaze | Pen | Door | Kitchen | Avg. |
|---|---|---|---|---|---|---|---|---|---|
| A2PR | 66.7 | 32.4 | 49.7 | 56.5 | 42.5 | -0.1 | -0.3 | 0.3 | 31.0 |
| A2PR + $C^4$ | 92.6 | 49.0 | 83.6 | 101.3 | 75.8 | 47.9 | 0.1 | 10.2 | **57.6 (+85.9%)** |

penalties (e.g., $\lambda \in [0.3, 0.5]$). A practical heuristic is to choose $\lambda$ such that the weighted penalty term is on the same order of magnitude as the temporal-difference loss.

**Computational efficiency of clustering.** We now analyze the computational profile of the gradient-space clustering procedure. Theoretically, we implement clustering via an Expectation–Maximization algorithm with complexity $O(KNm)$ for $N$ samples, $K$ clusters, feature dimension $m$, and number of EM iterations $t$ (Bishop & Nasrabadi, 2006). Importantly, we do *not* re-cluster at every gradient step: we subsample representative points from the replay buffer and perform clustering only periodically. Once clusters are formed, computing the localized covariance penalty during each critic update is linear in the batch size and adds negligible overhead.

Empirically, Table 13 reports the proportion of total training time spent in clustering across tasks and dataset scales. For our primary target regime of scarce data (e.g., $10^3$–$10^4$ samples), the overhead is about $1\%$ of the total training time, which is effectively negligible. On larger, million-scale datasets, the overhead increases to roughly $15$–$20\%$, which we consider a reasonable trade-off given the substantial stability and performance improvements (up to $\sim 30\%$ gains in return) observed in our main results. Overall, the cost of $C^4$ remains comparable to that of commonly used regularizers.

**Clustering update frequency.** Finally, we examine how often clusters need to be updated. Table 14 reports the performance of CQL+$C^4$ when varying the clustering frequency, defined as the number of critic updates between two clustering runs. We observe a clear "sweet spot" around 100–200 updates: updating too frequently (e.g., every 20 steps) can destabilize the critic by introducing rapidly changing group assignments, while updating too infrequently (e.g., every 1000 steps) leads to stale cluster statistics and degraded performance.

Consequently, we adopt a moderate default frequency of 200 critic updates, which captures distributional shifts in the replay buffer while maintaining stable learning. Importantly, this robustness to update frequency directly supports our efficiency claims: because optimal performance is achieved with intermittent rather than per-step clustering, the clustering cost can be heavily amortized over many critic updates, keeping the overall runtime overhead low (cf. Table 13).

Overall, these analyses show that $C^4$ is robust to reasonable choices of $K$, $\lambda$, and clustering frequency, and that it introduces only modest computational overhead even on large-scale offline RL benchmarks.

Table 9: Normalized scores on locomotion tasks with full datasets ($\sim 10^6$ samples).

| Task | BC | TD3BC | IQL | DOGE | TSRL | BPPO | A2PR | CQL | CQL+$C^4$ |
|------|------|------|------|------|------|------|------|------|------|
| Hopper-m | 52.9 | 59.3 | 66.3 | 98.6 | 86.7 | 93.9 | **100.8** | 58.5 | 85.9 (+27.4) |
| Hopper-mr | 18.1 | 60.9 | 94.7 | 76.2 | 78.7 | 92.5 | **101.5** | 95.0 | 100.7 (+5.7) |
| Hopper-me | 52.5 | 98.0 | 91.5 | 102.7 | 95.9 | **112.8** | 112.1 | 105.4 | 89.4 (−16.0) |
| Hopper-e | 108.0 | 100.1 | 99.3 | 107.4 | 110.0 | 113.2 | **115.0** | 98.4 | 110.1 (+11.7) |
| Halfcheetah-m | 42.6 | 48.3 | 47.4 | 40.6 | 48.2 | 44.0 | **68.6** | 41.0 | 48.5 (+7.5) |
| Halfcheetah-mr | 55.2 | 44.6 | 44.0 | 42.8 | 42.2 | 41.0 | **56.6** | 45.5 | 44.7 (−0.8) |
| Halfcheetah-me | 55.2 | 90.7 | 86.7 | 78.7 | 92.0 | 92.5 | **98.3** | 91.6 | 91.6 (+0.0) |
| Halfcheetah-e | 92.2 | 82.1 | 88.9 | 93.5 | 94.3 | 95.3 | **103.2** | 95.6 | 93.2 (−2.4) |
| Walker2d-m | 75.3 | 83.7 | 78.3 | 86.8 | 77.5 | 83.6 | **89.7** | 72.5 | 81.8 (+9.3) |
| Walker2d-mr | 26.0 | 81.8 | 73.9 | 87.3 | 66.1 | 77.6 | **94.4** | 77.2 | 90.9 (+13.7) |
| Walker2d-me | 107.5 | 110.1 | 109.6 | 110.4 | 106.4 | 113.1 | **114.7** | 108.8 | 108.6 (−0.2) |
| Walker2d-e | 107.9 | 108.2 | 109.7 | 107.3 | 110.2 | 113.8 | **114.2** | 110.3 | 109.7 (−0.6) |
| Average | 66.1 | 80.7 | 82.5 | 86.1 | 84.0 | 89.4 | **97.4** | 83.3 | 87.9 (+4.6) |

Table 10: Sensitivity of CQL+$C^4$ to the number of clusters $K$ on 10k-sample D4RL tasks.

| Method | $K$ | Ant | Halfcheetah | Hopper | Walker2d | AntMaze | Pen | Kitchen |
|------|------|------|------|------|------|------|------|------|
| CQL | – | 21.0 | 39.7 | 43.2 | 26.0 | 21.1 | 1.8 | 0.6 |
| CQL+$C^4$ | 1 | 59.3 | 41.3 | 68.3 | 51.2 | 55.1 | 29.8 | 11.7 |
| CQL+$C^4$ | 2 | 66.5 | **48.2** | 75.8 | 82.6 | 62.9 | 36.0 | 17.1 |
| CQL+$C^4$ | 3 | 70.6 | 47.5 | **90.8** | 87.8 | 67.3 | 51.3 | 19.5 |
| CQL+$C^4$ | 5 | **71.3** | 46.0 | 85.3 | **96.3** | 68.7 | **58.2** | **20.9** |
| CQL+$C^4$ | 7 | 69.6 | 46.4 | 82.1 | 94.9 | **72.4** | 58.0 | 19.3 |
| CQL+$C^4$ | 10 | 68.5 | 46.3 | 78.6 | 90.6 | 68.5 | 52.4 | 18.0 |
| CQL+$C^4$ | 20 | 69.0 | 44.9 | 73.3 | 87.3 | 62.8 | 54.0 | 19.8 |

Table 11: Average number of effective clusters for CQL+$C^4$ when initialized with $K = 20$.

| Samples | Ant | Halfcheetah | Hopper | Walker2d | AntMaze | Pen | Kitchen |
|------|------|------|------|------|------|------|------|
| 10k | 6 | 4 | 4 | 5 | 5 | 5 | 6 |
| $\sim 10^6$ samples | 3 | 2 | 2 | 3 | 4 | 3 | 3 |

Table 12: Sensitivity of CQL+$C^4$ to the penalty weight $\lambda$.

| Task name | $\lambda = 0.0$ | $\lambda = 0.05$ | $\lambda = 0.1$ | $\lambda = 0.3$ | $\lambda = 0.5$ | $\lambda = 1.0$ |
|------|------|------|------|------|------|------|
| Halfcheetah-m | 30.1 | 38.9 | 44.1 | **46.3** | 45.0 | 40.3 |
| Hopper-m | 55.6 | 68.6 | **75.0** | 69.2 | 69.6 | 62.8 |
| Walker2d-m | 55.3 | 57.9 | 61.1 | **65.9** | 65.2 | 63.7 |
| Halfcheetah-mr | 33.6 | 32.3 | 37.8 | 35.6 | **43.1** | 36.6 |
| Hopper-mr | 25.0 | 35.5 | 49.6 | **51.0** | 45.9 | 50.2 |
| Walker2d-mr | 20.1 | 35.3 | 38.1 | 44.7 | **55.4** | 39.0 |
| Halfcheetah-me | 39.9 | **52.2** | 46.9 | 42.1 | 27.4 | 27.0 |
| Hopper-me | 65.0 | 74.2 | **81.3** | 70.8 | 67.6 | 63.5 |
| Walker2d-me | 63.6 | 95.9 | **96.3** | 89.9 | 87.9 | 79.2 |

Table 13: Proportion of total training time spent in gradient-space clustering for CQL+$C^4$.

| Samples | Ant | HalfCheetah | Hopper | Walker2d | AntMaze | Pen | Door | Kitchen |
|------|------|------|------|------|------|------|------|------|
| $10^3$ | 1.0% | 0.9% | 1.2% | 1.2% | 2.9% | 1.7% | 1.5% | 2.0% |
| $\sim 10^6$ | 18.7% | 15.3% | 14.4% | 13.8% | 22.6% | 17.9% | 15.1% | 16.4% |

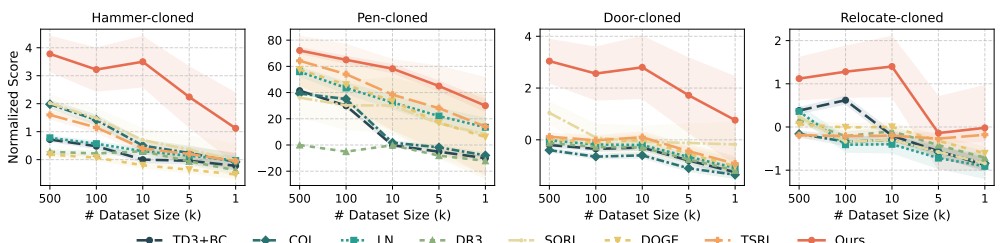

Figure 9: Comparison on Locomotion tasks over different data size.

Figure 10: Comparison of Adroit tasks over different data sizes.

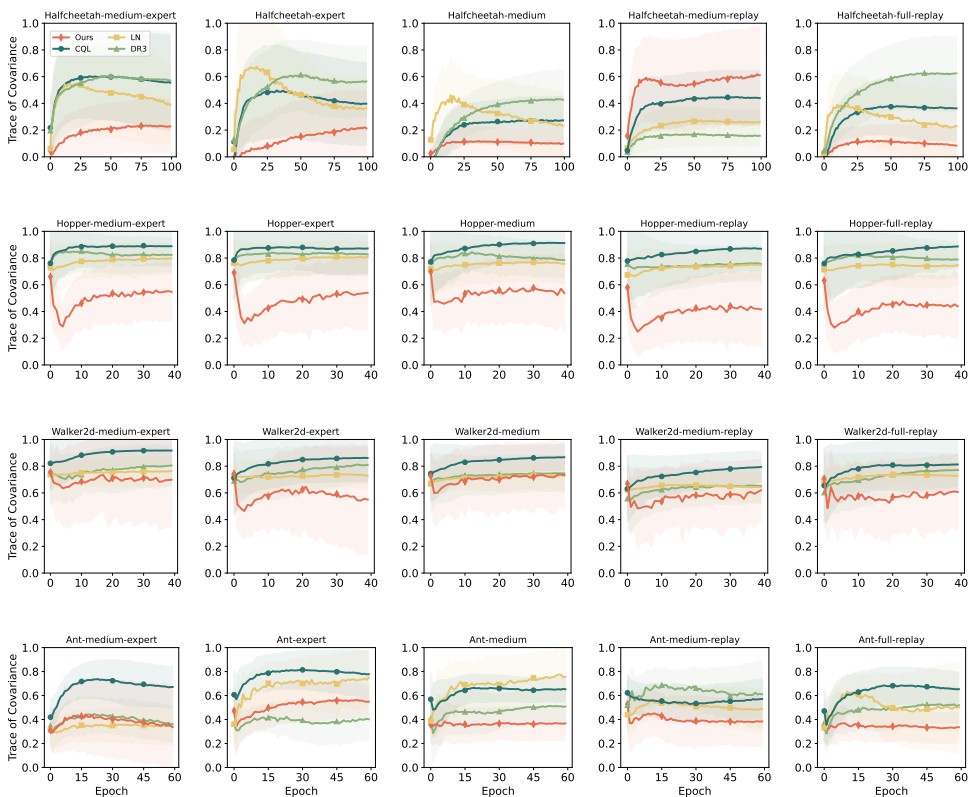

Figure 11: Trace of cross covariance during training on MuJoCo locomotion. Tasks HalfCheetah Hopper Walker2d Ant. Dataset regimes: medium-expert, expert, medium, medium-replay, full-replay. The x-axis is epoch. *The y axis is the trace of the empirical cross covariance, lower is better.* Methods include Ours (CQL+$C^4$), LN (CQL+LN), CQL, and DR3 (CQL+DR3).

Table 14: Sensitivity of CQL+$C^4$ to clustering frequency (number of updates between clustering).

| Task name | Freq. = 20 | Freq. = 50 | Freq. = 100 | Freq. = 200 | Freq. = 500 | Freq. = 1000 |
|---|---|---|---|---|---|---|
| Halfcheetah-m | 39.1 | 36.7 | 41.0 | **46.3** | 36.6 | 22.7 |
| Hopper-m | 66.9 | 65.0 | 64.7 | **69.2** | 60.8 | 61.4 |
| Walker2d-m | 58.9 | 64.1 | **66.5** | 65.9 | 57.8 | 46.5 |
| Halfcheetah-mr | 29.4 | 29.1 | 38.5 | **43.1** | 37.2 | 36.9 |
| Hopper-mr | 51.8 | 54.0 | **54.6** | 45.9 | 36.1 | 37.7 |
| Walker2d-mr | 43.5 | 52.9 | 55.2 | **55.4** | 35.9 | 22.3 |
| Halfcheetah-me | 27.6 | 55.0 | **63.6** | 46.9 | 44.3 | 23.9 |
| Hopper-me | 45.5 | 79.7 | **87.7** | 81.3 | 73.0 | 61.6 |
| Walker2d-me | 69.3 | 74.1 | 91.7 | **96.3** | 75.1 | 74.9 |

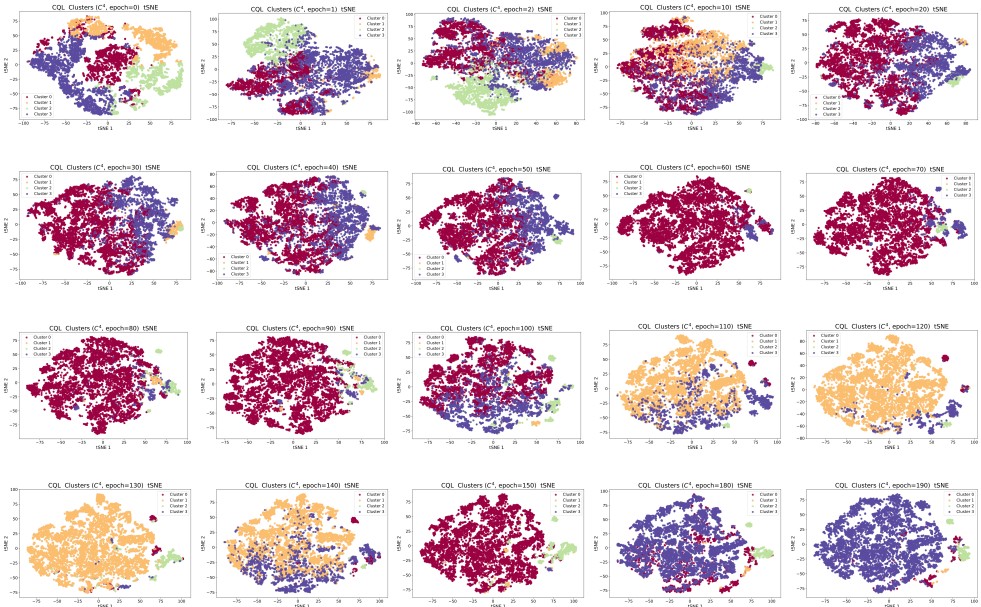

Figure 12: Clustering visualization for CQL+$C^4$.

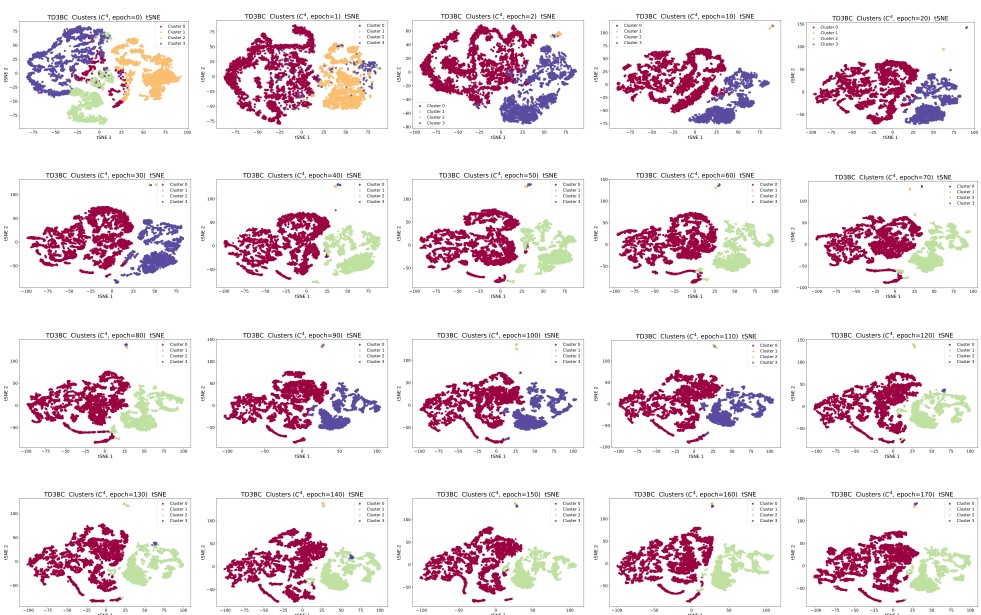

Figure 13: Clustering visualization for TD3BC+$C^4$.

Table 15: Datasets overview.

| Task Name | # Samples | # Traj | random | expert |
|---|---|---|---|---|
| *Maze2D* | | | | |
| maze2d-open | $10^6$ | 22148 | 0.01 | 20.7 |
| maze2d-umaze | $10^6$ | 12460 | 23.9 | 161.9 |
| maze2d-medium | $2 \times 10^6$ | 11889 | 13.1 | 277.4 |
| maze2d-large | $4 \times 10^6$ | 16727 | 6.7 | 274.0 |
| *AntMaze* | | | | |
| antmaze-umaze | $10^6$ | 10154 | 0.0 | 1.0 |
| antmaze-umaze-diverse | $10^6$ | 1035 | 0.0 | 1.0 |
| antmaze-medium-play | $10^6$ | 10767 | 0.0 | 1.0 |
| antmaze-medium-diverse | $10^6$ | 2958 | 0.0 | 1.0 |
| antmaze-large-play | $10^6$ | 13516 | 0.0 | 1.0 |
| antmaze-large-diverse | $10^6$ | 7188 | 0.0 | 1.0 |
| antmaze-ultra-play | $10^6$ | 10536 | 0.0 | 1.0 |
| antmaze-ultra-diverse | $10^6$ | 6076 | 0.0 | 1.0 |
| *Gym-MuJoCo* | | | | |
| hopper-expert | $10^6$ | 1028 | -20.3 | 3234.3 |
| hopper-medium | $10^6$ | 2187 | -20.3 | 3234.3 |
| hopper-medium-replay | 402000 | 2041 | -20.3 | 3234.3 |
| hopper-medium-expert | 1999906 | 3214 | -20.3 | 3234.3 |
| hopper-full-replay | $10^6$ | 3515 | -20.3 | 3234.3 |
| halfcheetah-expert | $10^6$ | 1000 | -280.2 | 12135.0 |
| halfcheetah-medium | $10^6$ | 1000 | -280.2 | 12135.0 |
| halfcheetah-medium-replay | 202000 | 202 | -280.2 | 12135.0 |
| halfcheetah-medium-expert | $2 \times 10^6$ | 2000 | -280.2 | 12135.0 |
| halfcheetah-full-replay | $10^6$ | 1000 | -280.2 | 12135.0 |
| walker2d-expert | $10^6$ | 1001 | 1.6 | 4592.3 |
| walker2d-medium | $10^6$ | 1191 | 1.6 | 4592.3 |
| walker2d-medium-replay | 302000 | 1093 | 1.6 | 4592.3 |
| walker2d-medium-expert | 1999995 | 2191 | 1.6 | 4592.3 |
| walker2d-full-replay | $10^6$ | 1888 | 1.6 | 4592.3 |
| ant-expert | $10^6$ | 1035 | -325.6 | 3879.7 |
| ant-medium | $10^6$ | 1203 | -325.6 | 3879.7 |
| ant-medium-replay | 302000 | 485 | -325.6 | 3879.7 |
| ant-medium-expert | 1999946 | 2237 | -325.6 | 3879.7 |
| ant-full-replay | $10^6$ | 1319 | -325.6 | 3879.7 |
| *Adroit* | | | | |
| pen-human | 5000 | 25 | 96.3 | 3076.8 |
| pen-cloned | $5 \times 10^5$ | 3755 | 96.3 | 3076.8 |
| pen-expert | 499206 | 5000 | 96.3 | 3076.8 |
| hammer-human | 11310 | 25 | -274.9 | 12794.1 |
| hammer-cloned | $10^6$ | 3606 | -274.9 | 12794.1 |
| hammer-expert | $10^6$ | 5000 | -274.9 | 12794.1 |
| door-human | 6729 | 25 | -56.5 | 2880.6 |
| door-cloned | $10^6$ | 4358 | -56.5 | 2880.6 |
| door-expert | $10^6$ | 5000 | -56.5 | 2880.6 |
| relocate-human | 9942 | 25 | -6.4 | 4233.9 |
| relocate-cloned | $10^6$ | 3758 | -6.4 | 4233.9 |
| relocate-expert | $10^6$ | 5000 | -6.4 | 4233.9 |

## F    Use of Large Language Models

In this manuscript, the authors make limited use of large language models, including OpenAI's GPT-5 and DeepSeek, mainly for proofreading, language polishing, improving textual clarity, and reviewing the logical flow of arguments. In addition, Anthropic's Claude is used as an auxiliary tool to assist with coding tasks during the research process. The use of these models is restricted to supportive roles in writing and technical implementation; they do not contribute to research design, data collection, data analysis, or the formulation of scientific claims. All substantive content and conclusions remain entirely the responsibility of the authors.

