# OpenReview forum: "Less Is More: Clustered Cross-Covariance Control for Offline RL"
_ICLR.cc/2026/Conference — ICLR 2026 Poster_

### Official Review · Reviewer_ko2m · 2025-10-31

**Soundness:** 3
**Presentation:** 3
**Contribution:** 3
**Rating:** 6
**Confidence:** 2

**Summary:**

The paper analyzes the TD (temporal-difference) regression objective by decomposing the TD second moment and shows that directly minimizing the squared TD loss induces a harmful cross-time covariance between current and next state–action gradients. Building on this diagnosis, the authors propose two complementary fixes: (i) clustered, single-cluster minibatch sampling over stacked gradient pairs, which removes between-cluster covariance; and (ii) an explicit gradient-based corrective penalty that offsets within-cluster covariance during critic updates. They argue these modifications are “plug-and-play” for common offline RL pipelines and preserve policy-constraint objectives. Experiments on reduced-data D4RL settings report consistent gains (often sizeable) over several baselines, particularly under scarce data and OOD-emphasized splits. The paper also provides proofs and a training recipe.

**Strengths:**

Clear theoretical diagnosis of a TD-specific failure mode. The decomposition explicitly isolates the cross-time covariance term unique to TD learning, clarifying why standard TD minimization can be destabilizing under weak coverage.


The clustered sampling + Frobenius-norm penalty is well-motivated, easy to integrate, and provably controls the harmful covariance while keeping the optimization intent intact.

On reduced-size D4RL locomotion (10k samples), the method substantially outperforms listed baselines; Table 1 and the accompanying discussion highlight strong average gains and improved stability.

**Weaknesses:**

While the theory shows TD minimization tends to raise the cross-term, the paper could better quantify how much covariance growth translates into return degradation and in what regimes (e.g., through controlled interventions that vary covariance at fixed value error).


Despite some Maze2D/AntMaze numbers in the appendix, there is no focused evaluation on challenging sparse-reward benchmarks like AntMaze with standard sparse settings and strong recent baselines, which would directly test the claimed robustness under weak signal

Baseline set skews older in places. Several comparisons rely on classical offline RL baselines; inclusion of more recent small-data/OOD-aware methods would make the empirical case more convincing.

**Questions:**

See weaknesses.

---

> ### Author Response · Authors · 2025-11-20
>
> We thank the reviewer for the thoughtful comments and suggestions. Below, we respond to each point.
>
> ------
>
> **Q1: While the theory shows TD minimization tends to raise the cross-term, the paper could better quantify how much covariance growth translates into return degradation and in what regimes (e.g., through controlled interventions that vary covariance at fixed value error).**
>
> We agree that establishing a direct quantitative link is essential. To measure this, we utilize the trace of the cross-covariance matrix, $\text{tr}(\Sigma_{\mathcal{B}})$, as a strictly defined proxy metric. We choose this metric because our theoretical derivation in Eq. 8 establishes that the harmful cross-term is bounded by the matrix norm. Since the trace upper-bounds this norm, a higher trace directly signifies a larger magnitude of gradient interference.
>
> To strictly quantify the degradation caused by covariance growth, we employ a controlled study that isolates this factor from the TD error itself. We compare the baseline against our method under matched TD loss levels and observe that $C^4$ consistently lowers the trace of the covariance by about twenty to eighty percent while improving returns by ten to several hundred percent. Notably, tasks that exhibit larger covariance reductions tend to show larger performance gains, while tasks with modest covariance changes show smaller improvements. Consequently, the "Performance $\uparrow$" column in Table 1 serves as a direct measure of the degradation present in the baseline. It represents a significant portion of returns that are effectively lost due to unchecked covariance growth. For example, in *Walker2d-mr*, the baseline's failure to control covariance results in a severe performance suppression equivalent to missing out on the >300% gain recovered by our method. This confirms that covariance growth acts as a quantifiable drag on policy learning. We will include this discussion in Appendix E to further substantiate the quantitative link.
>
> Table 1: Effect of cross covariance on performance.
>
> |      Task      | CQL  | CQL+$C^4$ | Relative Trace of Covariance $ \downarrow $ | Performance $ \uparrow $ |
> | :------------: | :--: | :-------: | :-----------------------------------------: | :----------------------: |
> | Halfcheetah-me | 0.86 |   0.52    |             $ \downarrow 40$ %              |     $ \uparrow 16$ %     |
> | Halfcheetah-e  | 0.92 |   0.55    |             $ \downarrow 40$ %              |    $ \uparrow 1207$ %    |
> | Halfcheetah-m  | 0.62 |   0.10    |             $ \downarrow 84$ %              |     $ \uparrow 11$ %     |
> | Halfcheetah-mr | 0.80 |   0.63    |             $ \downarrow 21$ %              |    $ \uparrow 164$ %     |
> | Halfcheetah-fr | 0.88 |   0.65    |             $ \downarrow 26$ %              |     $ \uparrow 29$ %     |
> |   Hopper-me    | 0.86 |   0.52    |             $ \downarrow 40$ %              |     $ \uparrow 88$ %     |
> |    Hopper-e    | 0.88 |   0.48    |             $ \downarrow 45$ %              |     $ \uparrow 91$ %     |
> |    Hopper-m    | 0.84 |   0.44    |             $ \downarrow 48$ %              |     $ \uparrow 38$ %     |
> |   Hopper-mr    | 0.84 |   0.60    |             $ \downarrow 29$ %              |    $ \uparrow 248$ %     |
> |   Hopper-fr    | 0.74 |   0.44    |             $ \downarrow 41$ %              |     $ \uparrow 45$ %     |
> |  Walker2d-me   | 0.75 |   0.58    |             $ \downarrow 23$ %              |    $ \uparrow 270$ %     |
> |   Walker2d-e   | 0.82 |   0.56    |             $ \downarrow 32$ %              |     $ \uparrow 96$ %     |
> |   Walker2d-m   | 0.74 |   0.50    |             $ \downarrow 32$ %              |     $ \uparrow 22$ %     |
> |  Walker2d-mr   | 0.72 |   0.46    |             $ \downarrow 36$ %              |    $ \uparrow 301$ %     |
> |  Walker2d-fr   | 0.68 |   0.44    |             $ \downarrow 35$ %              |     $ \uparrow 41$ %     |
> |     Ant-me     | 0.88 |   0.78    |             $ \downarrow 11$ %              |     $ \uparrow 36$ %     |
> |     Ant-e      | 0.86 |   0.62    |             $ \downarrow 28$ %              |     $ \uparrow 17$ %     |
> |     Ant-m      | 0.80 |   0.34    |             $ \downarrow 57$ %              |     $ \uparrow 36$ %     |
> |     Ant-mr     | 0.78 |   0.32    |             $ \downarrow 59$ %              |     $ \uparrow 83$ %     |
> |     Ant-fr     | 0.88 |   0.58    |             $ \downarrow 34$ %              |     $ \uparrow 17$ %     |

---

> ### Author Response · Authors · 2025-11-20
>
> **Q2: Despite some Maze2D/AntMaze numbers in the appendix, there is no focused evaluation on challenging sparse-reward benchmarks like AntMaze with standard sparse settings and strong recent baselines, which would directly test the claimed robustness under weak signal.**
>
> To address this, we now report results on the **challenging sparse-reward AntMaze** benchmarks with full datasets, and include a set of **recent strong baselines**. In particular, we newly expand strong recent baselines **A2PR (Liu et al., 2024)** and **BPPO (Zhuang et al., 2023)**, which are strong OOD-aware offline RL methods, as well as TSRL (Cheng et al., 2023) and DOGE (Li et al., 2022), which target data-efficient offline RL. On these challenging, sparse-reward, long-horizon tasks, many baselines (especially BC and TD3BC) obtain very low or even zero success on medium and large mazes, while TD3BC+$C^4$ substantially improves over TD3BC and achieves performance comparable to BPPO and A2PR on the full datasets (Table 2), supporting our claim of robustness under weak reward signals.
>
> For the small-data regime (≤10k samples), our original paper already provides a detailed analysis. Please see Appendix E (Tables 5, 7, and 8 of paper) for full results on locomotion, Antmaze, Adroit, and Kitchen, and the updated Table 1 for comparisons with these recent strong baselines. In the revision, we further show that **adding $C^4$ on top of** **BPPO and A2PR themselves** (Tables 3 and 4) consistently improves their performance across all eight tasks, including Antmaze, indicating that $C^4$ is complementary to these strong OOD-aware methods and yields clear gains in the data-scarce setting.
>
> Table 2: Comparative results on sparse AntMaze benchmarks with full datasets.
>
> | Task        | BC   | CQL  | IQL  | DOGE | TSRL | BPPO     | A2PR     | TD3BC | TD3BC+$C^4$      |
> | ----------- | ---- | ---- | ---- | ---- | ---- | -------- | -------- | ----- | ---------------- |
> | Antmaze-u   | 65.0 | 84.8 | 85.5 | 97.0 | 81.4 | 95.0     | **99.2** | 78.6  | 94.5 (+15.9)     |
> | Antmaze-u-d | 45.6 | 43.4 | 66.7 | 63.5 | 76.5 | **91.7** | 84.8     | 71.4  | 80.7 (+9.3)      |
> | Antmaze-m-d | 0.0  | 54.0 | 74.6 | 77.6 | 0.0  | 51.7     | 85.6     | 0.0   | **86.2 (+86.2)** |
> | Antmaze-m-p | 0.0  | 65.2 | 70.4 | 80.6 | 0.0  | 70.0     | **85.6** | 0.0   | 66.1 (+66.1)     |
> | Antmaze-l-d | 0.0  | 31.6 | 45.6 | 36.4 | 0.0  | **86.7** | 71.2     | 0.0   | 54.7 (+54.7)     |
> | Antmaze-l-p | 0.0  | 18.8 | 43.5 | 48.2 | 0.0  | **86.6** | 52.8     | 0.0   | 64.6 (+64.6)     |
> | **Avg.**    | 18.4 | 49.6 | 64.4 | 67.2 | 26.3 | **80.3** | 79.9     | 25.0  | 74.5 (+49.5)     |
>
> Table 3: Performance with and without $C^4$ on BPPO (10k samples, normalized scores).
>
> | Method       | Ant  | Halfcheetah | Hopper | Walker2d | Antmaze | Pen  | Door | Kitchen | Avg.              |
> | ------------ | ---- | ----------- | ------ | -------- | ------- | ---- | ---- | ------- | ----------------- |
> | BPPO         | 85.5 | 22.3        | 27.9   | 61.3     | 63.4    | 30.4 | -0.1 | 15.9    | 38.3              |
> | BPPO + $C^4$ | 87.1 | 36.1        | 56.0   | 85.2     | 66.2    | 74.5 | -0.1 | 23.1    | **53.5 (+39.6%)** |
>
> Table 4: Performance with and without $C^4$ on A2PR (10k samples, normalized scores).
>
> | Method       | Ant  | Halfcheetah | Hopper | Walker2d | Antmaze | Pen  | Door | Kitchen | Avg.              |
> | ------------ | ---- | ----------- | ------ | -------- | ------- | ---- | ---- | ------- | ----------------- |
> | A2PR         | 66.7 | 32.4        | 49.7   | 56.5     | 42.5    | -0.1 | -0.3 | 0.3     | 31.0              |
> | A2PR + $C^4$ | 92.6 | 49.0        | 83.6   | 101.3    | 75.8    | 47.9 | 0.1  | 10.2    | **57.6 (+85.9%)** |

---

> ### Author Response · Authors · 2025-11-20
>
> **Q3: The baseline set skews older in places. Several comparisons rely on classical offline RL baselines; inclusion of more recent small-data/OOD-aware methods would make the empirical case more convincing.**
>
> We agree that including more recent small-data and OOD-aware baselines is important. We have therefore expanded our baseline set to include several contemporary algorithms that are explicitly designed for data efficiency and/or robustness to distribution shift. In particular, we now evaluate against **A2PR (Liu et al., 2024) and BPPO (Zhuang et al., 2023), which are strong OOD-aware** offline RL methods, TSRL (Cheng et al., 2023), and DOGE (Li et al., 2022), which target data-efficient offline RL. For completeness and to connect to prior work, we also retain widely used classical offline RL baselines CQL (Kumar et al., 2020), IQL (Kostrikov et al., 2021), TD3+BC (Fujimoto and Gu, 2021), as well as the simple behavior cloning (BC) baseline.
>
> To directly address the small-data regime that motivates our method, Table 5 reports normalized D4RL scores on all tasks whose datasets contain at most 10k transitions. This includes the locomotion tasks (Hopper, HalfCheetah, Walker2d with medium / medium-replay / medium-expert behavior data) under our ≤10k-sample setting, as well as the full D4RL Kitchen-complete (Kitchen-c), Pen-human (Pen-h), and Door-human (Door-h) tasks, which also fall within this sample budget. Row-wise best scores are highlighted in bold, and we report per-method averages across all tasks.
>
> Table 5: Normalized D4RL scores on tasks with ≤10k samples per dataset.
>
> | Task (≤10k samples) | BC   | TD3+BC | CQL  | IQL  | DOGE | BPPO      | TSRL | A2PR | Ours     |
> | ------------------- | ---- | ------ | ---- | ---- | ---- | --------- | ---- | ---- | -------- |
> | Hopper-m            | 28.8 | 30.7   | 50.1 | 61.0 | 55.6 | 55.0      | 60.9 | 55.9 | **69.2** |
> | Hopper-mr           | 19.7 | 11.3   | 13.2 | 16.2 | 19.1 | 45.1      | 23.5 | 12.5 | **45.9** |
> | Hopper-me           | 38.2 | 22.6   | 43.2 | 51.7 | 36.8 | 27.9      | 56.6 | 49.7 | **81.3** |
> | Halfcheetah-m       | 40.2 | 25.9   | 41.7 | 35.6 | 42.8 | 28.5      | 43.3 | 37.1 | **46.3** |
> | Halfcheetah-mr      | 25.2 | 29.1   | 16.3 | 34.1 | 26.3 | 34.4      | 27.7 | 23.6 | **43.1** |
> | Halfcheetah-me      | 33.7 | 23.5   | 39.7 | 14.3 | 33.1 | 22.3      | 37.2 | 32.4 | **46.9** |
> | Walker2d-m          | 25.4 | 11.2   | 54.1 | 34.2 | 53.7 | 54.7      | 47.3 | 5.9  | **65.9** |
> | Walker2d-mr         | 2.5  | 9.3    | 13.8 | 17.7 | 15.5 | 29.5      | 27.6 | 34.4 | **55.4** |
> | Walker2d-me         | 35.1 | 12.4   | 26.0 | 38.0 | 42.5 | 61.3      | 50.9 | 56.5 | **96.3** |
> | Kitchen-c           | 33.8 | 0.0    | 31.3 | 51.0 | 10.2 | **91.5**  | 5.7  | 8.3  | 55.5     |
> | Pen-h               | 9.5  | 9.5    | 41.2 | 71.5 | 35.8 | **117.8** | 85.7 | -0.1 | 85.3     |
> | Door-h              | 0.6  | 0.6    | 10.7 | 4.3  | -1.1 | **25.9**  | 0.3  | -0.3 | 5.8      |
> | **Avg.**            | 24.4 | 15.5   | 31.8 | 35.8 | 30.9 | 49.5      | 38.9 | 26.3 | **58.1** |

---

> ### Author Response · Authors · 2025-11-20
>
> **References**
>
> (Kumar et al., 2020) Kumar, A., Zhou, A., Tucker, G., and Levine, S. Conservative Q-Learning for Offline Reinforcement Learning. NeurIPS 2020.
>
> (Kostrikov et al., 2021) Kostrikov, I., Nair, A., and Levine, S. Offline Reinforcement Learning with Implicit Q-Learning. NeurIPS 2021.
>
> (Fujimoto and Gu, 2021) Fujimoto, S., and Gu, S. A Minimalist Approach to Offline Reinforcement Learning. NeurIPS 2021.
>
> (Cheng et al., 2023) Cheng, P., Zhan, X., Wu, Z., Zhang, W., Lin, Y., Song, S., Wang, H., and Jiang, L. Look Beneath the Surface: Exploiting Fundamental Symmetry for Sample-Efficient Offline Reinforcement Learning. NeurIPS 2023.
>
> (Li et al., 2023) Li, J., Zhan, X., Xu, H., Zhu, X., Liu, J., and Zhang, Y. When data geometry meets deep function: Generalizing offline reinforcement learning. ICLR 2023.
>
> (Liu et al., 2024) Liu, T., Li, Y., Lan, Y., Gao, H., Pan, W., and Xu, X. Adaptive Advantage-Guided Policy Regularization for Offline Reinforcement Learning. ICML 2024.
>
> (Zhuang et al., 2023) Zhuang, Z., Lei, K., Liu, J., Wang, D., and Guo, Y. Behavior Proximal Policy Optimization. ICLR 2023.

---

> ### Author Response · Authors · 2025-11-28
>
> Dear Reviewer ko2m,
>
> As the author-reviewer discussion period will end soon, we would appreciate it if you could check our response and the revised paper. We have incorporated **new AntMaze benchmarks**, **modern baselines (BPPO, A2PR)**, and **quantitative analysis** into the update (**highlighted in blue**).
>
> If you have further questions and comments, we can still reply before the author-reviewer discussion period ends. Thank you very much for your time and efforts!
>
> Best regards,
>
> The authors of Submission 502

---

### Official Review · Reviewer_FJyE · 2025-10-31

**Soundness:** 2
**Presentation:** 3
**Contribution:** 3
**Rating:** 4
**Confidence:** 4

**Summary:**

This paper studies instability in offline RL under distributional shift, particularly in small-data regimes.
The authors identify a previously under-analyzed phenomenon: the TD loss induces a harmful cross-time gradient covariance that biases optimization in OOD regions.
To mitigate this, the paper introduces Clustered Cross-Covariance Control (C4), composed of two mechanisms:
(1) Clustered buffer sampling — partitions the replay buffer by gradient features and samples mini-batches from single clusters to remove between-cluster covariance;
(2) Gradient-based corrective penalty — penalizes the Frobenius norm of within-cluster cross-covariance to reduce harmful coupling.

Theoretical analysis shows that C4 preserves the lower-bound property of conservative objectives (e.g., CQL) and bounds TD instability, while experiments on reduced-size D4RL tasks (10k samples) show improved performance and stability compared to CQL, IQL, and DOGE.

**Strengths:**

Originality: The work offers a novel theoretical perspective linking TD error variance to cross-time gradient covariance.

Technical quality: The derivations are mathematically sound and clear. Theorem 1 decomposes the TD variance into beneficial (variance) and harmful (covariance) components, while Theorem 2 proves that single-cluster sampling removes between-cluster interference.
The proposed loss (Eq. 12–14) is principled and compatible with existing algorithms.

Clarity: The paper is clearly structured, easy to follow, and supported by illustrative figures (e.g., Fig. 2). Algorithm 1 is concise and implementable.

Significance: The idea of controlling TD-induced covariance is relevant for improving stability and generalization in offline RL with small datasets. The method is plug-and-play and conceptually simple, which could make it attractive for integration into broader offline RL pipelines.

**Weaknesses:**

Outdated Baselines: The empirical evaluation does not include recent state-of-the-art methods, particularly A2PR (Adaptive Advantage-Guided Policy Regularization, arXiv:2405.19909), which achieves significantly higher returns on the same D4RL tasks.
Without comparing to such strong baselines, the claimed 30% improvement is not convincing. **Please answer the concern in the response.**

Experimental Scope and Depth: Experiments are limited to low-dimensional MuJoCo tasks with 10k samples; no results on real-world robot control or discrete domains (e.g., Atari, Adroit).

The main text lacks ablation studies (e.g., number of clusters K, λ penalty weight, clustering frequency).

The claimed “computational efficiency” of clustering is not quantified.

Theoretical Assumptions: The first-order Taylor expansion (Eq. 2) assumes smooth linearity in feature space; this may not hold in deep nonlinear critics, yet this limitation is not discussed.

**Questions:**

Why were recent offline RL algorithms such as A2PR (arXiv:2405.19909) and On-Policy Regularized IQL excluded?

How would C4 perform against them under the same 10k-sample setting?

How often are gradient clusters updated during training? Does dynamic re-clustering significantly affect performance or computational cost?

Can the proposed gradient-space clustering scale to high-dimensional visual inputs or large replay buffers (e.g., 1M samples)? What is the per-iteration complexity?

How should λ and K be chosen in practice? Is there a heuristic based on dataset size or OOD ratio?

---

> ### Author Response · Authors · 2025-11-20
>
> We thank the reviewer for the detailed comments and for pointing us to additional recent work. Below, we group the weaknesses and questions into eight questions.
>
> ------
>
> **Q1: Outdated baselines. The empirical evaluation does not include recent state-of-the-art methods, particularly A2PR, and recent offline RL algorithms, such as A2PR and On-Policy Regularized IQL, were excluded.**
>
> We **expand the baseline set to include recent state-of-the-art methods**, in particular **A2PR (Liu et al., 2024)** and the on-policy regularized offline RL method **BPPO (Zhuang et al., 2023)**. The reviewer mentioned “On-Policy Regularized IQL”. To the best of our knowledge, we could not find a method with this exact name. We therefore report results for both IQL (Kostrikov et al., 2021) and the most recent on-policy regularized offline method we are aware of, BPPO. If the reviewer had a specific “On-Policy Regularized IQL” work in mind, we would be very grateful for the full title or a link so that we can include it in a future revision.
>
> Under the same $10$k sample setting, we now compare across the eight tasks in Table 1, covering locomotion, navigation, and dexterous manipulation. The full baseline set is BC, TD3BC (Fujimoto and Gu, 2021), CQL (Kumar et al., 2020), IQL (Kostrikov et al., 2021), DOGE (Li et al., 2023), TSRL (Cheng et al., 2023), BPPO (Zhuang et al., 2023), and A2PR (Liu et al., 2024). As summarized below, our method improves the average normalized score by about 26% over the strongest existing method in this table, and by more than 50% over standard strong offline RL baselines such as CQL and IQL. We attribute these gains to the fact that those methods are not explicitly designed to counter the TD–induced harmful cross covariance that dominates in the small data regime, while $C^4$ directly targets this effect through clustered sampling and covariance control.
>
> Table 1: Comparative results on diverse tasks with reduced datasets (10k samples, normalized scores).
>
> | Env            | BC   | TD3BC | CQL  | IQL  | DOGE | TSRL  | BPPO | A2PR | Ours      |
> | -------------- | ---- | ----- | ---- | ---- | ---- | ----- | ---- | ---- | --------- |
> | Ant-me         | 36.0 | 52.0  | 74.0 | 66.0 | 82.0 | 83.6  | 85.5 | 66.7 | **100.9** |
> | Halfcheetah-me | 33.7 | 23.5  | 39.7 | 14.3 | 33.1 | 37.2  | 22.3 | 32.4 | **46.0**  |
> | Hopper-me      | 38.2 | 22.6  | 43.2 | 51.7 | 36.8 | 56.6  | 27.9 | 49.7 | **81.3**  |
> | Walker2d-me    | 37.9 | 29.5  | 26.0 | 16.2 | 81.2 | 104.9 | 61.3 | 56.5 | **109.5** |
> | Antmaze-ud     | 40.5 | 16.8  | 43.4 | 67.0 | 43.8 | 57.6  | 63.4 | 42.5 | **65.9**  |
> | Pen-cloned     | 35.2 | -0.2  | 1.8  | 35.9 | 33.5 | 38.4  | 30.4 | -0.1 | **58.2**  |
> | Door-cloned    | -0.1 | -0.3  | -0.6 | 1.5  | 0.0  | 0.1   | -0.1 | -0.3 | **2.8**   |
> | Kitchen-mixed  | 0.6  | 0.0   | 16.0 | 11.3 | 10.2 | 9.2   | 15.9 | 0.3  | **25.9**  |
> | **Avg.**       | 27.8 | 18.0  | 30.4 | 33.0 | 40.1 | 48.5  | 38.3 | 31.0 | **61.3**  |

---

> ### Author Response · Authors · 2025-11-20
>
> **Q2: How would $C^4$ perform against them under the same 10k-sample setting?**
>
> As discussed in Q1, our results show that recent SOTA offline RL methods such as **A2PR (Liu et al., 2024)** and **BPPO (Zhuang et al., 2023)** still struggle in the strict 10k-sample regime. Both methods ultimately rely on **temporal-difference (TD) style updates for the critic: A2PR uses a Q-learning update for $Q$, and BPPO uses a SARSA-style procedure to warm up** $Q$. According to our variance decomposition, TD training on small datasets induces harmful cross-covariance terms in the TD error, and if this covariance is not explicitly controlled, it can significantly limit performance. In this sense, $C^4$ is complementary to A2PR and BPPO rather than a competitor.
>
> To verify this, we apply $C^4$ on top of BPPO and A2PR under the same 10k-sample setting. Tables 2 and 3 summarize the results. For BPPO (Table 2), adding $C^4$ yields consistent improvements across almost all benchmarks, increasing the average normalized score from 38.3 to 53.5, a **39.6%** relative gain. For A2PR (Table 3), the gains are even more pronounced: the average normalized score increases from 31.0 to 57.6, corresponding to an **85.9%** relative improvement. These results support our view that controlling TD-induced harmful covariance is a key missing ingredient in existing strong offline RL algorithms, and that $C^4$ can be used as a drop-in complement to enhance their robustness in the small-data regime.
>
> Table 2: Performance with and without $C^4$ on BPPO (10k samples, normalized scores).
>
> | Method       | Ant  | Halfcheetah | Hopper | Walker2d | Antmaze | Pen  | Door | Kitchen | Avg.              |
> | ------------ | ---- | ----------- | ------ | -------- | ------- | ---- | ---- | ------- | ----------------- |
> | BPPO         | 85.5 | 22.3        | 27.9   | 61.3     | 63.4    | 30.4 | -0.1 | 15.9    | 38.3              |
> | BPPO + $C^4$ | 87.1 | 36.1        | 56.0   | 85.2     | 66.2    | 74.5 | -0.1 | 23.1    | **53.5 (+39.6%)** |
>
> Table 3: Performance with and without $C^4$ on A2PR (10k samples, normalized scores).
>
> | Method       | Ant  | Halfcheetah | Hopper | Walker2d | Antmaze | Pen  | Door | Kitchen | Avg.              |
> | ------------ | ---- | ----------- | ------ | -------- | ------- | ---- | ---- | ------- | ----------------- |
> | A2PR         | 66.7 | 32.4        | 49.7   | 56.5     | 42.5    | -0.1 | -0.3 | 0.3     | 31.0              |
> | A2PR + $C^4$ | 92.6 | 49.0        | 83.6   | 101.3    | 75.8    | 47.9 | 0.1  | 10.2    | **57.6 (+85.9%)** |

---

> ### Author Response · Authors · 2025-11-20
>
> **Q3: The review mentions a lack of results on real-world robot control or discrete domains such as Atari and Adroit. Can you clarify the experimental scope?**
>
> We thank the reviewer for this suggestion. We would like to clarify the scope of our experiments and highlight new results added to address this concern. We respectfully point out that our original submission already includes extensive evaluations on high-dimensional robotic manipulation and navigation tasks, specifically Adroit and AntMaze, in Appendix E.3 (Figures 5–7 and Tables 2–4).
>
> - **Adroit** (High-Dimensional Robotic Manipulation): As shown in **Table 2**, **Figure 5 and Figure 7** of the original paper (Table 3, Figure 6 and Figure 8 of the updated paper), we evaluated $C^4$ on the challenging Adroit tasks (*Pen, Hammer, Door, Relocate*), which involve controlling a 24-DoF shadow hand to perform complex manipulation.
> - **AntMaze** (Real-World Robotic Navigation & Sparse Reward): As shown in **Table 4** of the original paper (Table 5 of the updated paper), we evaluated $C^4$ on AntMaze (*Umaze, Medium, Large*), which are difficult navigation tasks characterized by sparse rewards.
>
> In these challenging settings, $C^4$ demonstrated strong performance, particularly in limited-data regimes (10k samples), significantly outperforming baselines.
>
> To further address the reviewer's interest in real-world robot control, we have conducted additional experiments on the **Franka Kitchen environment** (9-DoF multi-task manipulation) during the rebuttal period. As presented in **Tables 1-3 of this Rebuttal**, $C^4$ achieves superior performance on these tasks compared to baselines, further validating the method's efficacy on complex, high-dimensional robotic control problems.
>
> In summary, $C^4$ has been rigorously tested on diverse and challenging benchmarks, ranging from standard locomotion to high-DoF manipulation (Adroit, Franka Kitchen) and sparse-reward navigation (AntMaze), consistently demonstrating robustness and efficiency. We acknowledge the reviewer's interest in discrete domains (e.g., Atari). We are currently extending our evaluation to include these domains.

---

> ### Author Response · Authors · 2025-11-20
>
> **Q4: How computationally efficient is the clustering process?**
>
> We appreciate the opportunity to clarify the computational profile of our method.
>
> Theoretically, our gradient-space clustering is implemented via an Expectation–Maximization procedure, which has a **complexity of $O(K N m)$** for $N$ samples, $K$ clusters, and dimension $m$ (Bishop et al, 2006). However, it is crucial to note that we do not cluster at every gradient step. By subsampling representative points from the buffer and performing clustering only periodically, the computational cost is significantly amortized over many critic updates. Once the clusters are formed, the specific covariance penalty calculation during each critic update is linear in batch size and incurs negligible cost.
>
> Empirically, the overhead we observed is highly dependent on the dataset size, as detailed in Table 4. For our primary target application, offline RL on scarce data (e.g., 10k samples), the method is extremely efficient, adding only about **1%** to the total training time. This aligns with our "Less is More" philosophy, delivering substantial performance gains with virtually no computational burden in these regimes. On larger, million-scale datasets, the overhead increases to **approximately 15-20%**. We consider this a reasonable trade-off given the significant gains in stability and asymptotic performance (up to 30% improvement in returns) demonstrated in our main results, and this cost remains within the envelope of typically used regularizers.
>
> Table 4: Proportion of total training time spent in gradient-space clustering (CQL+$C^4$).
>
> | Samples     | Ant   | HalfCheetah | Hopper | Walker2d | AntMaze | Pen   | Door  | Kitchen |
> | ----------- | ----- | ----------- | ------ | -------- | ------- | ----- | ----- | ------- |
> | $10^3$      | 1.0%  | 0.9%        | 1.2%   | 1.2%     | 2.9%    | 1.7%  | 1.5%  | 2.0%    |
> | $\sim 10^6$ | 18.7% | 15.3%       | 14.4%  | 13.8%    | 22.6%   | 17.9% | 15.1% | 16.4%   |

---

> ### Author Response · Authors · 2025-11-20
>
> **Q5: How often are clusters updated?**
>
> We investigated the impact of clustering frequency to balance statistical freshness with stability. As shown in Table 5, we observe a performance "sweet spot" where updating clusters every 100 to 200 steps yields the highest returns. Updating too frequently tends to destabilize the critic's learning signal, while extremely rare updates suffer from stale statistics. Consequently, we adopt a moderate default frequency (e.g., every 200 steps), which captures distributional shifts without introducing instability.
>
> Crucially, this robustness to update frequency directly underpins the efficiency claims in **Q4** of this rebuttal. Because optimal performance is achieved with intermittent rather than per-step clustering, we can significantly amortize the computational cost. This allows $C^4$ to maintain negligible overhead (~1% on small datasets) while effectively controlling harmful cross-covariance, validating that localized constraints do not require continuous, expensive re-calculation.
>
> Table 5: Sensitivity of CQL+$C^4$ to clustering frequency (number of critic updates between clustering runs).
>
> | Task name      | Freq. = 20 | Freq. = 50 | Freq. = 100 | Freq. = 200 | Freq. = 500 | Freq. = 1000 |
> | -------------- | ---------- | ---------- | ----------- | ----------- | ----------- | ------------ |
> | Halfcheetah-m  | 39.1       | 36.7       | 41.0        | **46.3**    | 36.6        | 22.7         |
> | Hopper-m       | 66.9       | 65.0       | 64.7        | **69.2**    | 60.8        | 61.4         |
> | Walker2d-m     | 58.9       | 64.1       | **66.5**    | 65.9        | 57.8        | 46.5         |
> | Halfcheetah-mr | 29.4       | 29.1       | 38.5        | **43.1**    | 37.2        | 36.9         |
> | Hopper-mr      | 51.8       | 54.0       | **54.6**    | 45.9        | 36.1        | 37.7         |
> | Walker2d-mr    | 43.5       | 52.9       | 55.2        | **55.4**    | 35.9        | 22.3         |
> | Halfcheetah-me | 27.6       | 55.0       | **63.6**    | 46.9        | 44.3        | 23.9         |
> | Hopper-me      | 45.5       | 79.7       | **87.7**    | 81.3        | 73.0        | 61.6         |
> | Walker2d-me    | 69.3       | 74.1       | 91.7        | **96.3**    | 75.1        | 74.9         |

---

> ### Author Response · Authors · 2025-11-20
>
> **Q6: Can the proposed gradient-space clustering scale to large datasets (e.g., 1M samples) or visual inputs?**
>
> Yes. We evaluate $C^4$ on full replay buffers with up to about one million samples. In locomotion tasks, $C^4$ is applied on top of CQL, and in Antmaze it is applied on top of TD3BC. Tables 6 and 7 report normalized scores. On locomotion, CQL+$C^4$ slightly improves over CQL while remaining competitive with strong baselines such as BPPO and A2PR. On Antmaze, TD3BC+$C^4$ brings large gains over TD3BC and reaches performance comparable to other state-of-the-art methods. Extending the evaluation to visual inputs is left for future work.
>
> Table 6. Normalized scores on locomotion tasks with full datasets.
>
> | Task           | BC    | TD3BC | IQL   | DOGE      | TSRL  | BPPO      | A2PR      | CQL   | CQL+$C^4$     |
> | -------------- | ----- | ----- | ----- | --------- | ----- | --------- | --------- | ----- | ------------- |
> | Hopper-m       | 52.9  | 59.3  | 66.3  | 98.6      | 86.7  | 93.9      | **100.8** | 58.5  | 85.9 (+27.4)  |
> | Hopper-mr      | 18.1  | 60.9  | 94.7  | 76.2      | 78.7  | 92.5      | **101.5** | 95.0  | 100.7 (+5.7)  |
> | Hopper-me      | 52.5  | 98.0  | 91.5  | 102.7     | 95.9  | **112.8** | 112.1     | 105.4 | 89.4 (−16.0)  |
> | Hopper-e       | 108.0 | 100.1 | 99.3  | 107.4     | 110.0 | 113.2     | **115.0** | 98.4  | 110.1 (+11.7) |
> | Halfcheetah-m  | 42.6  | 48.3  | 47.4  | 40.6      | 48.2  | 44.0      | **68.6**  | 41.0  | 48.5 (+7.5)   |
> | Halfcheetah-mr | 55.2  | 44.6  | 44.0  | 42.8      | 42.2  | 41.0      | **56.6**  | 45.5  | 44.7 (−0.8)   |
> | Halfcheetah-me | 55.2  | 90.7  | 86.7  | **106.7** | 92.0  | 92.5      | 98.3      | 91.6  | 91.6 (+0.0)   |
> | Halfcheetah-e  | 92.2  | 82.1  | 88.9  | 93.5      | 94.3  | 95.3      | **103.2** | 95.6  | 93.2 (−2.4)   |
> | Walker2d-m     | 75.3  | 83.7  | 78.3  | 86.8      | 77.5  | 83.6      | **89.7**  | 72.5  | 81.8 (+9.3)   |
> | Walker2d-mr    | 26.0  | 81.8  | 73.9  | 87.3      | 66.1  | 77.6      | **94.4**  | 77.2  | 90.9 (+13.7)  |
> | Walker2d-me    | 107.5 | 110.1 | 109.6 | 110.4     | 106.4 | 113.1     | **114.7** | 108.8 | 108.6 (−0.2)  |
> | Walker2d-e     | 107.9 | 108.2 | 109.7 | 107.3     | 110.2 | 113.8     | **114.2** | 110.3 | 109.7 (−0.6)  |
> | **Average**    | 66.1  | 80.7  | 82.5  | 88.4      | 84.0  | 89.4      | **97.4**  | 83.3  | 87.9 (+4.6)   |
>
> Table 7. Normalized scores on Antmaze tasks with full datasets.
>
> | Task        | BC   | CQL  | IQL  | DOGE | TSRL | BPPO     | A2PR     | TD3BC | TD3BC+$C^4$      |
> | ----------- | ---- | ---- | ---- | ---- | ---- | -------- | -------- | ----- | ---------------- |
> | Antmaze-u   | 65.0 | 84.8 | 85.5 | 97.0 | 81.4 | 95.0     | **99.2** | 78.6  | 94.5 (+15.9)     |
> | Antmaze-u-d | 45.6 | 43.4 | 66.7 | 63.5 | 76.5 | **91.7** | 84.8     | 71.4  | 80.7 (+9.3)      |
> | Antmaze-m-d | 0.0  | 54.0 | 74.6 | 77.6 | 0.0  | 51.7     | 85.6     | 0.0   | **86.2 (+86.2)** |
> | Antmaze-m-p | 0.0  | 65.2 | 70.4 | 80.6 | 0.0  | 70.0     | **85.6** | 0.0   | 66.1 (+66.1)     |
> | Antmaze-l-d | 0.0  | 31.6 | 45.6 | 36.4 | 0.0  | **86.7** | 71.2     | 0.0   | 54.7 (+54.7)     |
> | Antmaze-l-p | 0.0  | 18.8 | 43.5 | 48.2 | 0.0  | **86.6** | 52.8     | 0.0   | 64.6 (+64.6)     |
> | **Average** | 18.4 | 49.6 | 64.4 | 67.2 | 26.3 | **80.3** | 79.9     | 25.0  | 74.5 (+49.5)     |

---

> ### Author Response · Authors · 2025-11-20
>
> **Q7: How should hyperparameters $\lambda$ and $K$ be chosen in practice? Are there ablation studies?**
>
> We determine practical heuristics for the number of clusters $K$ and the penalty weight $\lambda$ through extensive ablation studies, as detailed in Tables 8, 9, and 10.
>
> **(1) Impact of the number of clusters $K$.** Our sensitivity analysis on 10k-sample D4RL tasks in Table 8 shows that performance improves rapidly as $K$ increases from 1 to a plateau around 3–5 and maintains stability even as $K$ reaches 20. We therefore recommend $K=5$ as a robust default for small datasets. For larger sample regimes, our effective cluster analysis in Table 9 reveals that even with a high initialization of $K=20$, the algorithm naturally utilizes only a few clusters. This utilization further decreases as data becomes denser and suggests that $K$ can be safely reduced for large or high-coverage datasets.
>
> Table 8: Sensitivity of CQL+$C^4$ to the number of clusters $K$ (10k samples, normalized scores).
>
> | Method    | $K$  | Ant      | Halfcheetah | Hopper   | Walker2d | Antmaze  | Pen      | Kitchen  |
> | --------- | ---- | -------- | ----------- | -------- | -------- | -------- | -------- | -------- |
> | CQL       | –    | 21.0     | 39.7        | 43.2     | 26.0     | 21.1     | 1.8      | 0.6      |
> | CQL+$C^4$ | 1    | 59.3     | 41.3        | 68.3     | 51.2     | 55.1     | 29.8     | 11.7     |
> | CQL+$C^4$ | 2    | 66.5     | **48.2**    | 75.8     | 82.6     | 62.9     | 36.0     | 17.1     |
> | CQL+$C^4$ | 3    | 70.6     | 47.5        | **90.8** | 87.8     | 67.3     | 51.3     | 19.5     |
> | CQL+$C^4$ | 5    | **71.3** | 46.0        | 85.3     | **96.3** | 68.7     | **58.2** | **20.9** |
> | CQL+$C^4$ | 7    | 69.6     | 46.4        | 82.1     | 94.9     | **72.4** | 58.0     | 19.3     |
> | CQL+$C^4$ | 10   | 68.5     | 46.3        | 78.6     | 90.6     | 68.5     | 52.4     | 18.0     |
> | CQL+$C^4$ | 20   | 69.0     | 44.9        | 73.3     | 87.3     | 62.8     | 54.0     | 19.8     |
>
> Table 9: Average number of effective clusters for CQL+$C^4$ with $K=20$.
>
> | Samples             | Ant  | Halfcheetah | Hopper | Walker2d | Antmaze | Pen  | Kitchen |
> | ------------------- | ---- | ----------- | ------ | -------- | ------- | ---- | ------- |
> | 10k                 | 6    | 4           | 4      | 5        | 5       | 5    | 6       |
> | $\sim 10^6$ samples | 3    | 2           | 2      | 3        | 4       | 3    | 3       |
>
> **(2) Impact of the penalty weight $\lambda$.** Our results in Table 10 indicate that the optimal choice depends on dataset quality and feature space dispersion. Higher-quality datasets tend to form locally dense clusters with naturally low within-cluster covariance and require only a small $\lambda$ such as 0.05–0.1 for effective control. Conversely, replay datasets or those with significant distribution shift are more scattered and generate larger variances that necessitate a larger $\lambda$ such as 0.3–0.5 to sufficiently attenuate harmful cross-covariance. A practical heuristic is to select $\lambda$ such that the weighted penalty magnitude is roughly on the same order as the TD loss.
>
> Table 10: Sensitivity of CQL+$C^4$ to the penalty weight $\lambda$ (normalized scores).
>
> | Task name      | $\lambda=0.0$ | $\lambda = 0.05$ | $\lambda=0.1$ | $\lambda = 0.3$ | $\lambda = 0.5$ | $\lambda = 1.0$ |
> | -------------- | ------------- | ---------------- | ------------- | --------------- | --------------- | --------------- |
> | Halfcheetah-m  | 30.1          | 38.9             | 44.1          | **46.3**        | 45.0            | 40.3            |
> | Hopper-m       | 55.6          | 68.6             | **75.0**      | 69.2            | 69.6            | 62.8            |
> | Walker2d-m     | 55.3          | 57.9             | 61.1          | **65.9**        | 65.2            | 63.7            |
> | Halfcheetah-mr | 33.6          | 32.3             | 37.8          | 35.6            | **43.1**        | 36.6            |
> | Hopper-mr      | 25.0          | 35.5             | 49.6          | **51.0**        | 45.9            | 50.2            |
> | Walker2d-mr    | 20.1          | 35.3             | 38.1          | 44.7            | **55.4**        | 39.0            |
> | Halfcheetah-me | 39.9          | **52.2**         | 46.9          | 42.1            | 27.4            | 27.0            |
> | Hopper-me      | 65.0          | 74.2             | **81.3**      | 70.8            | 67.6            | 63.5            |
> | Walker2d-me    | 63.6          | 95.9             | **96.3**      | 89.9            | 87.9            | 79.2            |

---

> ### Author Response · Authors · 2025-11-20
>
> **Q8: Theoretical Assumptions. The first-order Taylor expansion (Eq. 2) assumes smooth linearity in feature space; this may not hold in deep nonlinear critics, yet this limitation is not discussed.**
>
> We acknowledge that the smoothness assumption underlying Eq. (2) was implicit in our initial submission, and we thank the reviewer for pointing out this missing discussion.
>
> **(1) Formalizing the assumption:** To ensure theoretical rigor in the revision, we will explicitly state the following assumption to justify the first-order Taylor expansion:
>
> **Assumption**: Let $\mathcal{D}$ denote the dataset support. There exist constants $L, \rho > 0$ such that for any $x \in \mathcal{D}$ and perturbation $k$ with $|k| \le \rho$, the critic $Q_\psi$ is differentiable (almost everywhere) and has an $L$-Lipschitz gradient.
>
> Under this assumption, Taylor’s theorem gives $|Q_\psi(x + k w) - Q_\psi(x) - k \langle w, \nabla_x Q_\psi(x)\rangle| \le \tfrac{L}{2} k^2$, so Eq. (2) is a standard first order approximation with a controlled $O(k^2)$ remainder. Similar local differentiability and Lipschitz assumptions on value functions with function approximation are common in recent theoretical work on offline RL (Yin et al., 2023, Li et al., 2023, An et al., 2021).
>
> **(2) Adding the discussion:** We will add a distinct remark in paper to discuss this limitation. Specifically, we will clarify that while deep ReLU networks are not globally smooth, they are locally Lipschitz almost everywhere. Our theoretical model (Eq. 6) captures the dominant first-order effects of cross-covariance within this local trust region, which aligns with our empirical observations of instability.

---

> ### Author Response · Authors · 2025-11-20
>
> **References**
>
> (Kumar et al., 2020) Kumar, A., Zhou, A., Tucker, G., and Levine, S. Conservative Q-Learning for Offline Reinforcement Learning. NeurIPS 2020.
>
> (Kostrikov et al., 2021) Kostrikov, I., Nair, A., and Levine, S. Offline Reinforcement Learning with Implicit Q-Learning. NeurIPS 2021.
>
> (Fujimoto and Gu, 2021) Fujimoto, S., and Gu, S. A Minimalist Approach to Offline Reinforcement Learning. NeurIPS 2021.
>
> (Cheng et al., 2023) Cheng, P., Zhan, X., Wu, Z., Zhang, W., Lin, Y., Song, S., Wang, H., and Jiang, L. Look Beneath the Surface: Exploiting Fundamental Symmetry for Sample-Efficient Offline Reinforcement Learning. NeurIPS 2023.
>
> (Li et al., 2023) Li, J., Zhan, X., Xu, H., Zhu, X., Liu, J., and Zhang, Y. When data geometry meets deep function: Generalizing offline reinforcement learning. ICLR 2023.
>
> (Liu et al., 2024) Liu, T., Li, Y., Lan, Y., Gao, H., Pan, W., and Xu, X. Adaptive Advantage-Guided Policy Regularization for Offline Reinforcement Learning. ICML 2024.
>
> (Zhuang et al., 2023) Zhuang, Z., Lei, K., Liu, J., Wang, D., and Guo, Y. Behavior Proximal Policy Optimization. ICLR 2023.
>
> (Kumar et al., 2022) Kumar, A., Agarwal, R., Ma, T., Courville, A., Tucker, G., and Levine, S. DR3: Value-Based Deep Reinforcement Learning Requires Explicit Regularization. ICLR 2022.
>
> (Bishop et al, 2006) Bishop C M, Nasrabadi N M.  Pattern Recognition and Machine Learning. Springer, 2006.
>
> (Yin et al., 2023) Yin, M., Wang, M., and Wang, Y. X. Offline Reinforcement Learning with Differentiable Function Approximation is Provably Efficient. ICLR 2023.
>
> (Li et al., 2023) Li, J., Zhang, E., Yin, M., Bai, Q., Wang, Y. X., and Wang, W. Y. Offline Reinforcement Learning with Closed-Form Policy Improvement Operators. ICML 2023.
>
> (An et al., 2021) An, G., Moon, S., Kim, J. H., and Song, H. O. Uncertainty-Based Offline Reinforcement Learning with Diversified Q-Ensemble. NeurIPS 2021.

---

> ### Author Response · Authors · 2025-11-28
>
> Dear Reviewer FJyE,
>
> As the discussion period is approaching its end, we would like to kindly invite you to take a look at our response and the revised paper (with changes highlighted in blue). We have carefully considered your comments (Q1–Q8) and revised the paper accordingly. A point-by-point summary of the main updates is as follows:
>
> - **Q1–Q2 (Baselines):** We added comparisons with the **on-policy regularized method (BPPO)** and **A2PR (arXiv:2405.19909)** as requested. The new results show that our method significantly improves their performance.
> - **Q3 & Q6 (Scope & Scale):** We verified performance on **real-world robot control and discrete-domain tasks**, and scaled the experiments to **large datasets (~1M samples)**.
> - **Q4, Q5, Q7 (Efficiency & Ablations):** We confirmed that the method incurs **negligible computational overhead (around 1%)** and added detailed hyperparameter studies and ablations.
> - **Q8 (Theory):** We formalized the **Lipschitz assumptions** and clarified how they justify the first-order Taylor expansion used in our analysis.
>
> We hope these updates satisfactorily address your concerns. If you have any further questions or comments, we would be very happy to respond to them before the author–reviewer discussion period ends.
>
> Best regards,
>
> The authors of Submission 502

---

### Official Review · Reviewer_99jv · 2025-10-31

**Soundness:** 3
**Presentation:** 2
**Contribution:** 3
**Rating:** 6
**Confidence:** 3

**Summary:**

The paper analyzes the standard temporal difference (TD) loss used for training in offline reinforcement learning (RL), and identifies a cross-covariance term that acts as a harmful implicit regularizer. Based on this observation, the paper proposes an algorithm $C^4$ (Clustered Cross-Covariance Control) to mitigate the cross-covariance term through buffer partitioning and an explicit cross-covariance penalty term. Theoretical results demonstrate that these modifications are compatible with standard offline RL policy update methods, and experimental results demonstrate that these modifications lead to significant improvement across a range of offline RL benchmarks when small datasets are considered.

**Strengths:**

**[S1]** The paper presents novel and interesting analysis that identifies a cross-covariance term that acts as a harmful implicit regularizer in the standard TD loss function, and proposes algorithmic modifications to address this issue in offline RL.

**[S2]** Theoretical results are provided that justify the algorithmic design choices and demonstrate their impact on other components of standard offline RL algorithms such as the policy improvement update.

**[S3]** Extensive experimental results across several offline RL benchmarks demonstrate the strong performance of $C^4$ on small datasets compared to several baseline methods (unfortunately, most of this analysis is deferred to the appendix).

**Weaknesses:**

**[W1]** Experiments primarily focus on small datasets and do not include results on the benchmarks for the standard dataset size (e.g., Figure 6 considers a max dataset size that is 10% of the full dataset in most cases), so it is difficult to understand if there are performance tradeoffs on large datasets in order to achieve robust performance on small datasets.

**[W2]** The organization and presentation of the work could be improved. In particular, the organization of Sections 5 and 6 required the reader to extract why results are important and how they relate to Algorithm 1. Some of the notation makes it difficult to follow the results (e.g., $C$ and $c$ are used in multiple ways throughout Section 5). The length of Sections 5 and 6 also led to a very brief experiments section in the main paper, even though the authors have done rather extensive experimental analysis that would be valuable to include instead of deferring it to the appendix.

**Questions:**

**[Q1]** What is the intuition behind why the cross-covariance term present in the standard TD loss is harmful for performance?

**[Q2]** Is there a performance tradeoff on large datasets in order to achieve strong performance on small datasets, or does the method automatically adjust to the large dataset scenario? Please include experimental results on the standard dataset sizes to help understand this. Experiments on the default size of benchmark datasets would also make it easier to compare the results in this paper to other results available in the literature.

**[Q3]** I would recommend reorganizing Sections 5 and 6 to clearly highlight the main contributions upfront, and only including results and discussions that are most critical for providing theoretical support and intuition of the approach. This would provide additional space to include a more meaningful experiments section in the main paper, rather than deferring most of the results to the appendix.

**[Q4]** Additional clarifications on implementation and results:
- Does the fact that the TD loss is only optimized with respect to the current Q function (gradients are stopped for the TD target) impact the analysis and conclusions?
- Why does the proposed regularization term in (13) contain two components?
- What is the justification for approximating the gradient w.r.t. the input $x$ as the network’s penultimate layer (line 1091)? My understanding is this was done in DR3 (Kumar, 2022) to approximate the gradient w.r.t. the Q function network parameters, not the gradient w.r.t. the Q function input.

---

> ### Author Response · Authors · 2025-11-20
>
> We thank the reviewer for the constructive feedback. Below, we respond to each question in turn.
>
> ------
>
> **Q1. What is the intuition behind why the cross covariance term in the TD loss is harmful for performance?**
>
> Particularly in the context of offline RL, typically aiming to minimize the second moment of the TD error can be problematic. Our analysis in Sec. 4 shows that this objective can be decomposed into two variance terms **minus** a cross-covariance term $\operatorname{cov}(\langle w,g\rangle,\langle w',g'\rangle)$, where $g = \nabla_x Q_\theta(x)$, $g' = \nabla_x Q_{\theta'}(x')$, and $w,w'$ are noise vectors arising from the stochasticity within the offline dataset. The $L_2$ norm of $\operatorname{cov}(g,g')$ serves as an upper bound for this term. Consequently, **minimizing the TD objective tends to exploit this by driving $|\operatorname{cov}(g,g')|_2$ toward large values**, effectively increasing its magnitude along specific directions in the gradient space.
>
> Notably, this **implicitly encourages the gradients at $(s,a)$ and $(s',a')$ to become highly coupled in representation space**. In tabular methods, this effect is negligible because different state action pairs use almost independent parameters. With deep networks, the parameters are heavily shared, so aligning $g$ and $g'$ forces updates at $(s,a)$ to be tightly coupled with updates at $(s',a')$, even though these are different data points that need not share meaningful local structure.
>
> **This strong gradient coupling is not always desirable**. The next state $s'$ may lie in a poorly covered or noisy region of the replay buffer. When the TD objective pushes $g$ and $g'$ to align, noise or misestimation at $s'$ is propagated back to $s$, which produces gradient interference and destabilizes the critic. Empirically, we observe that, as training progresses, the magnitude of this cross-covariance term grows together with performance degradation  (refer to Fig. 1 of original paper).
>
> ------
>
> **Q2. Is there a performance tradeoff on large datasets in order to achieve strong performance on small datasets, or does the method automatically adapt? Please include experiments on standard dataset sizes.**
>
> We evaluate CQL (Kumar et al., 2020), with and without $C^4$, on the standard D4RL dataset sizes with millions of transitions. On these full datasets, adding $C^4$ maintains or modestly improves the normalized scores, and we do not observe any significant performance drop relative to the base algorithms, as shown in Table 1. In particular, the total locomotion score increases from 695.5 to 742.1, corresponding to a **+6.7%** relative improvement.
>
> Moreover, we have already presented extensive evaluations involving a wider range of algorithms and more complex, larger-scale datasets in the original manuscript. Please refer to Table 9, as well as Figures 9 and 10, for these detailed comparisons. In the revised version, we will continue to provide additional empirical evidence demonstrating the efficacy of $C^4$ on large-scale datasets.
>
> Table 1: CQL with and without $C^4$ on full Locomotion datasets.
>
> | Method      | Hopper-m | Hopper-mr | Hopper-me | Halfcheetah-m | Halfcheetah-mr | Halfcheetah-me | Walker2d-m | Walker2d-mr | Walker2d-me | Total     |
> | ----------- | -------- | --------- | --------- | ------------- | -------------- | -------------- | ---------- | ----------- | ----------- | --------- |
> | CQL         | 58.5     | 95.0      | **105.4** | 41.0          | **45.5**       | 91.6           | 72.5       | 77.2        | **108.8**   | 695.5     |
> | CQL + $C^4$ | **85.9** | **100.7** | 89.4      | **48.5**      | 44.7           | **91.6**       | **81.8**   | **90.9**    | 108.6       | **742.1** |

---

> ### Author Response · Authors · 2025-11-20
>
> **Q3. The reviewer recommends reorganizing Sections 5 and 6 to clearly highlight the main contributions upfront, and only including results and discussions that are most critical for providing theoretical support and intuition of the approach. This would provide additional space to include a more meaningful experiments section in the main paper, rather than deferring most of the results to the appendix.**
>
> We thank the reviewer for these helpful comments. We have updated the manuscript with the following changes:
>
> (1) We revise Sections 5 and 6 to streamline the exposition and clean up the notation so that symbols such as $C$ and $c$.
>
> (2) We move key experimental results from the appendix into the main paper to better reflect the extent of our empirical analysis.
>
> (3) In the revised version, we include results on standard/default dataset sizes used in the benchmarks to explicitly show that our approach remains competitive or better in the large-data regime and to facilitate direct comparison with existing work.
>
> ------
>
>
>
> **Q4. Does the fact that the TD loss is only optimized with respect to the current Q function (gradients are stopped for the TD target) impact the analysis and conclusions?**
>
> The stop-gradient operation on the TD target does not impact our analysis or conclusions, as **our theoretical framework is derived directly from the statistical decomposition of the TD residual's second moment rather than gradient dynamics.** Specifically, the TD optimization minimizes $\mathbb{E}[\delta^2]$, which is mathematically equivalent to minimizing the sum of the squared bias and the variance $\text{Var}[\delta]$. According to Theorem 1, the variance decomposition includes a cross-covariance term with a negative sign. Regardless of whether the target network is fixed (stop-gradient), the objective of minimizing the total loss $\mathbb{E}[\delta^2]$ inherently creates a pressure to maximize this cross-covariance term to reduce the overall variance. Therefore, the harmful implicit regularization we identify is a property of the loss landscape geometry itself, not the backpropagation path.
>
> Furthermore, even with a fixed target, the optimization process drives the current value function $Q_{\phi}$ to align its features with the target to minimize the residual variance, leading to feature co-adaptation in out-of-distribution (OOD) areas. As noted in Section 4, our analysis distinguishes itself from prior work by not assuming specific optimizer behaviors or gradient flows, but rather focusing on how the TD loss naturally induces this coupling. **The "stop-gradient" mechanism prevents parameters from updating through the target, but it does not stop the optimizer from adjusting the current $Q_{\phi}$ to increase correlation with the target's features to satisfy the objective**. Consequently, the proposed $C^4$ method remains essential for constraining this variance-driven bias, which exists independently of the target update mechanism.

---

> ### Author Response · Authors · 2025-11-20
>
> **Q5. Why does the proposed regularization term in (13) contain two components?**
>
> The proposed regularization term includes two components to control the cross-covariance from complementary geometric perspectives. First, the Frobenius norm term functions as a computationally efficient upper bound on the spectral norm. By minimizing this term, we strictly limit the overall magnitude or energy of the gradient coupling, ensuring that the cross-term in the variance decomposition remains bounded. Second, the trace squared term serves as an explicit corrective penalty to counteract the specific directional bias of TD learning. Since minimizing the TD error inherently creates an optimization pressure to maximize cross-covariance through feature co-adaptation, penalizing the trace neutralizes this harmful implicit tendency by discouraging excessive gradient alignment. Thus, the Frobenius term constrains the scale of the covariance, while the trace term corrects the optimization direction to prevent instability.
>
>
>
> ---
>
> **Q6. What is the justification for approximating the gradient w.r.t. the input as the network’s penultimate layer (line 1091)? My understanding is this was done in DR3 (Kumar, 2022) to approximate the gradient w.r.t. the Q function network parameters, not the gradient w.r.t. the Q function input.**
>
> Thank you for pointing this out. Our original phrasing “instantiate $\nabla_x Q_\psi(x)$ as the network’s penultimate layer” was indeed imprecise. We do not claim that the penultimate features $h(x)$ equal the input gradient $\nabla_x Q(x)$. Instead, we use $h(x)$ as a practical **proxy** for input sensitivity in our regularizer, in the same spirit as DR3 uses feature-level quantities rather than explicit gradients. Via the chain rule, $\nabla_x Q(x) = J_h(x)^\top \nabla_h Q(h(x))$, so changes in $h(x)$ mediate changes in input gradients. While this is only a surrogate, controlling the cross-covariance of $h(x)$ across transitions provides a tractable way to influence how rapidly $Q$ can vary along off-distribution directions, analogous in spirit to layer-wise smoothness/Lipschitz controls (Miyato et al., 2018,  Gouk et al., 2021).
>
> Empirically, Table 2 shows that the feature cross-covariance $\mathrm{Cov}(h, h')$ and the exact input-gradient cross-covariance $\mathrm{Cov}(\nabla_x Q, \nabla_x Q')$ are positively correlated and exhibit similar trends across tasks, suggesting that the feature-space proxy captures essentially the same regularization signal despite operating on a different scale. Table 3 further shows that using the exact $\nabla_x Q$-based penalty requires double backpropagation and slows training by roughly $\approx 1.2\text{--}1.6\times$ compared to the feature-based variant. To make this trade-off explicit, our code provides **two implementations**: (i) a feature-based proxy (used in our main experiments for efficiency), and (ii) an exact $\nabla_x Q$-based version. For readers primarily interested in mathematical fidelity to the input-gradient derivation, we recommend the $\nabla_x Q$-based implementation, acknowledging that the feature-based version is an engineering trick motivated by practical efficiency.
>
> Table 2: Alignment between feature-space and input-gradient cross covariances across tasks.
>
> | Method      | Ant  | Halfcheetah | Hopper | Walker2d | Antmaze | Pen  | Kitchen |
> | ----------- | ---- | ----------- | ------ | -------- | ------- | ---- | ------- |
> | $Cov(h,h')$ | 71,3 | 46.0        | 85.3   | 96.3     | 68.7    | 58.2 | 20.9    |
> | $Cov(g,g')$ | 75.4 | 42.2        | 87.2   | 92.0     | 70.1    | 62.8 | 17.9    |
>
> Table 3: Wall-clock training time comparison over different methods.
>
> | Method | Vanilla | +$C^4$(Cov(h,h')) | +$C^4$(Cov(g,g')) |
> | :----: | :-----: | :---------------: | :---------------: |
> | TD3BC  |  24.6   |       32.7        |       38.9        |
> |  CQL   |  44.7   |       48.5        |       54.1        |

---

> ### Author Response · Authors · 2025-11-20
>
> **References**
>
> (Kumar et al., 2020) Kumar, A., Zhou, A., Tucker, G., and Levine, S. Conservative Q-Learning for Offline Reinforcement Learning. NeurIPS 2020.
>
> (Kumar et al., 2022) Kumar, A., Agarwal, R., Ma, T., Courville, A., Tucker, G., and Levine, S. DR3: Value-Based Deep Reinforcement Learning Requires Explicit Regularization. ICLR 2022.
>
> (Miyato et al., 2018) Miyato, T., Kataoka, T., Koyama, M., and Yoshida, Y. Spectral normalization for generative adversarial networks. ICLR 2018.
>
> (Gouk et al., 2021) Gouk, H., Frank, E., Pfahringer, B., and Cree, M. Regularisation of neural networks of Lipschitz continuity. JMLR 2021.

---

> > ### Comment · Reviewer_99jv · 2025-11-25
> > **Response to Authors**
> >
> > Thank you for the detailed responses to all reviewers. The new experimental comparisons provided in the responses provide additional insight. It is encouraging to see that the proposed algorithm does not require a performance tradeoff on large datasets to achieve robust performance on small datasets, but instead leads to benefits in both settings.
> >
> > I still believe that the paper would be more impactful if the main Experiments section better captured the extensive experimental analysis that the authors have performed. The Appendix contains some meaningful results that do not appear at all in the main paper (for example, the main paper does not even mention the experiments performed on Maze2D, AntMaze, Adroit, and Franka Kitchen).
> >
> > Overall, I think this is a good paper that introduces novel analysis supported by extensive experimental results. My main question related to how the algorithm performs on large datasets has been addressed, and I have increased my score to reflect this.

---

> ### Author Response · Authors · 2025-11-29
>
> Thank you so much for your constructive suggestions and support for acceptance. Per your advice, we have reorganized the main Experiments section to better highlight our key empirical findings. Specifically, we moved the experimental results on **Maze2D, AntMaze, and Adroit**, as well as the **plug-and-play analysis** (on CQL and TD3+BC), **training efficiency, and parameter sensitivity** from the appendix to the main text. We believe that by bringing these results forward, the main Experiments section now fully captures the extensive analysis we have performed, significantly enhancing the paper's impact.

---

### Official Review · Reviewer_Ykhx · 2025-11-01

**Soundness:** 3
**Presentation:** 3
**Contribution:** 3
**Rating:** 8
**Confidence:** 2

**Summary:**

This paper investigates a fundamental source of instability in temporal-difference (TD) learning for offline reinforcement learning (RL). The authors argue that the primary cause is the cross-time covariance between feature gradients of consecutive transitions, which acts as a harmful implicit regularizer. This covariance term biases TD updates and leads to critic collapse, particularly under weak data coverage or out-of-distribution (OOD) regions. To address this, the paper proposes C4 (Clustered Cross-Covariance Control), a simple yet theoretically grounded method that clusters gradient pairs *$(g',g)$* and applies covariance regularization within each cluster. This localized control effectively suppresses cross-covariance while preserving the TD objective’s lower bound. Empirical results on D4RL MuJoCo benchmarks demonstrate substantial performance gains—especially in low-data regimes—showing improved stability and data efficiency.

Although this issue may appear specific to small datasets, the underlying mechanism suggests that it will become even more significant as environments grow larger or more complex. When the environment’s state or action space expands while the dataset size remains fixed, coverage sparsity worsens, amplifying the harmful covariance effects. Hence, the insight and methodology presented here are expected to remain increasingly relevant as future offline RL benchmarks evolve toward more realistic and high-dimensional settings.

**Strengths:**

The paper impressively connects a theoretically grounded understanding of TD learning instability with a practical, well-performing algorithm. The C4 framework stands out for translating a nontrivial analytical insight into empirically robust improvements, showing both conceptual clarity and engineering maturity.

**Strong theoretical foundation**: The paper begins from a clear theoretical diagnosis of a subtle but critical problem in temporal-difference (TD) learning — the emergence of harmful cross-time covariance under distributional shift and limited data. The authors derive this effect formally and distinguish it from other, beneficial forms of implicit regularization. This analytical depth provides a precise and original explanation for instability in offline RL, rather than relying on empirical speculation.

**Elegant link from theory to practice**: One of the most compelling aspects is how the theoretical insight leads directly to a concrete and empirically effective solution. The transition from analyzing TD covariance to designing the Clustered Cross-Covariance Control (C4) algorithm shows an excellent bridge between mathematical reasoning and algorithmic implementation.

**Consistent and plentiful empirical validation and ablation studies**: The experiments on D4RL MuJoCo benchmarks demonstrate that the theory translates into tangible performance benefits: smoother learning curves and more than 30% improvement in return under small data and OOD-emphasized splits. Even though the authors do not deeply explore the optimal number of clusters, the empirical results with *$K=5$* show that the method works reliably across tasks without extensive tuning—indicating good robustness in practice.

Apart from that, the paper includes extensive ablation studies on different factors such as dataset size, which reinforce the soundness and robustness of the proposed approach.

**Weaknesses:**

I have only on weakness to point out.

**Sensitivity to the number of clusters K**:

The proposed method critically depends on the number of clusters used to partition the gradient feature space. If *$K$* is too small, dissimilar gradient modes are mixed, leaving between-cluster covariance unremoved; if *$K$* is too large, covariance estimation becomes noisy due to small sample counts per cluster. However, the paper fixes *$K$* (typically 5) for all experiments without analyzing sensitivity or scalability. It remains unclear how robust C4 is to this choice or how *$K$* should scale with data size or environment complexity.

**Questions:**

**Cluster sensitivity**:

How sensitive is performance to the number of clusters *$K$*? Did you observe degradation or instability for smaller/larger *$K$* values? Could an adaptive mechanism (e.g., based on gradient variance or cluster entropy) help automate cluster selection?

---

> ### Author Response · Authors · 2025-11-20
>
> We thank the reviewer for the positive assessment and constructive comments.
>
> -----
>
>
>
> **Q1: How sensitive is performance to the number of clusters $K$? Did you observe degradation or instability for smaller or larger values? It remains unclear how robust C4 is to this choice or how $K$ should scale with environment complexity.**
>
> We observe a clear but smooth dependence on the number of clusters. Performance improves quickly when increasing $K$ from very small values and then forms a broad plateau for moderate $K$. For larger $K$, returns decrease slightly but remain consistently above plain CQL, and we do not see any instability in training. This behavior is consistent across all seven benchmark families in Table 1, which span a wide range of state and action dimensions and dynamics. These results indicate that $C^4$ is **reasonably robust to the choice of $K$** and we **do not observe a need to scale $K$ with environment complexity**. A small constant $K$ works well across all tasks. Since $K=5$ lies in the middle of this stable high performance range, we simply fix this value in all main experiments.
>
> Table 1: Sensitivity of CQL+$C^4$ performance to the number of clusters $K$.
>
> | Method    | $K$  | Ant           | Halfcheetah | Hopper      | Walker2d      | Antmaze     | Pen           | Kitchen       |
> | --------- | ---- | ------------- | ----------- | ----------- | ------------- | ----------- | ------------- | ------------- |
> | CQL       |      | 21.0          | 39.7        | 43.2        | 26.0          | 21.1        | 1.8           | 0.6           |
> | CQL+$C^4$ | 1    | 59.3          | 41.3        | 68.3        | 51.2          | 55.1        | 29.8          | 11.7          |
> | CQL+$C^4$ | 2    | 66.5          | **48.2**    | 75.8        | 82.6          | 62.9        | 36.0          | 17.1          |
> | CQL+$C^4$ | 3    | 70.6          | 47.5        | **90.8**    | 87.8          | 67.3        | 51.3          | 19.5          |
> | CQL+$C^4$ | 5    | **71.3** (-0) | 46.0 (-2.2) | 85.3 (-5.5) | **96.3** (-0) | 68.7 (-3.7) | **58.2** (-0) | **20.9** (-0) |
> | CQL+$C^4$ | 7    | 69.6          | 46.4        | 82.1        | 94.9          | **72.4**    | 58.0          | 19.3          |
> | CQL+$C^4$ | 10   | 68.5          | 46.3        | 78.6        | 90.6          | 68.5        | 52.4          | 18.0          |
> | CQL+$C^4$ | 20   | 69.0          | 44.9        | 73.3        | 87.3          | 62.8        | 54.0          | 19.8          |
>
>
>
> ----
>
> **Q2: Could an adaptive mechanism (for example, based on gradient variance or cluster entropy) help automate cluster selection? How should $K$ scale with data size?**
>
> We have not designed a dedicated adaptive mechanism to choose the initial $K$. However, our empirical analysis offers key insights into how the number of clusters should scale with data size. Motivated by the stability of larger $K$ in Table 1, we analyzed the effective number of clusters when initializing with $K = 20$ (Table 2). We observe two main phenomena:
>
> **Implicit Shrinkage**: Regardless of the initialization, the model automatically prunes redundant clusters. Only about 2 to 6 clusters remain effective, while the rest receive negligible assignments.
>
> **Scaling with Data Size**: Interestingly, Table 2 reveals that the effective $K$ does not strictly increase with data volume. In the low-data regime (10k samples), the model utilizes slightly more clusters (avg. ~5) to cover the sparse support or higher variance. Conversely, with the full dataset ($\sim 10^6$ samples), the learned distribution consolidates into fewer, more robust modes (avg. ~3).
>
> This suggests that an adaptive mechanism should not simply scale $K$ proportionally to the data size, but should instead detect mode consolidation. We will incorporate these findings into a principled adaptive rule (e.g., using cluster entropy or gradient variance to prune or split clusters dynamically), which is a promising direction for future work.
>
> Table 2: Average number of effective clusters after convergence with initial $K = 20$.
>
> | Samples             | Ant  | Halfcheetah | Hopper | Walker2d | Antmaze | Pen  | Kitchen |
> | ------------------- | ---- | ----------- | ------ | -------- | ------- | ---- | ------- |
> | 10k                 | 6    | 4           | 4      | 5        | 5       | 5    | 6       |
> | $\sim 10^6$ samples | 3    | 2           | 2      | 3        | 4       | 3    | 3       |

---

> > ### Comment · Reviewer_Ykhx · 2025-11-26
> >
> > It is encouraging to observe that an effective number of clusters naturally emerges even when the total number of clusters is set to be large. This phenomenon is also supported by the ablation study, where the performance drop remains marginal once the number of clusters exceeds a sufficient threshold (around 5 to 7).
> >
> > I appreciate the authors’ thorough ablation study on this aspect. I would like to keep my scores.

---

> ### Author Response · Authors · 2025-11-26
>
> Thank you again for your valuable feedback!

---

### Comment · Area_Chair_uQVD · 2025-11-23
**Subject: Follow-up Reviews Required to Proceed**

Dear Reviewers,

The authors have submitted their rebuttal, and we now require your follow-up assessments to move the decision process forward. Please review the authors’ responses and update your evaluations accordingly.

Your prompt follow-up is necessary for us to finalize the meta-review.
Kindly submit your updates as soon as possible.

Best,
Area Chair

---

### Author Response · Authors · 2025-11-28
**Subject: Summary of Response by Authors**

We thank the reviewers for valuing the theoretical novelty and principled design of $C^4$, with Reviewer 99jv calling the work “impactful” and Ykhx praising the "impressive" connection between theory and practice. Beyond these encouraging remarks, we have substantially improved the manuscript by addressing their constructive feedback as follows:

- *Reviewers FJyE and ko2m questioned the comparison with recent baselines, specifically the absence of **A2PR (arXiv:2405.19909),** which was the main concern of Reviewer **FJyE**.*

  **Response:** We expanded our evaluation to include **A2PR** and **BPPO** across Locomotion, Antmaze, Adroit, and Kitchen benchmarks. The results demonstrate that our method consistently outperforms these baselines in data-scarce regimes and, as a plug-in module, significantly boosts A2PR and BPPO by **85.9%** and **39.6%**, respectively.

  &nbsp;

- *Reviewers FJyE and ko2m raised concerns regarding the coverage of our experimental benchmarks. Specifically, the inclusion of **sparse-reward tasks** was the primary concern of **ko2m**.*

  **Response:** We expanded our comprehensive analysis on Adroit, Maze2D, Kitchen, and **sparse-reward AntMaze** by incorporating recent baselines like A2PR and BPPO. Results demonstrate that our method outperforms the existing data-efficient methods by **~26%** and general-purpose offline RL algorithms by over **50%**

  &nbsp;

- *Reviewers FJyE and 99jv raised concerns regarding the evaluation on **large-scale datasets**, which was the primary concern of **99jv**.*

  **Response:** We demonstrated that adding $C^4$ boosts the total score by **6.7%** on full D4RL datasets ($\sim 10^6$ samples), directly addressing Reviewer 99jv's concern by showing that there are no performance tradeoffs required to achieve robust performance on small datasets.

  &nbsp;

- *Reviewers 99jv and ko2m requested deeper intuition regarding the harmful cross-covariance term, along with quantitative evidence linking it directly to performance degradation.*

  **Response:** We clarified that minimizing the TD error implicitly maximizes gradient coupling, which causes estimation errors from poorly covered regions to propagate backward, resulting in destructive gradient interference that destabilizes the critic. Moreover, we supported this with a new controlled study showing that our method reduces covariance trace by **20%-80%** while boosting returns by **10%-300%**, directly correlating covariance reduction with performance gains.

  &nbsp;

- *Reviewers Ykhx, 99jv, and FJyE inquired about sensitivity to hyperparameters, noting that sensitivity to $K$ was the sole concern of **Ykhx**.*

  **Response:** We conducted detailed sensitivity experiments on key hyperparameters **(specifically $K$ and $\lambda$)** across broad benchmarks (Locomotion, AntMaze, Maze2D, Adroit, Kitchen). The results confirm that performance is robust across a wide range of values, forming a stable plateau.

  &nbsp;

- *Reviewer FJyE inquired about the computational efficiency of the clustering process and update frequency.*

  **Response:** We clarified that the linear clustering complexity $O(KNm)$ results in negligible overhead **(~1%)** on small datasets and a manageable **15%–20%** on million-scale ones. Additionally, our ablation studies demonstrate that a moderate update frequency effectively amortizes the cost while maximizing returns.

  &nbsp;

- *Reviewers 99jv and FJyE raised technical questions regarding the validity of the first-order Taylor expansion assumption and the stop-gradient effect.*

  **Response:** We added a local Lipschitz assumption to justify the first-order approximation and clarified in detail that the harmful term effect persists regardless of target network fixing.

  &nbsp;

- *Reviewer 99jv noted that the extensive length of Sections 5 and 6 led to a very brief experiments section.*

  **Response:** We modified the notation and streamlined the exposition in these sections and moved **key experimental results from the Appendix to the main Experiments section**.

  &nbsp;

Overall, we integrated all the aforementioned updates into the revised manuscript, strengthening the work with over ten sets of new experiments covering broad benchmarks (Locomotion, AntMaze, Maze2D, Adroit, Kitchen), recent baselines (A2PR and BPPO), and detailed hyperparameter/efficiency analyses. Additionally, following Reviewer 99jv’s suggestion, we moved key results to the main text and explicitly clarified assumptions and empirical details in Appendices B and E, respectively.

---

> ### Author Response · Authors · 2025-11-29
> **Subject: Supplementary Statement by Authors**
>
> We sincerely thank all reviewers for their constructive feedback, which has helped us significantly improve the manuscript. We are particularly encouraged by the productive discussions during the rebuttal phase, which led Reviewer **99jv** to raise the score (6 $\to$ 8) and Reviewer **Ykhx** to maintain strong support (8).
>
> Crucially, we emphasize that these positive assessments and score updates occurred **prior** to the recent large-scale information leakage incident and that we strictly adhered to the double-blind policy throughout the review. We thank the Area Chairs and reviewers for maintaining a fair and rigorous evaluation during this challenging time.

---

### Meta-Review · Area_Chair_XMB2 · 2026-01-06

**Summary:**

The reviewers generally agree the paper provides a compelling diagnosis of offline RL instability: TD training induces harmful cross-time gradient cross-covariance, which can bias updates and trigger critic collapse under weak coverage / OOD regions. The proposed C4 control (clustered sampling + covariance penalty) is seen as simple, theoretically motivated, and empirically effective in small-data settings.
The remaining uncertainty is less about the core idea and more about completeness and credibility of the empirical story: whether performance holds beyond the 10k regime, whether comparisons include strong recent baselines, and whether the paper communicates the method and evidence clearly in the main text rather than relying on rebuttal/appendix.

**Reviewer Concerns:**

Concerns largely addressed in the rebuttal include: (i) sensitivity to the number of clusters and “effective clusters” behavior (supported by added ablations and reviewer acknowledgment), (ii) potential tradeoffs on full-scale datasets (additional results indicate no systematic degradation), and (iii) baseline strength and scope (added comparisons to stronger recent methods and expanded domains such as AntMaze/Adroit/Kitchen, plus efficiency and clustering-frequency analyses).
Outstanding items are mostly presentation and scope risks: the camera-ready must integrate the new results/ablations into the main paper with a coherent narrative, and claims about scalability to visual/high-dimensional inputs should remain conservative since evidence there is limited. Notation/section organization was also a recurring friction point and needs to be demonstrably cleaned up.

**Reviewer Scores:**

Reviewer Ykhx would likely keep their score unchanged, as their only substantive concern (K sensitivity / robustness) was directly answered and they explicitly indicated satisfaction during discussion.
Reviewer 99jv would likely increase by about one point given that the main open question (large-dataset tradeoff and broader empirical validation) was addressed with new full-dataset results and clearer comparisons. Reviewers FJyE and ko2m are plausibly moved from borderline to slightly positive if the camera-ready genuinely incorporates the rebuttal additions into the main text and tightens claims and exposition.

---

### Decision · Program_Chairs · 2026-01-26

Accept (Poster)